# Learning Robust EEG Representations with a Large Spatiotemporal Transformer as a Foundation Model

## Abstract

Electroencephalography (EEG)-based brain-computer interfaces (BCIs) serve many control paradigms by relying on a variety in active brain regions and EEG features. Developing a universal EEG foundation model[1] has been challenging due to the large variety in recording setups and experimental tasks. Additionally, researchers often contend with limited labeled data, making it difficult to utilize large deep-learning models effectively. While there have been successful attempts to develop EEG foundation models, few studies have systematically evaluated their adaptability across diverse BCI control paradigms. To address this gap, we propose a novel, yet simple spatiotemporal EEG transformer (ST-EEGFormer) that projects segments ("patches") of raw EEG data into an embedding space enriched with a spatial and temporal embedding, allowing the model to effectively handle EEG data exhibiting various channel set-ups and time lengths. To improve data efficiency, we first employed a masked autoencoder (MAE) task to pre-train the ST-EEGFormer in a self-supervised learning manner on a dataset combining six different motor imagery (MI) datasets, a P300 dataset, and a steady-state visual evoked potential (SSVEP) dataset, all of which are public. Next, we benchmarked the pre-trained model, after fine-tuning, on diverse downstream classification tasks. To evaluate the generalization capability, we conducted additional experiments on two public datasets, not used for pre-training: a seizure classification dataset and an online MI BCI dataset. We compared the performance against a simple linear model, EEGNet (a classic CNN-based benchmark model), the state-of-the-art supervised EEG Conformer model, and two foundation models, BIOT and Large Brain Model (LaBraM). The pre-trained ST-EEGFormers could learn robust EEG representations, achieving higher classification accuracies than the benchmarked models across all eight pre-training datasets and exhibiting strong generalization on new datasets with limited training data. Finally, we report several visualizations of the model including the features on which the results are based.

## 1 Introduction

Electroencephalography (EEG) is a non-invasive recording technique widely utilized in the development of Brain-Computer Interfaces (BCIs), the aim of which is to enhance the quality of life of disabled patients and to augment the performance of healthy individuals. Several EEG paradigms have been explored to facilitate BCI development. Motor imagery (MI), for instance, involves the mental simulation of physical movement and can be used to control various applications, such as exoskeletons (Soekadar et al., 2016; Choi et al., 2020), navigation in real or virtual environment (Choi & Cichocki, 2008; Tsui et al., 2011; Yang & Van Hulle, 2023), or assist in rehabilitation (Baniqued et al., 2021; Liao et al., 2023). Event-related potentials (ERPs), such as the P300 response—a positive potential elicited when a user experiences an infrequent event—have been employed to decode user attention and to develop smart home control applications (Holzner et al., 2009; Masud et al., 2017). Visual-evoked potentials (VEPs), including

---

[1]A model that is trained on broad data (generally using self-supervision with a large scale of data) that can be adapted (e.g., fine-tuned) to a wide range of downstream tasks (Bommasani et al., 2021)

steady-state visual-evoked potentials (SSVEPs), are EEG amplitude changes in response to visual stimuli, which in turn can be used to construct high-speed spelling devices (Nakanishi et al., 2018; Xing et al., 2018).

Although the aforementioned paradigms can be recorded with a standard EEG cap, they differ in spatial and temporal response patterns. This often implies training decoding models tailored to specific paradigms and applications. Traditional decoding methods include steps such as temporal filtering, spatial filtering (Blankertz et al., 2008), feature extraction (Singh & Krishnan, 2023), and classification using linear classifiers like linear discriminant analysis (LDA) or support vector machines (SVM) (Neto et al., 2016; Richhariya & Tanveer, 2018). These models typically require prior knowledge about the EEG signal. For example, the prime feature for MI is the temporal change in the mu and beta frequency bands, the extraction of which involves a pipeline of band-pass filtering, and the application of common spatial pattern (CSP) filtering, using variance as a feature (Ang et al., 2012). For P300, the time-locked feature is obtained by sampling the EEG amplitude over 500 ms post-onset of the infrequent event. For SSVEP, different flashing stimuli elicit EEG rhythm in sync with the stimulation frequencies, therefore canonical-correlation analysis (CCA) can be applied to occipital channels to find out the best-matching template representing the stimulation frequency (Lin et al., 2007).

Recently, deep learning has rapidly advanced in the BCI field, achieving state-of-the-art performance in various tasks. However, differences in BCI paradigms, recording devices, and the relatively small dataset sizes necessitate training models on individual tasks (Murad & Rahimi, 2024). Even within the same paradigm, significant subject variance hinders the development of a universal EEG decoder. Additionally, the small dataset sizes discourage the use of large deep-learning models. Consequently, small models that typically rely on convolutional neural networks (CNNs) impose restrictions on input shape (e.g., the number of EEG channels, and the number of samples), further complicating the use of different datasets as they usually exhibit different data formats. A few pioneering works have attempted to address these limitations. For instance, BIOT (Yang et al., 2023), a biosignal foundation model, was designed to learn from diverse biosignal data and handle missing data, while the Large Brain Model (LaBraM) (Jiang et al., 2024), an EEG foundation model pre-trained on large-scale EEG datasets, demonstrated superior performance in tasks such as seizure classification, emotion recognition, and gait prediction. However, the applicability of such models to diverse BCI paradigms with limited data remains unexplored.

**Contributions:** This paper introduces the ST-EEGFormer, a large transformer-based model for BCI applications. The model was pre-trained on 3 million 5-second EEG epochs sourced from diverse EEG-BCI datasets using the masked autoencoder (MAE) method (He et al., 2021). For the first time, we evaluated such a large model on various small BCI datasets, demonstrating promising results across six MI datasets, one P300 dataset, and one SSVEP dataset. Additionally, experiments conducted on a seizure dataset and an online MI BCI dataset—neither of which were included in the pre-training data—highlight the model's robust representation capabilities, particularly in scenarios with limited training data. This work provides a versatile and high-performing BCI model, offering valuable insights and tools for advancing BCI research across diverse tasks.

## 2 PROPOSED APPROACH

In this study, we propose the ST-EEGFomer model, which, when pre-trained using MAE, can be directly fine-tuned on other datasets. The pre-training task involves reconstructing the original EEG inputs from masked tokens (see figure 1 and Appendix A.2.4 for detailed descriptions). The motivation for this architecture is to address the following issues:
The architecture is designed to address the following issues:
*1):*Achieve end-to-end EEG decoding that is not vulnerable to variations in EEG data formats.
*2):*Demonstrate the feasibility of pre-training on combined diverse BCI tasks and show the effectiveness of the model fine-tuning in different small BCI classification tasks.
*3):*Benchmark the model on a variety of datasets including new datasets to provide a fair comparison with other state-of-the-art models.

To address issue 1, we adopted a spatiotemporal slicing and encoding approach, similar to methods explored in previous studies (Xie et al., 2022; Yang et al., 2023; Jiang et al., 2024), which encodes

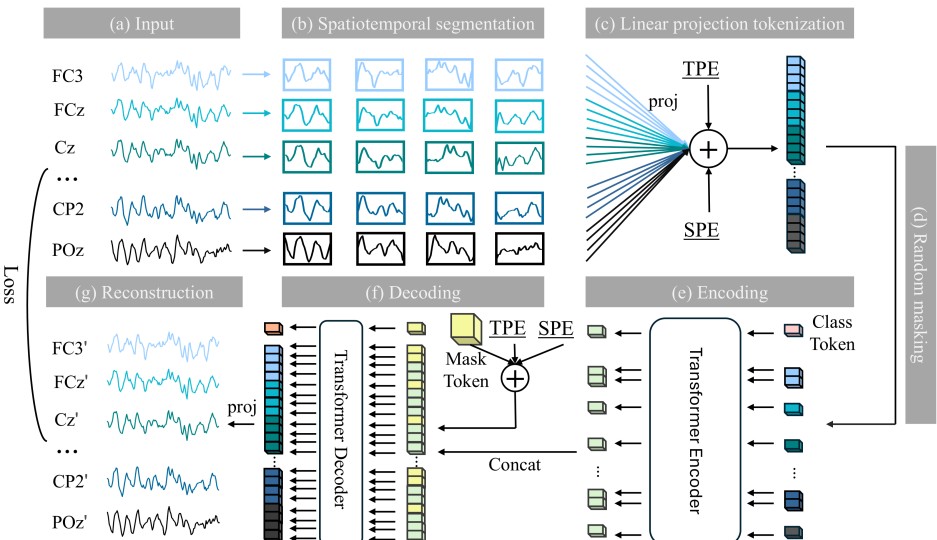

Figure 1: **Model architecture**: During pre-training, the input EEG data (a) are divided into segments along spatial and temporal dimensions (b). Each segment is tokenized through a linear projection layer (c), with each token receiving its corresponding temporal positional embedding (TPE) and spatial positional embedding (SPE). After randomly masking 75% of all tokens (d), the encoder (e) processes the remaining unmasked tokens. The mask tokens, with their added temporal and spatial positional embeddings, are then concatenated with the encoder output to form a full set of tokens. This full set of tokens is input to a small decoder (f), which reconstructs the original EEG signal. Once the model is pre-trained, only the encoder is utilized as the ST-EEGFormer model for fine-tuning on a downstream dataset.

individual channels rather than small patches consisting of all channels. The concept of slicing time series along temporal and spatial dimensions can be traced back to works such as the channel encoder proposed by (Mohsenvand et al., 2020), PatchTST (Nie et al., 2023), iTransformer (Liu et al., 2024), and the spatial-transformer combined with the temporal-transformer proposed by Xie et al. (Xie et al., 2022). This modeling approach has proven effective in learning from various biosignal sources, as demonstrated in BIOT (Yang et al., 2023) and the recent EEG foundation model LaBraM (Jiang et al., 2024). However, compared to these foundation models, the proposed model is characterized by its simplicity. Inspired by ViT (Dosovitskiy et al., 2021), it employs a straightforward linear projection for tokenization and a single-stage pre-training process focused solely on the MAE reconstruction task. A graphical representation of the proposed model is shown in figure 1.

Given prior successes with MAE in EEG decoding, we also selected MAE as the pre-training task. However, a distinction between our approach and other EEG-MAE attempts lies in the use of spatiotemporal segmentation, incorporating both spatial and temporal positional embeddings. While this spatiotemporal masking-and-reconstruction approach shares similarities with LaBraM, our model is simpler in both its architecture and pre-training procedures. To address issues 2 and 3, we pre-trained our model using data from eight different datasets collectively and then fine-tuned it on six MI classification tasks, one P300 task, and one SSVEP task. We compared the performance of our model against a simple common spatial pattern (CSP)-based linear model, the classic EEG benchmark model EEGNet (Lawhern et al., 2016), and the state-of-the-art EEG Conformer model (Song et al., 2023).

## 3 EXPERIMENTS

### 3.1 MAE PRE-TRAINING

Three different ST-EEGFormers (small, base, and large models) were pre-trained using 5-second EEG segments extracted from eight publicly available EEG-BCI datasets. The base and large models have the same architecture as the base, large models proposed in the ViT implementa-

Table 1: Details of ST-EEG-MAE variants, all with an EEG segment (patch) size of 16 samples and a mask ratio of 0.75.

| Model | Encoder layers | Encoder embed size | Encoder MLP size | Encoder heads | Decoder layers | Decoder embed size | Decoder MLP size | Decoder heads | Params |
|-------|-------|-------|-------|-------|-------|-------|-------|-------|-------|
| small | 8 | 512 | 2048 | 8 | 4 | 384 | 1536 | 16 | 32.7M |
| base | 12 | 768 | 3072 | 12 | 8 | 512 | 2048 | 16 | 110.9M |
| large | 24 | 1024 | 4096 | 16 | 8 | 512 | 2048 | 16 | 328.4M |

tion (Dosovitskiy et al., 2021), while the corresponding decoders have the same architecture as in the MAE implementation (He et al., 2021). Details about the proposed model can be found in table 1, and details of the datasets in Appendix A.2.1. Minimal preprocessing steps were applied to these datasets, including:

*1)*: Power-line noise filtering using a notch filter to remove power-line noise at 50 $Hz$ or 60 $Hz$.

*2)*: Broadband band-pass filtering with all channels filtered within the 1 to 64 $Hz$ range.

*3)*: Downsampling all datasets to 128 $Hz$.

Note: When benchmarking BIOT and LaBraM, we applied their respective preprocessing steps and recommended training strategies, resulting in identical trials but with a different sampling frequency (200 $Hz$). For further details, please refer to their publications. (Yang et al., 2023; Jiang et al., 2024) Detailed preprocessing implementations are provided in Appendix A.2.2 and A.2.3, the complete MAE pre-training methodology in Appendix A.2.4 and the experiment settings in Appendix A.2.5. The results of the pre-training are listed in Appendix A.7.

## 3.2 BENCHMARK ON MULTIPLE BCI TASKS

After MAE pre-training, each dataset was individually benchmarked using the pre-trained ST-EEGFormer and compared against several established models: a linear spatial filtering model that mimics the classic common spatial pattern (CSP) filtering approach (Zheng & Lu, 2015); EEGNet (Lawhern et al., 2016), a classic CNN-based backbone model known for its robust generalization across different BCI tasks; EEG Conformer (Song et al., 2023), which combines a CNN feature extractor with a transformer module, achieving state-of-the-art performance in motor imagery tasks. Additionally, we also compared our model with two open-source pre-trained EEG foundation models, BIOT (Yang et al., 2023) and LaBraM (Jiang et al., 2024). Note that for classification, the ST-EEGFormer could either use the class token or the average token as the feature to train a classification head. Meanwhile, one can perform end-to-end fine-tuning or freeze all layers and only train a linear head (linear probing). This study primarily focuses on motor imagery tasks to assess the model's performance. However, experiments with BIOT were not conducted on BCI-Comp-IV2a and BCI-Comp-IV2b datasets as the original channels could not be re-referenced to the required bipolar channels. Additionally, a P300 and an SSVEP dataset were included to introduce more variety during pre-training and to demonstrate the model's generalization capability across different classification tasks. For SSVEP, we also benchmarked the state-of-the-art SSVEP decoding model SSVEP DNN (Guney et al., 2022). Model details are described in Appendix A.3. For all experiments, population decoding with 5-fold cross-validation was performed, within each fold, 20% of the current training data were used as a validation set for model selection. If a hidden test set was available, it was used as an additional test set. We adopted a population decoding approach, where the model is trained on data from all available subjects to develop a subject-independent model. This approach is more computationally efficient, as per-subject training can be time-consuming and computationally intensive in a multi-subject study. Implementation details for this experiment can be found in Appendix A.4. The benchmark results for all MI datasets are presented in table 2. SSVEP dataset results and P300 results are presented in table 3 and figure 2. Detailed results are presented in Appendix A.8.

## 3.3 GENERALIZATION TO A NEW DATASET

To assess the generalization ability of the pre-trained models, two additional datasets were used. Detailed descriptions of the two datasets and tasks can be found in Appendix A.5.

Table 2: MI datasets benchmark results. "-cv" represents the average k-fold cross-validation accuracy, while "-test" represents the average accuracy on the hidden test set. The highest and second-highest accuracies are in bold, with the highest one marked in bold and surrounded by a box. For ST-EEGFormer, the default fine-tuning strategy is end-to-end fine-tuning with the average token, "lp" denotes a linear probed model, and "cls" refers to an end-to-end fine-tuned model using the class token.

| Dataset | Linear | EEGNet | EEG Conformer | BIOT | LaBraM | ST-EEGFormer small | ST-EEGFormer base | ST-EEGFormer base-lp | ST-EEGFormer base-cls | ST-EEGFormer large |
|---|---|---|---|---|---|---|---|---|---|---|
| EEG-MI-BCI-cv (Cho et al., 2017) | 0.683±0.007 | 0.781±0.011 | 0.821±0.012 | 0.718±0.020 | 0.736±0.010 | 0.905±0.020 | $\boxed{0.937 \pm 0.005}$ | 0.693±0.011 | **0.936 ± 0.010** | 0.931±0.005 |
| HGD-cv (Schirrmeister et al., 2017) | 0.631±0.017 | 0.899±0.010 | **0.914 ± 0.003** | 0.651±0.005 | 0.892±0.007 | 0.888±0.010 | 0.874±0.011 | 0.630±0.014 | 0.873±0.009 | $\boxed{0.954 \pm 0.004}$ |
| HGD-test (Schirrmeister et al., 2017) | 0.593±0.021 | 0.859±0.003 | 0.878±0.010 | 0.612±0.015 | **0.902 ± 0.040** | 0.858±0.011 | 0.838±0.006 | 0.579±0.014 | 0.817±0.007 | $\boxed{0.935 \pm 0.002}$ |
| BCI-Comp-IV2a-cv (Tangermann et al., 2012) | 0.436±0.030 | $\boxed{0.684 \pm 0.025}$ | 0.598±0.016 | *NA* | 0.381±0.020 | 0.502±0.025 | 0.480±0.024 | 0.435±0.022 | 0.489±0.018 | **0.673 ± 0.028** |
| BCI-Comp-IV2a-test (Tangermann et al., 2012) | 0.431±0.011 | $\boxed{0.651 \pm 0.011}$ | 0.566±0.017 | *NA* | 0.389±0.006 | 0.510±0.014 | 0.472±0.012 | 0.441±0.006 | 0.475±0.024 | **0.642 ± 0.004** |
| BCI-Comp-IV2b-cv (Tangermann et al., 2012) | 0.623±0.020 | 0.749±0.011 | $\boxed{0.777 \pm 0.014}$ | *NA* | 0.734±0.012 | **0.752 ± 0.026** | 0.737±0.029 | 0.696±0.011 | 0.751±0.009 | 0.692±0.040 |
| BCI-Comp-IV2b-test (Tangermann et al., 2012) | 0.697±0.011 | **0.810 ± 0.004** | $\boxed{0.831 \pm 0.005}$ | *NA* | 0.798±0.006 | 0.776±0.015 | 0.752±0.010 | 0.723±0.007 | 0.775±0.004 | 0.722±0.013 |
| Large-MI-Classic-cv (Kaya et al., 2018) | 0.442±0.009 | 0.644±0.004 | 0.722±0.004 | 0.455±0.012 | **0.763 ± 0.005** | **0.763 ± 0.008** | 0.754±0.006 | 0.439±0.004 | 0.731±0.004 | $\boxed{0.831 \pm 0.003}$ |
| Large-MI-5F-cv (Kaya et al., 2018) | 0.320±0.015 | 0.479±0.006 | **0.529 ± 0.004** | 0.287±0.008 | 0.464±0.023 | 0.500±0.008 | 0.483±0.010 | 0.294±0.008 | 0.462±0.003 | $\boxed{0.627 \pm 0.013}$ |

Table 3: SSVEP dataset benchmark results. The average accuracies from the leave-one-session-out experiment are reported. The highest and second-highest accuracies are in bold, with the highest one marked in bold and surrounded by a box. For ST-EEGFormer, the default fine-tuning strategy is end-to-end fine-tuning using the average token. Models denoted by "-cls" indicate end-to-end fine-tuned models utilizing the class token.

| Model | Window = 1s | | Window = 2s | |
|---|---|---|---|---|
| | Top1-Acc | Top2-Acc | Top1-Acc | Top2-Acc |
| Linear | 0.047 | 0.088 | 0.047 | 0.087 |
| EEGNet | 0.433 | 0.625 | 0.646 | 0.785 |
| EEG Conformer | 0.328 | 0.517 | 0.419 | 0.618 |
| BIOT | 0.316 | 0.449 | 0.492 | 0.627 |
| LaBraM | **0.518** | **0.669** | **0.700** | **0.818** |
| SSVEP-DNN | 0.385 | 0.570 | 0.442 | 0.606 |
| ST-EEGFormer-small | 0.387 | 0.551 | 0.441 | 0.604 |
| ST-EEGFormer-base | 0.218 | 0.344 | 0.217 | 0.342 |
| ST-EEGFormer-large | $\boxed{0.590}$ | $\boxed{0.748}$ | $\boxed{0.807}$ | $\boxed{0.893}$ |
| ST-EEGFormer-base-cls | 0.251 | 0.385 | 0.267 | 0.404 |

### 3.3.1 SEIZURE CLASSIFICATION

Firstly, we tested our approach on a single-channel seizure classification task using the famous Bonn dataset (Andrzejak et al., 2002). The hypothesis is that if the model learns robust EEG representations from normal EEG-BCI recordings during the pre-training step, it should be able to classify abnormal EEG data as well. Therefore, in the first experiment, we varied the amount of learning examples from only 5% to 60% and compared the classification accuracies among different models. In this experiment, we tested the performance of 1) directly applying linear probing on the pre-trained model; 2) directly fine-tuning the pre-trained model; 3) further calibrating the model by performing the MAE task, followed by linear probing on the seizure dataset, and 4) further calibrating the model by performing the MAE task and then fine-tuning on the seizure dataset. This was done to determine which approach yields the best performance. The results are presented in figure 3 a). The confusion matrix of the base model is shown in figure 3 b). Moreover, we also checked the effects of the mask ratio in the MAE pre-training step by varying the mask ratio and comparing the finetuned model and linear probing model performance under different mask ratios with only 5% training data. The results are presented in figure 3 c).

Figure 3 a) demonstrates that all pre-trained ST-EEGFormer models outperformed both EEGNet and Conformer, particularly when training data were limited. Moreover, performance could be further improved by calibration, as the highest accuracy was obtained by the ST-EEGFormer

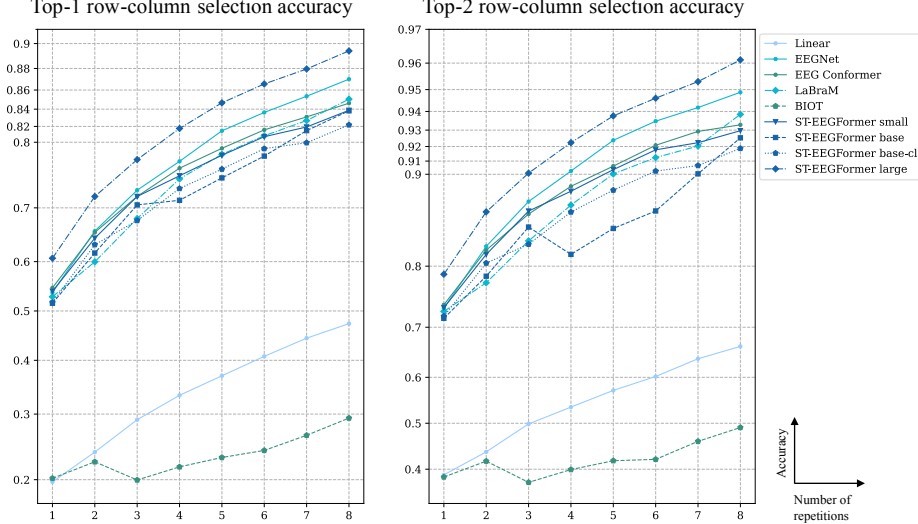

Figure 2: **P300 benchmark results**: Row-column selection accuracy of the P300 BCI. The original interface consists of 6 rows and 6 columns. A prediction is made for the row in which the attended character is present after all rows have flashed for the specified number of repetition rounds and, similarly, for the columns. EEG data of the same row or column, but from different repetition rounds, are averaged to create an averaged epoch for classification. The random chance level for selection accuracy is therefore 1/6.

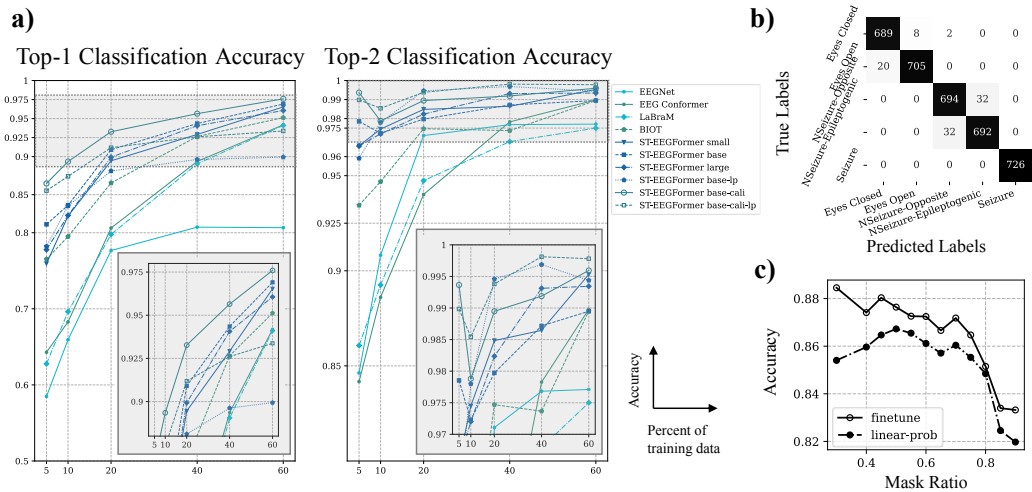

Figure 3: **a)**: Top-1 and top-2 classification accuracy on the seizure dataset with varying training data sizes, comparing EEGNet, EEG Conformer, fine-tuned ST-EEGFormer models (small, base, and large), and the linearly-probed base model (ST-EEGFormer base-lp). Additionally, the base model was further calibrated on the seizure dataset by performing the MAE SSL task using a mask ratio of 0.75, then fine-tuned and linearly-probed, referred to as ST-EEGFormer base-cali and ST-EEGFormer base-cali-lp, respectively. **b)**: Confusion matrix of the ST-EEGFormer base model trained with 60% of the data. **c)**: Accuracy of fine-tuned and linearly-probed ST-EEGFormer base-cali models with varying mask ratios during the calibration stage.

base-cali model. In contrast to results from MI datasets, where linear-probed models significantly underperformed finetuned models, the linear-probed models in this study achieved satisfactory results, especially after calibration, surpassing other models. This success can be attributed not only to the robust EEG representations learned during the MAE pre-training stage that help classify abnormal EEG data but also to the relatively straightforward classification task, which exhibits distinguishable characteristics that are easily visually inspected, making linear probing more effective. These findings provide a solid foundation for the future application of ST-EEGFormer in seizure classification, as the model could potentially learn even better representations from large open public seizure datasets not included in this study.

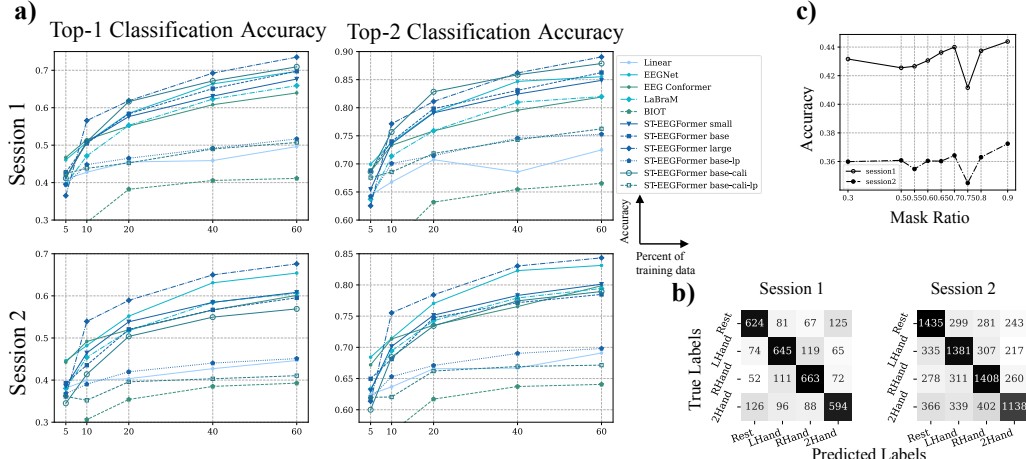

Figure 4: **a)**: Top-1 and top-2 classification accuracy on the 2-session MI dataset. Models were trained on the first session's training data with varying training data sizes and tested on both the first session's test data (first row) and zero-shot tested on the second session (second row). The comparison includes EEGNet, EEG Conformer, fine-tuned ST-EEGFormer models (small, base, and large), and the linearly-probed base model (ST-EEGFormer base-lp). Additionally, the base model was further calibrated on the seizure dataset by performing the MAE SSL task using a mask ratio of 0.75, then fine-tuned and linearly-probed, which is referred to as ST-EEGFormer base-cali and ST-EEGFormer base-cali-lp, respectively. **b)**: Confusion matrix of the ST-EEGFormer base model trained with 60% of the first session data with results reported for both sessions. **c)**: Accuracy of fine-tuned ST-EEGFormer base-cali models, with varying mask ratios during the calibration stage, with accuracies reported for both sessions.

### 3.3.2 TWO-SESSION ONLINE MI BCI CLASSIFICATION

In order to show the usability of the pre-trained models on future datasets or BCI applications, we validated the performance on a new MI BCI dataset (Stieger et al., 2021) not included in the initial MAE learning stage. This dataset contains EEG recordings of 4 classes of MI online tasks: imagined left hand- (LHand), imagined right hand- (RHand), imagined both hand movement (2Hand), and resting task (Rest). Similar to the seizure task described in Section 3.3.1, we varied the number of training examples to evaluate the model's performance with limited data. Additionally, we tested the models in a zero-shot manner on data from the second session to assess cross-session generalization ability. A detailed description of this dataset and experiment can be found in Appendix A.5.2. The results are presented in figure 4. Similar to the seizure classification experiments, the pre-trained ST-EEGFormer models demonstrated superior performance, particularly when training data was limited. The ST-EEGFormer large model consistently achieved the highest classification accuracy across all experiments. However, unlike the seizure classification results, the linearly-probed model did not perform as well, suggesting that motor imagery EEG data may contain more nonlinear features. Additionally, the cross-session experiment revealed a decrease in accuracy during zero-shot testing, indicating that the model still struggles with cross-session variability.

Finally, the two experiments yielded different optimal mask ratios: 0.3 for the single-channel seizure dataset and 0.9 for the 62-channel MI dataset, indicating varying behaviors that warrant further investigation.

## 4 DISCUSSION

### 4.1 PRE-TRAINING A LARGE EEG FOUNDATION MODEL FOR BCI TASKS

Results from table 2 demonstrate that the pre-trained ST-EEGFormer achieves the highest classification accuracies across multiple BCI datasets. The large model outperformed both the base and small models on most datasets. For all MI datasets, the finetuned base models exhibited higher accuracies compared to the linearly probed models, though the latter performed similarly to supervised linear models trained from scratch. Additionally, the linear model used as a baseline in this study was validated by comparison to existing benchmarks, as reported in figure 3 of (Gwon et al., 2023). The mean accuracies across all subjects using subject-specific linear models were comparable to our population models, achieving an average accuracy of 60%. This demonstrates the effectiveness

of the proposed approach, where performing self-supervised learning (SSL) pre-training on a foundation model with large EEG recordings enables the model to learn robust EEG features, comparable to a linear model.

Furthermore, table 3 and figure 2 also demonstrate the effectiveness of the proposed MAE-pre-trained ST-EEGFormer as a foundation model for general BCI decoding, outperforming EEGNet, and EEG conformer. However, the relatively poor performance on the SSVEP dataset requires further investigation. We acknowledge that the classification accuracies listed in table 3 are significantly lower than those reported in other studies. For instance, in the original SSVEP deep neural network (DNN) paper (Guney et al., 2022), using a 1-second window, the model achieved an accuracy exceeding 0.9. This substantial performance difference could be attributed to several factors:

*1)*: We only conducted population training, which is considered the first stage in (Guney et al., 2022). The reported accuracy is only around 47% with first-stage training alone (Guney et al., 2022).

*2)*: Typical SSVEP decoders utilize EEG data with filter banks, whereas in this study, we benchmarked our model on raw EEG data in an end-to-end manner, effectively using only one frequency band, which generally results in suboptimal performance, as shown in Table 1 of (Guney et al., 2022).

*3)*: The amount of training data was limited. Unlike other MI datasets, where subjects performed many trials per class, the SSVEP benchmark dataset contains only four trials per target. In the leave-one-trial classification setting, we used a sliding window approach to increase training examples, which also increases the risk of overfitting.

Despite these challenges, the ST-EEGFormer large model still achieved the highest performance, reaching above 80% accuracy with a 2-s window, suggesting potential future SSVEP applications. Possible ways to improve the model can be further pre-training the model with additional SSVEP datasets and exploring subject-specific fine-tuning strategies. Moreover, the large model performed well across all tasks, showing evidence of using deep, large models and demonstrating a successful approach to overcoming the challenge of limited data availability, which often hampers the development of deep learning models (Ahn et al., 2022; Dong et al., 2023; Khademi et al., 2023; Forenzo et al., 2024). The present study proposes a new approach for developing improved BCI decoders: pre-training the model on a large-scale BCI dataset composed of open public datasets with related tasks, followed by fine-tuning on smaller, application-specific datasets.

## 4.2 PERFORMANCE COMPARISON

When comparing the proposed foundation model with BIOT and LaBraM, it is evident that BIOT performs the worst, with some tasks—such as P300—even underperforming the linear model. This can be attributed to two primary factors. First, and most significantly, the pre-trained EEG model in BIOT uses only 18 bipolar channels. This limited spatial coverage hinders its ability to perform well in tasks requiring broader spatial coverage, such as in the case of P300, where the P300 component spans the occipital, central, and parts of the parietal regions. Second, BIOT's pre-training datasets primarily consist of clinical seizure, sleep, and resting-state EEG data, which differ substantially from EEG data associated with BCI tasks.

In contrast, LaBraM performs better than BIOT and ranks as the second-highest-performing model on several datasets, such as the HGD-test and the SSVEP dataset. This highlights the effectiveness of LaBraM's pre-training process, during which it was exposed to over 2,500 hours of EEG data, including several BCI-specific datasets. However, we acknowledge that this comparison is not entirely fair, as BIOT and LaBraM did not undergo the same pre-training steps on the datasets used in this study. Additionally, the pre-training datasets for our model were much smaller than those used for BIOT and LaBraM.

Another notable difference lies in model size: the open-sourced BIOT and LaBraM models are relatively small compared to the proposed ST-EEGFormer. The reported performance differences provide valuable insights into the importance of pre-training datasets and highlight the potential bias introduced by mismatches between pre-training datasets and downstream tasks. However, since a key motivation for foundation models is their generalization ability to unseen datasets, performance

on such datasets becomes more critical than the specifics of pre-training. This is particularly relevant when little training data is available for real-life BCI tasks.

Our experiments demonstrate superior generalization ability of the proposed ST-EEGFormer, which outperformed both BIOT and LaBraM on unseen datasets. Notably, LaBraM also outperformed classic models when limited training data was available, further demonstrating the effectiveness of a pre-trained model. Based on these findings, we recommend that future foundation model developers not only report performance on datasets included in their pre-training (as was done for BIOT) but also evaluate and compare performance on unseen datasets. This approach would encourage the development of models that generalize well across diverse and unseen BCI tasks, even when trained on different pre-training datasets.

### 4.3 Towards an interpretable model

#### 4.3.1 Learned EEG Channel Embeddings

First, we visualized the learned spatial embeddings after MAE pre-training by performing hierarchical clustering analysis using cosine similarity as the distance metric and by identifying different channel clusters in a topographic plot. The corresponding figures for the small, base, and large models are shown in Appendix figures I.1, I.2, and I.3. In all three models, a consistent pattern emerged with two clusters: one cluster in the front and another in the back. However, differences were observed when additional clusters were introduced. Although each model tended to learn slightly different clusters, these clusters generally corresponded to conventional functional mappings of EEG channels, with regions such as frontal, central, occipital, and temporal areas being identified by the model. This demonstrates the effectiveness of learning spatial information from EEG data using the proposed channel embeddings.

#### 4.3.2 Attention Weights Visualization for Classification

A typical way of analyzing deep learning-based EEG decoders is by drawing learned channel weights on a topo map (Cecotti & Graser, 2011; Lawhern et al., 2016; Salami et al., 2022; Song et al., 2023). Here we demonstrate that by looking at the attention matrix of the ST-EEGFormer, we can get more interesting visual interpretations directly from the raw EEG data. We performed the analysis on the seizure dataset (Andrzejak et al., 2002) and two MI-BCI datasets (EEG-MI-BCI (Cho et al., 2017) and Large-MI-5F (Kaya et al., 2018)) as the first shows a simple case of single-channel EEG data with visually distinguishable features while the two MI-BCI datasets involve multiple channels and have more subtle features that are not easy to spot.

**Single-Channel Seizure Classification Visualization** First, we visualized some representative trials using the attention rollout method (Abnar & Zuidema, 2020). The corresponding results are shown in Appendix figure I.4. Then we used the gradient rollout approach proposed in (Chefer et al., 2021) to visualize all EEG segments for class-specific explainability. The corresponding results are shown in Appendix figure I.5. Detailed descriptions can be found in Appendix A.6.

From figure I.4, we observe that the attention rollout with a discard ratio of 0.9 appears to be the most informative. A higher discard ratio highlights only a few segments, while a lower discard ratio tends to attend to all EEG segments, potentially diluting the focus on key areas. Interestingly, different head fusion methods highlight different regions, indicating that the model focuses on distinct features across different heads. Figure I.4 e) provides a clear example of how the model focuses on abnormal EEG signals, with each segment corresponding to a seizure spike being prominently highlighted. This finding suggests promising potential for explainable seizure detection and diagnosis applications.

In Figure I.5, the use of min-fusion often results in all segments being highlighted (as seen in figures I.5 a, b, c, e, g, h, i), due to the final rollout weights being too small. After scaling the weights to the range of 0–1, they become inflated, leading to a failure in identifying the most important segments. On the other hand, the mean and max gradient rollouts highlight different segments of

interest within the signal. However, since this dataset contains only selected signals with clear, continuous features, a more realistic evaluation is needed in future studies. Such an evaluation could leverage gradient rollout to detect specific intervals where one of the classes of interest occurs during continuous EEG monitoring.

**Multi-Channel MI-BCI Classification Visualization** For multi-channel EEG data classification, we first compared the learned spatial filters from the linear model, EEGNet, and EEG Conformer with the ST-EEGFormer attention rollout weights. A detailed description of this comparison can be found in Appendix A.6.5. The results are shown in figures I.6 and I.7. Figure I.8 to I.14 also visualize the attention weights on top of the raw EEG signal along with a spatiotemporal topo map.

As observed in the figures, all convolution-based kernels fail to capture spatially localized information, as each spatial filter is spread across the scalp, with multiple red regions indicating that the filters are less sparse compared to the attention rollout results from the ST-EEGFormer. In the ST-EEGFormer, important regions are clustered and sparse. It is difficult to draw meaningful conclusions from the spatial filters learned by the convolutional kernels, whereas the attention rollout results provide clear indications of important channels.

It is also worth noting that head fusion may not be the optimal approach for the analysis. Both the mean and max head-fused plots in figure I.6 d) show a similar pattern that closely resembles the pattern of head 11 in figure I.6 e). The differences observed between different heads in figure I.6 e) highlight that each attention head learns to focus on different regions of the scalp. Similar results can be obtained in figure I.7 d) and e), in which each head has a unique pattern, whereas the head fused patterns yield similar regions of interest.

## 4.4 LIMITATIONS AND FUTURE WORKS

Due to computational resource constraints, this study did not attempt to search for the most optimal combination of hyperparameters, such as embedding dimension size, the number of encoder layers, or the pre-training mask ratio. Additionally, to benchmark the effectiveness of the model, we limited our selection to eight datasets. Expanding the pre-training dataset, especially with high-density EEG data, could further enhance generalization. While this study demonstrated promising offline population classification results, future work should explore the model's online performance in real-world subject-dependent BCI settings. We believe the demonstrated effectiveness of the ST-EEGFormer will facilitate BCI decoders, and encourage the use of large models in the field. Furthermore, the experiment on seizure detection with intracranial data suggests a clinical application for monitoring epilepsy patients. Future works can focus on exploring different tokenization methods, developing lightweight models, and studying possible strategies to develop subject-dependent models from the pre-trained population model.

## 4.5 REPRODUCIBILITY

We open-source our codes and pre-trained model weights in the following repository: [...]. (Upon acceptance of this paper, we will make the link public)

## 5 CONCLUSION

We propose a novel yet simple ST-EEGFormer architecture as a foundation model for general EEG-based BCIs, designed to be insensitive to channel configurations and recording setups. The model leverages self-supervised pre-training on open public EEG datasets through a masked autoencoder task. Experimental results demonstrate that the proposed model outperforms the classic benchmark model (EEGNet), the state-of-the-art model (EEG Conformer), and two BCI foundation models (BIOT and LaBraM) in population decoding tasks across diverse BCI applications. Moreover, the ST-EEGFormer learned robust EEG representations that generalized effectively to unseen datasets, even with limited training data. Additionally, we analyzed the attention matrix for improved model interpretation and visualization. We believe this study establishes a strong foundation for future research into leveraging pre-trained large models to advance EEG BCI decoding performance.

ACKNOWLEDGMENTS

Temporally hiding for the double-blind review.

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

APPENDIX

## A.1 RELATED WORKS

**Supervised learning in BCI**

Early attempts in applying deep learning models to BCI decoding primarily focused on CNNs, utilizing 1D-convolutional kernels for spatial and temporal filtering. Notable architectures include ConvNet (Cecotti & Graser, 2011) for P300 classification and both shallow and deep ConvNets for motor imagery classification (Schirrmeister et al., 2017), EEGNet (Lawhern et al., 2016), which builds upon these earlier architectures, demonstrated superior performance across various EEG-BCI tasks, including P300, error-related negativity (ERN), event-related potentials (ERP), movement-related cortical potential (MRCP), and sensory-motor rhythm (SMR) (Lawhern et al., 2016). Since its introduction, EEGNet has been regarded as a classic benchmark model, inspiring numerous subsequent studies that incorporated specific modules to create its variants (Ingolfsson et al., 2020; Riyad et al., 2020; Salami et al., 2022). More recently, advances in attention-based transformer models from natural language processing, e.g., BERT (Devlin et al., 2019), and computer vision, e.g., ViT (Dosovitskiy et al., 2021), have inspired BCI researchers, among which is the EEG Conformer (Song et al., 2023), one of the most successful models as it achieved state-of-the-art performance on motor imagery and emotion datasets. However, these models still rely on convolutional kernels for feature extraction, making transferring them between different datasets cumbersome. the spatial filters in these models have a fixed size of $\mathbb{R}^{C \times 1}$, where $C$ represents the number of EEG channels, which can vary significantly across datasets. Additionally, these models employ temporal filters of size $\mathbb{R}^{1 \times T}$ and pooling layers to extract temporal features, making them dependent on the input size, which also varies between tasks and datasets. As a result, these models are typically trained on individual datasets and are not easily transferable to other datasets. Lashgari et al. (Lashgari et al., 2021) proposed an architecture to use CNN as a single-channel feature extractor followed by the attention mechanism to learn from datasets with varying channels. Spatiotemporal patching was explored by Xie, etal (Xie et al., 2022), in which the authors explored various ways of applying the transformer to a motor imagery dataset, achieving state-of-the-art performance. To solve the cross-subject variability issue, one can use a generative adversarial network (GAN) to learn invariant representations across subjects (Özdenizci et al., 2020), The same problem was also addressed using contrastive learning to learn subject-invariant EEG representations for cross-subject emotion recognition (Shen et al., 2021). Kobler et al. used multi-source batch normalization directly on the space of SPD matrices from EEG signals to reduce inter-subject variability issue (Kobler et al., 2022). Although these approaches show promising results, they still focus on specific tasks and remain challenging to transfer to other datasets and tasks.

**Self-supervised pre-training in BCI**

Several attempts have been made to utilize different sources of EEG data to create large datasets for representation learning. This typically involves pre-training tasks. For example, Jiang et al. applied contrastive learning for EEG-based sleep staging classification (Jiang et al., 2021). Mohsenvand et al. extended the SimCLR framework (Chen et al., 2020) and combined multiple datasets to learn data representations (Mohsenvand et al., 2020) by introducing channel encoders to process one channel at a time. However, their model was only tested on the SEED emotion dataset (Duan et al., 2013; Zheng & Lu, 2015), the normal vs. abnormal EEG dataset (López et al., 2015), and the sleep stage classification dataset (Goldberger et al., 2000), which are large, non-BCI-related datasets. BENDR (Kostas et al., 2021) employs contrastive learning to extract representations from raw EEG signals. However, as the model is adapted from Wav2vec 2.0 (Baevski et al., 2020), it faces challenges in transferring effectively between different datasets without discarding non-shared channels. Chien, et al. applied a masked autoencoder to learn EEG representations combined with a CNN-based feature extractor (Chien et al., 2022). Although the model performed well on the sleep stage dataset, it is still unknown whether it could perform well on small BCI datasets, and furthermore, the CNN feature extractor is hard to transfer between datasets. NeuroGPT faces a similar limitation, as its EEG encoder has a fixed channel constraint, making it difficult to adapt to varying channel configurations (Cui et al., 2024). BrainBERT (Wang et al., 2023) focuses on learning single-channel representations from stereo-electroencephalographic (SEEG) spectrograms,

enabling its use as a feature extractor for downstream tasks. More recently, Cai and Zeng (Cai & Zeng, 2024) employed MAE pre-training task on small segments consisting of all EEG channels across two motor imagery datasets, BCI Competition IV Dataset 2a and 2b (Tangermann et al., 2012) and the PhysioNet EEG Dataset (Goldberger et al., 2000). However, since their encoder module works with patches of EEG signals in fixed channels and time length, transferring between datasets remains unrealistic and challenging.

**BCI foundation models**

Biosignal Transformer (BIOT) (Yang et al., 2023) is the first biosignal encoding model that can handle biosignals of various formats. It was first pre-trained on multiple EEG, ECG, and sensory datasets by first dropping out channels and part of tokens from the remaining channels, and tried to predict the correct original tokens using a contrastive loss. The EEG datasets used in this study are resting-state, sleep EEG, and clinical seizure datasets, focusing on clinical EEG applications. Very recently, the Large Brain Model (LaBraM) (Jiang et al., 2024) was proposed to enable cross-dataset learning and was trained on 2500 hours of EEG data. The model pre-training involves two stages, first to train a temporal encoder and a neural tokenizer that compress the EEG signal into vector-quantized encodings by reconstructing the spectral amplitude and phase of input signals. Then in the second stage, the model is trained to reconstruct the encodings from a masked input. Although LaBraM was pre-trained using EEG data from various sources, the model was only benchmarked on TUAB (abnormal detection), TUEV (different event type classification), SEED-V (emotion recognition), and MoBI (gait prediction) datasets, all large datasets that are not classic BCI control paradigms. EEGFormer (Chen et al., 2024) was proposed which relies on a masked autoencoder approach to be pre-trained on the Temple University EEG Corpus (TUH Corpus) (Obeid & Picone, 2016) and onwards fine-tuned on downstream datasets. Although they reported successful results, their benchmarks were limited to high-quality clinical EEG datasets thereby focusing on similar clinical applications. Additionally, the number of channels was restricted, using only 24 or 36 channels from the TUH Corpus dataset. Moreover, this approach also required applying fast Fourier transformation (FFT) to obtain frequency domain amplitude as the input feature, rather than a pure end-to-end approach only using raw EEG signals as the input. A more detailed comparison of various models with our proposed model is presented in Appendix Table A.1. As highlighted in the table, the ST-EEGFormer stands out due to its simple architecture, straightforward pre-training process, focus on BCI applications, and ease of use.

## A.2 PRE-TRAINING

### A.2.1 PRE-TRAINING DATASETS

The MAE reconstruction task was conducted on eight public datasets. These datasets include:

*1) EEG-MI-BCI (Cho et al., 2017)*: This dataset contains 52 subjects performing 2-class imagined left and right-hand movements, recorded with 64 EEG channels using a Biosemi ActiveTwo system. It includes approximately 5,000 trials of 3-second motor imagery (MI) data per class.

*2) HGD (Schirrmeister et al., 2017)*: The High Gamma Dataset comprises 20 subjects performing 4-second trials of executed movements with four classes (left hand, right hand, both feet, and rest). The data were recorded with 128 high-density EEG caps (WaveGuard Original, ANT, Enschede, NL) and sampled at 5 kHz using a NeurOne amplifier (Mega Electronics Ltd, Kuopio, FI). It includes roughly 3,000 trials per class.

*3) BCI-Comp-IV2a (Tangermann et al., 2012)*: This dataset includes nine subjects performing four-second trials of four classes (imagined left hand, right hand, feet, and tongue movements), recorded with 22-electrode EEG caps. It contains a training set and a test set from two separate sessions, each with roughly 600 trials per class.

*4) BCI-Comp-IV2b (Tangermann et al., 2012)*: This dataset consists of nine subjects performing 2-class imagined left and right-hand movements, recorded with three EEG channels. It contains a training set of approximately 1,800 trials per class and a separate test set of approximately 1400 trials per class.

*5) Large-MI-Classic (Kaya et al., 2018)*: This dataset comprises 13 subjects performing 1-second trials of six classes (imagined left hand, right hand, left foot, right foot, tongue, and rest). The data were recorded with 19-channel EEG caps plus 2 ground lead channels (Electro-Cap International, USA) and were mostly sampled at 200 Hz, with some recordings sampled at 1000 Hz using the EEG-1200 system. In total, it includes approximately 50,000 trials (different classes have an unequal

number of trials).

*6) **Large-MI-5F** (Kaya et al., 2018)*: From the same study as 5) but different experiments, this dataset comprises 13 subjects performing 1-second trials of five classes of finger movements (imagined thumb, index, middle, ring, pinkie). In total, it includes around 18000 trials.

*7) **P300** (Won et al., 2022)*: This dataset consists of 55 participants performing a P300 speller experiment and 50 participants performing a rapid serial visual representation (RSVP). In total, it includes 99000 training P300 trials and 277200 test trials.

*8) **SSVEP** (Liu et al., 2020)*: This dataset consists of 70 participants performing cue-guided SSVEP target-selecting experiments, comprising 40 flickering stimuli ranging between 8 Hz to 15.8 Hz with an interval of 0.2 Hz. For each target, it contains 20 trials of 5-s stimulation data.

The MI datasets were selected based on their size and quality as reported in the latest review paper (Gwon et al., 2023), with datasets 3 and 4 chosen due to their status as classic benchmark datasets. The P300 dataset was included as it is one of the largest available P300 datasets, and the SSVEP dataset was selected for its widespread use as a benchmark in SSVEP research.

### A.2.2 DATA PREPROCESSING

All datasets underwent the following minimal preprocessing steps:

*1) Power-line noise filtering*: Visual inspection of the power spectrum density plots was conducted for each dataset. For those with visibly strong power-line noise, The function `mne.filter.notch_filter()` (Python 3.8.19, MNE 1.6.1) was applied to remove it.

*2) Broadband band-pass filtering*: A bandpass filter with cutoff frequencies at 1 Hz and 64 Hz was applied to all channels using `mne.filter.filter_data()` (Python 3.8.19, MNE 1.6.1). The filter was designed with a windowed FIR design (`fir_design='firwin'`).

*3) Downsampling*: All channels were downsampled to 128 $Hz$ from their original sampling frequency using the `mne.filter.resample()` function with default settings.

*4) Standardization*: Each channel was standardized to have a mean value of 0 and a standard deviation of 1.

### A.2.3 DATA SEGMENTATIONS FOR PRE-TRAINING

Some datasets provide continuous EEG recordings while some datasets contain only task-related epochs. Therefore, the following data segmentation approach was used to generate examples for pre-training:

*1)*: For EEG-MI-BCI (Cho et al., 2017), HGD (Schirrmeister et al., 2017), BCI-Comp-IV2a (Tangermann et al., 2012), BCI-Comp-IV2b (Tangermann et al., 2012), P300 (Won et al., 2022), these datasets consist of full continuous recordings for each participant. Pre-training examples were generated using a sliding window of 5 seconds in length with a hop size of 0.25 seconds, resulting in over 360,000, 70,000, 190,000, 380,000, and 620,000 EEG segments, respectively.

*2)*: For Large-MI-Classic (Kaya et al., 2018), and Large-MI-5F (Kaya et al., 2018), since these two datasets jointly represent one of the largest datasets used in this study, a 5-second sliding window with a 0.5-second hop size was used to balance the number of pre-training examples, instead of the previously used 0.25-second hop size. Combined, they contribute approximately 500,000 EEG segments.

*3)* For SSVEP (Liu et al., 2020), this dataset provides only 5-second stimulation epochs. A small sliding window of 2 seconds with a small hop size of 0.125 seconds was used to generate pre-training examples, resulting in approximately 170,000 EEG segments.

In summary, the dataset specs are summarized in table A.2.

### A.2.4 MAE PRE-TRAINING METHODOLOGY

The same pre-training methodology was used as in the original MAE paper (He et al., 2021). However, since multiple datasets were combined in this step, a subsampling strategy was used to generate a small batch of training examples from one dataset at a time and the loss was accumulated across all datasets that are available in the current iteration.

**Tokenization**: First, EEG data are sliced into non-overlapping small segments in both spatial and temporal dimensions (Figure 1, step b); segmented EEG data are then projected into an embedding dimension, adding the corresponding spatial positional- (SPE) and temporal positional embeddings (TPE) (Figure 1, step c). For the spatial positional embeddings, a learned embedding per channel was used, similar to the learned positional embedding in (Gehring et al., 2017), while for the temporal positional embeddings, a sine-cosine positional embedding approach was used, as shown in Eq A.1

$$p_t^{(i)} = f(t)^{(i)} := \begin{cases} \sin(\omega_k \cdot t), & \text{if } i = 2k \\ \cos(\omega_k \cdot t), & \text{if } i = 2k+1 \end{cases} \quad , \text{where} \quad \omega_k = \frac{1}{10000^{\frac{2k}{d}}} \quad \text{(A.1)}$$

**Masking**: Tokenized EEG data are randomly masked with a high masking ratio (Figure 1, step d), here we fixed the masking ratio to 0.75, as it has been reported to be the best-performing one (Chien et al., 2022). The remaining unmasked tokens pass through the transformer encoder.

**MAE Encoder**: The encoder is a ViT (Dosovitskiy et al., 2021) implemented the same way as in the original MAE paper (He et al., 2021). The encoder only initially processes the unmasked tokens concatenated with a class token (Figure 1, step e).

**MAE Decoder**: The encoder output is concatenated with mask tokens added with the spatial and temporal embedding of the corresponding masked segments (Figure 1, step f).

**Reconstruction Task**: The MAE reconstructs the input EEG by projecting the MAE decoder output (without the class token) back to segments of EEG samples and the loss is the mean squared error (MSE) between the original and reconstructed signals using the masked segments only, in the same way as in MAE (He et al., 2021) and BERT (Devlin et al., 2019).

Table A.1: Comparison of large EEG transformer models

| Model | Architecture | Pre-training Method | Pre-training Datasets | Benchmark Datasets | Channel Limitations |
|---|---|---|---|---|---|
| BENDR (Kostas et al., 2021) | 1D CNN with transformer layers | Mask tokens and predict unmasked output | Clinical EEG data | P300, ERN, sleep stage, motor imagery datasets | 20 channels |
| BrainBERT (Wang et al., 2023) | Transformer layers | Mask and reconstruct spectrograms | Self-collected SEEG data | Same SEEG dataset | Single channel |
| NeuroGPT (Cui et al., 2024) | Spatiotemporal CNN with self-attention and GPT | Mask and predict the next token | Clinical EEG data | One motor imagery dataset | 22 channels |
| EEGFormer (Chen et al., 2024) | Transformer layers | Extract spectral features, encode to codebook, mask, and reconstruct tokens | Clinical EEG data | Same dataset and sleep stage dataset | Not specified |
| BIOT (Yang et al., 2023) | Linear transformer layers | Contrastive loss for predicting correct tokens | Clinical, sleep, and resting EEG | Same dataset | 18 bipolar channels |
| LaBraM (Jiang et al., 2024) | CNN and transformer layers | Train temporal encoder and reconstruct spectral features, followed by masked token reconstruction | 2500 hours of EEG (clinical + others) | Seizure detection, abnormal EEG detection, emotion recognition, gait prediction | 120 EEG + 16 bipolar channels |
| ST-EEGFormer (Ours) | Transformer layers (ViT) | Masked autoencoder for reconstructing EEG signals | Diverse BCI EEG datasets | Diverse BCI EEG datasets | 128 channels |

Table A.2: Overview of the datasets used in the pre-training step.

| Dataset Name | No.Sub | No.Ch | Device | Classes | Sampling rate | Trial length | No.Trial | Separate test set | No.Pre-training examples |
|---|---|---|---|---|---|---|---|---|---|
| EEG-MI-BCI (Cho et al., 2017) | 52 | 64 | Biosemi ActiveTwo | 2 motor imagery: left, right hand | $512Hz$ | $3S$ | 5260 trials per class | No | 361641 |
| HGD (Schirmeister et al., 2017) | 20 | 128 | WaveGuard Original Caps, NeurOne amplifier | 4 executed movements: left hand, right hand, both feet, rest | $5kHz$ | $4S$ | 2811 trials per class | Yes: 560 trials per class | 68357 |
| BCI-Comp-IV2a (Tangermann et al., 2012) | 9 | 22 | NA | 4 motor imagery: left, right hand, feet, tongue | $250Hz$ | $4S$ | 648 trials per class | Yes: 648 trials per class | 189370 |
| BCI-Comp-IV2b (Tangermann et al., 2012) | 9 | 3 | NA | 2 motor imagery: left, right hand | $250Hz$ | $4S$ | 1840 trials per class | Yes: 1420 trials per class | 377479 |
| Large-MI-Classic (Kaya et al., 2018) | 13 | 21 | Electro-Cap International, EEG-1200 system | 6 motor imagery: left hand, right hand, left foot, right foot, tongue, rest | $200Hz, 1000Hz$ | $1S$ | 51632 trials per class | No | 360506 |
| Large-MI-5F (Kaya et al., 2018) | 13 | 21 | Electro-Cap International, EEG-1200 system | 5 finger motor imagery: thumb, index, middle, ring, pinkie | $200Hz, 1000Hz$ | $1S$ | 17998 trials per class | No | 137573 |
| P300 (Won et al., 2022) | 55 | 32 | Biosemi ActiveTwo | P300 trials, non-P300 trials | $512Hz$ | $1S$ | 18000 trials in total | Yes: 277200 trials in total | 620455 |
| SSVEP (Liu et al., 2020) | 70 | 64 | Synamps2 EEG system | 40 SSVEP classes | $250Hz$ | $5S$ | 700 trials in total | No | 171200 |

Table A.3: Pre-training settings.

| Config | Value |
|---|---|
| optimizer | AdamW (Loshchilov & Hutter, 2019) |
| base learning rate | 3e-4 |
| weight decay | 0.05 |
| batch size | 256 |
| learning rate schedule | cosine decay (Loshchilov & Hutter, 2017) |
| warmup epochs (Goyal et al., 2018) | 10 |

Table B.1: Linear CSP model architecture. Input EEG data consist of $N_{ch}$ channels and $L$ time samples. The output corresponds to $N_{class}$, representing the number of different classes to classify.

| Layer | Name | Type | Layer specific settings | Output shape |
|---|---|---|---|---|
| 0 | input | NA | NA | $(N_{ch} \times L)$ |
| 1 | spatial filter | Conv1d | kernel size:$(N_{ch}, 1)$ number of kernels:8 | $(8 \times L)$ |
| 2 | drop out | Dropout | $p$=0.2 | $(8 \times L)$ |
| 3 | feature extractor | NA | see table B.2 | $(8 \times 12)$ |
| 4 | flatten | NA | NA | $(1 \times 96)$ |
| 5 | classification head | Linear | weights and bias shape: $(96, N_{class})$ | $(1, N_{class})$ |

### A.2.5 MAE PRE-TRAINING SETTINGS

The pre-training settings are listed in table A.3. In the pre-training stage, the model is initialized using the `xavier_uniform` (Glorot & Bengio, 2010) method. We used the same codes from the original MAE (He et al., 2021) implementation, which uses the official ViT (Dosovitskiy et al., 2021) implementation and the linear lr scaling rule (Goyal et al., 2018) as shown in Eq A.2.

$$lr = base\_lr \times (batch\ size/256) \tag{A.2}$$

### A.3 MODEL IMPLEMENTATIONS

#### A.3.1 LINEAR NEURAL NETWORK MODEL FOR COMMON SPATIAL FILTERING (LINEAR MODEL)

The linear model consists of a spatial filter, a feature extractor, and a fully connected layer, without any non-linear activation functions between layers. This model serves as a simple linear baseline for comparison. The traditional CSP approach was not directly implemented for two reasons: first, numerous benchmarks using the traditional approach have already been performed, and second, this linear model allows for a fair comparison by utilizing the same gradient-backpropagation training approach as other networks. The model architecture is detailed in table B.1, and the calculated features are listed in table B.2.

#### A.3.2 EEGNET

EEGNet is a CNN-based model (Lawhern et al., 2016). The architecture of EEGNet is listed in table B.3. EEGNet is designed for general EEG classification tasks, which has shown to return good results on multiple EEG-BCI tasks, especially MI classification, and it has been widely used as a benchmark model.

#### A.3.3 EEG CONFORMER

EEG Conformer (Song et al., 2023) is a compact convolutional transformer that integrates local and global features within a unified EEG classification framework. The EEG input first passes through a convolutional module, which learns low-level local features via one-dimensional temporal- and

Table B.2: Features calculated in the feature extractor layer.

| Feature | Definition | Remark |
|---|---|---|
| mean | $\bar{x} = \frac{1}{n} \sum_{i=1}^{n} x_i$ | $n$: total number of samples, $x_i$: the $i$-th sample. |
| variance | $\sigma^2 = \frac{1}{n} \sum_{i=1}^{n} (x_i - \bar{x})^2$ | $\bar{x}$: the mean value |
| power | $P = \frac{1}{n} \sum_{i=1}^{n} x_i^2$ | NA |
| skewness | $\tilde{\mu}_3 = \frac{\sum_{i=1}^{n}(x_i - \bar{x})^3}{(n-1) \cdot \sigma^3}$ | $\sigma$: the standard deviation |
| kurtosis | $K = \frac{1}{n} \sum_{i=1}^{n} \left( \frac{x_i - \bar{x}}{\sigma} \right)^4$ | NA |
| entropy | $H = -\sum_{i=1}^{n} p(x_i) \log(p(x_i) + \epsilon)$ | $n = 256$: the number of intensity bins $p(x_i)$: the probability of the $i$-th intensity bin $\epsilon = 10^{-8}$: a small constant for stability |
| maximum | $\max(\mathbf{x}) = \max_{i=1}^{n} x_i$ | NA |
| minimum | $\min(\mathbf{x}) = \min_{i=1}^{n} x_i$ | NA |
| the first quartile | $Q_1 = \text{Quantile}(\mathbf{x}, 0.25)$ | NA |
| the secoond quartile | $Q_2 = \text{Quantile}(\mathbf{x}, 0.50)$ | NA |
| the third quartile | $Q_3 = \text{Quantile}(\mathbf{x}, 0.75)$ | NA |
| zero cross rate | $\text{ZCR} = \frac{1}{2n} \sum_{i=2}^{n} |\text{sgn}(x_i) - \text{sgn}(x_{i-1})|,$ $\text{sgn}(x) = \begin{cases} 1, & \text{if } x > 0 \\ 0, & \text{if } x = 0 \\ -1, & \text{if } x < 0 \end{cases}$ | NA |

Table B.3: EEGNet architecture. Input EEG data consist of $N_{ch}$ channels and $L$ time samples. The output corresponds to $N_{class}$, representing the number of different classes to classify. The dropout ratio is set to 0.40.

| Layer | Type | Input shape | Output shape | Kernels | Kernel size | Stride | Padding |
|---|---|---|---|---|---|---|---|
| 0 | input | $(N_{ch} \times L)$ | $(N_{ch} \times L)$ | NA | NA | NA | NA |
| 1 | Conv2d | $(N_{ch} \times L)$ | $(8 \times N_{ch} \times L)$ | 8 | $(1, 64)$ | $(1, 1)$ | same |
| 2 | BatchNorm2d | $(8 \times N_{ch} \times L)$ | $(8 \times N_{ch} \times L)$ | NA | NA | NA | NA |
| 3 | Depthwise Conv2d | $(8 \times N_{ch} \times L)$ | $(32 \times 1 \times L)$ | 32 | $(N_{ch}, 1)$ | $(1, 1)$ | $(0, 0)$ |
| 4 | BatchNorm2d | $(32 \times 1 \times L)$ | $(32 \times 1 \times L)$ | NA | NA | NA | NA |
| 5 | ELU | $(32 \times 1 \times L)$ | $(32 \times 1 \times L)$ | NA | NA | NA | NA |
| 6 | AvgPool2d | $(32 \times 1 \times L)$ | $(32 \times 1 \times L/4)$ | NA | $(1, 4)$ | $(1, 4)$ | $(0, 0)$ |
| 7 | Dropout | $(32 \times 1 \times L/4)$ | $(32 \times 1 \times L/4)$ | NA | NA | NA | NA |
| 8 | Seperable Conv2d | $(32 \times 1 \times L/4)$ | $(32 \times 1 \times L/4)$ | 32 | $(1, 16)$ | $(1, 1)$ | same |
| 9 | BatchNorm2d | $(32 \times 1 \times L/4)$ | $(32 \times 1 \times L/4)$ | NA | NA | NA | NA |
| 10 | ELU | $(32 \times 1 \times L/4)$ | $(32 \times 1 \times L/4)$ | NA | NA | NA | NA |
| 11 | AvgPool2d | $(32 \times 1 \times L/4)$ | $(32 \times 1 \times L/16)$ | NA | $(1, 4)$ | $(1, 4)$ | $(0, 0)$ |
| 12 | Dropout | $(32 \times 1 \times L/16)$ | $(32 \times 1 \times L/16)$ | NA | NA | NA | NA |
| 13 | Linear | $(1 \times 2L)$ | $(1 \times N_{class})$ | NA | NA | NA | NA |

spatial convolution layers. Next, the self-attention module extracts global correlations from the localized temporal features. Finally, a simple classifier module, consisting of fully connected layers, is used to predict the categories of the EEG signals. EEG Conformer has achieved state-of-the-art performance in both motor imagery and emotion classification tasks, making it a suitable representative high-performance model. The model architecture is listed in table B.4.

### A.3.4 BIOT AND LABRAM

The BIOT model used in this study is the version pre-trained on all six EEG datasets, titled "EEG-six-datasets-18-channels.ckpt", obtained from the official repository (https://github.com/ycq091044/BIOT).

The LaBraM model utilized in this study is the "labram-base.pth", retrieved from its official repository (https://github.com/935963004/LaBraM/tree/main).

For further details on the two models, please refer to their respective publications (Yang et al., 2023; Jiang et al., 2024).

Table B.4: EEG Conformer architecture. Input EEG data consist of $N_{ch}$ channels and $L$ time samples. The output corresponds to $N_{class}$, representing the number of different classes to classify.

| Layer | Name | Type | Layer specific settings | Output shape |
|---|---|---|---|---|
| 0 | input | NA | NA | $(N_{ch} \times L)$ |
| 1 | CNN-module | Conv2d | kernel size:$(1, 25)$ number of kernels:40 | $(40 \times N_{ch} \times L)$ |
| 2 | CNN-module | Conv2d | kernel size:$(N_{ch}, 1)$ number of kernels:40 | $(40 \times 1 \times L)$ |
| 3 | CNN-module | BatchNorm2d | NA | $(40 \times 1 \times L)$ |
| 4 | CNN-module | ELU | NA | $(40 \times 1 \times L)$ |
| 5 | CNN-module | AvgPool2d | kernel size:$(1, 37)$ stride:$(1, 7)$ | $(40, \lfloor \frac{L-37}{7} \rfloor + 1)$ |
| 6 | CNN-module | Dropout | dropout_p=0.5 | $(40, \lfloor \frac{L-37}{7} \rfloor + 1)$ |
| 7 | CNN-module | Conv2d | kernel size:$(1, 1)$ number of kernels:40 | $(40, \lfloor \frac{L-37}{7} \rfloor + 1)$ |
| 8 | Transformer-module | Transformer encoder layers | embed size:40 number of heads:10 drop_p:0.5 forward_expansion:4 forward_drop_p:0.5 depth:6 | $(40, \lfloor \frac{L-37}{7} \rfloor + 1)$ |
| 9 | Classification head | Linear | weights and bias shape: $(40 \times \lfloor \frac{L-37}{7} \rfloor + 1, 256)$ | $(1, 256)$ |
| 10 | Classification head | ELU | NA | $(1, 256)$ |
| 11 | Classification head | Dropout | dropout_p=0.5 | $(1, 256)$ |
| 12 | Classification head | Linear | weights and bias shape: $(256, 32)$ | $(1, 32)$ |
| 13 | Classification head | Elu | NA | $(1, 32)$ |
| 14 | Classification head | Dropout | dropout_p=0.3 | $(1, 32)$ |
| 15 | Classification head | Linear | weights and bias shape: $(32, N_{class})$ | $(1, N_{class})$ |

### A.3.5  SSVEP DNN

In this study, we worked with broadband EEG signals, which are rarely used in SSVEP decoding, as frequency-band filtered EEG signals are typically used as input. To ensure a fair comparison, we also benchmarked the state-of-the-art SSVEP deep neural network (DNN) model (Guney et al., 2022) under this setting. The implemented model architecture is detailed in table B.5. To obtain the necessary frequency-band signals, as required by the DNN model, we adapted the same frequency-band filtering approach used in EEGNet by applying a `Conv1D` in the first layer. In the experiment, we set the number of frequency bands (`no_fb`) to 3, the number of combined channels (`no_comb_ch`) to 120, the first dropout ratio (`dropout_ratio_1`) to 0.2, and the second dropout ratio (`dropout_ratio_2`) to 0.9. The number of classes ($N_{class}$) was set to 40.

## A.4  BENCHMARK EXPERIMENTS

### A.4.1  DATASET SPLIT AND MODEL SELECTION

For all MI datasets, we employed a 5-fold cross-validation strategy, using the `StratifiedKFold` function from `sklearn.model_selection` to ensure class balance within each fold. This approach was applied individually to each recording. During the 5-fold cross-validation, 4 folds are used as the current training set, and the remaining set is the test set for this fold:
*1) Training and Validation Split*: For each fold, 20% of the training data was set aside as a validation set, used for model selection.
*2) Model Selection*: The model achieving the highest classification accuracy on this validation set was chosen as the best model during different training epochs.

Table B.5: SSVEP DNN architecture. $N_{ch}$ represents the number of EEG channels, $L$ represents the number of time samples.

| Layer | Type | Input shape | Output shape | Kernels | Kernel size | Stride | Padding |
|---|---|---|---|---|---|---|---|
| 0 | input | $(N_{ch} \times L)$ | $(N_{ch} \times 1 \times L)$ | NA | NA | NA | NA |
| 1 | Conv1d | $(N_{ch} \times 1 \times L)$ | $(N_{ch} \times no\_fb \times L)$ | $no\_fb$ | $(1, 33)$ | 1 | same |
| 2 | reshape | $(N_{ch} \times no\_fb \times L)$ | $(no\_fb \times N_{ch} \times L)$ | NA | NA | NA | NA |
| 3 | Conv2d | $(no\_fb \times N_{ch} \times L)$ | $(1 \times N_{ch} \times L)$ | 1 | $(1, 1)$ | $(1, 1)$ | $(0, 0)$ |
| 4 | Conv1d | $(1 \times N_{ch} \times L)$ | $(1 \times no\_comb\_ch \times L)$ | no\_comb\_ch | 1 | 1 | 0 |
| 5 | Dropout1 | $(1 \times no\_comb\_ch \times L)$ | $(1 \times no\_comb\_ch \times L)$ | NA | NA | NA | NA |
| 6 | Conv1d | $(1 \times no\_comb\_ch \times L)$ | $(1 \times no\_comb\_ch \times L/2)$ | no\_comb\_ch | 2 | 2 | 0 |
| 7 | Dropout1 | $(1 \times no\_comb\_ch \times L/2)$ | $(1 \times no\_comb\_ch \times L/2)$ | NA | NA | NA | NA |
| 8 | ReLU | $(1 \times no\_comb\_ch \times L/2)$ | $(1 \times no\_comb\_ch \times L/2)$ | NA | NA | NA | NA |
| 9 | Conv1d | $(1 \times no\_comb\_ch \times L/2)$ | $(1 \times no\_comb\_ch \times L/2)$ | no\_comb\_ch | 10 | 1 | same |
| 10 | Dropout2 | $(1 \times no\_comb\_ch \times L/2)$ | $(1 \times no\_comb\_ch \times L/2)$ | NA | NA | NA | NA |
| 11 | Flatten | $(1 \times no\_comb\_ch \times L/2)$ | $(1 \times no\_comb\_ch * L/2)$ | NA | NA | NA | NA |
| 12 | Linear | $(1 \times no\_comb\_ch * L/2)$ | $(1 \times N_{class})$ | NA | NA | NA | NA |

Table C.1: Model training settings for the linear model, EEGNet, EEG Conformer, and SSVEP DNN.

| Config | Value |
|---|---|
| optimizer | AdamW |
| base learning rate | 3e-3(Linear, EEGNet), 3e-4 (EEG Conformer, SSVEP DNN) |
| weight decay | 0.05 |
| Optimizer momentum | $\beta 1, \beta 2 = 0.9, 0.999$ |
| batch size | 64 |
| training epochs | 300 |

*3) Testing*: The selected model was then evaluated on the test set of the current fold.

Additionally, for the HGD (Schirrmeister et al., 2017), BCI-Comp-IV2a (Tangermann et al., 2012), and BCI-Comp-IV2b (Tangermann et al., 2012) datasets, separate hidden test sets were available. These hidden test sets were used as additional test sets in this study, and the models selected from the cross-validation step were further evaluated on these sets to assess their performance comprehensively.

For the SSVEP (Liu et al., 2020) dataset, we followed the approach outlined in the SSVEP DNN paper (Guney et al., 2022), using a sliding window method to generate training samples of 1-second and 2-second lengths, with a hop size of 0.1 seconds. The test set also contains small segments of EEG data generated using the same sliding window on the hidden test trial data. We employed the same leave-one-session-out validation strategy for the experiment, as in (Guney et al., 2022), and the model selection process was consistent with that used in the MI experiments.

For the P300 (Won et al., 2022) dataset, we utilized the provided training and test sets. As in other P300 decoding experiments, we evaluated the model's performance under varying numbers of trial averaging. These trials were averaged based on the flashing of rows and columns during the experiment.

Remark that the training data in each fold were kept the same when training different models.

### A.4.2 MODEL TRAINING SETTINGS

The linear model, EEGNet, EEG Conformer, and SSVEP DNN were trained using the settings listed in table C.1. End-to-end fine-tuning of the ST-EEGFormer model followed the common practice of supervised ViT training, with the specific settings outlined in table C.2. Similar to the original MAE implementation that follows the practice laid out in (Chen et al., 2021), when linear probing the ST-EEGFormer model, in which case we froze all pre-trained weights and only trained a linear end stage of the model. The linear probing settings are provided in table C.3. It is important to note that we also tested using the same fine-tuning settings from table C.2 to train other models; however, this approach did not empirically yield better performance than the settings provided in table C.1. Specifically, the linear model and EEGNet required a larger learning rate to achieve convergence.

Table C.2: ST-EEGFormer end-to-end fine-tuning settings.

| Config | Value |
|---|---|
| optimizer | AdamW |
| base learning rate | 3e-4 |
| weight decay | 0.05 |
| Optimizer momentum | $\beta_1, \beta_2 = 0.9, 0.999$ |
| layer-wise lr decay (Bao et al., 2022; Clark et al., 2020) | 0.75 |
| batch size | 128 |
| learning rate schedule | cosine decay |
| warmup epochs | 5 |
| training epochs | 60 |
| label smoothing (Szegedy et al., 2015) | 0.1 |
| drop path (Huang et al., 2016) | 0.1 |

Table C.3: ST-EEGFormer linear probing settings.

| Config | Value |
|---|---|
| optimizer | AdamW |
| base learning rate | 0.05 |
| weight decay | 0 |
| Optimizer momentum | $\beta_1, \beta_2 = 0.9, 0.999$ |
| batch size | 128 |
| learning rate schedule | cosine decay |
| warmup epochs | 10 |
| training epochs | 100 |
| label smoothing | 0.1 |

## A.5 GENERALIZATION TO A NEW DATASET

### A.5.1 SEIZURE DATASET EXPERIMENT

#### A.5.1.1 DATASET DESCRIPTION

This Bonn dataset (Andrzejak et al., 2002) consists of human expert-selected single-channel EEG data from five healthy volunteers and five individuals with epilepsy. The data are divided into two classes for healthy volunteers, including scalp EEG segments recorded while the volunteers were relaxed and awake with eyes closed and open, respectively (Dataset A and B, referred to as "Eyes Closed" and "Eyes Open" in figure 3b). Three classes of data are from epileptic patients, consisting of intracranial EEG (iEEG) segments recorded during pre-surgical evaluation. Specifically, one class contains interictal iEEG segments from the epileptogenic zone in the opposite hemisphere (dataset C, referred to as "NSeizure-Opposite" in figure 3b), while another class includes interictal iEEG segments from the epileptogenic zone itself (dataset D, referred to as "NSeizure-Epileptogenic" in figure 3b). The final class consists of iEEG segments recorded from the epileptogenic zone during seizure activity (dataset E, referred to as "Seizure" in figure 3b). Each subset contains 100 single-channel EEG segments, each 23.6 seconds in duration (4096 samples). The data were sampled at 173.61 $Hz$, and any artifacts caused by muscle activity or eye movement were manually removed by the database owners after visual inspection.

#### A.5.1.2 EXPERIMENT DETAILS

In this experiment, the objective was to evaluate the pre-trained model's performance on previously unseen abnormal EEG data. Two conditions were considered: first, using the pre-trained model directly; and second, further calibrating the model on the new dataset by performing the MAE pre-training task. For the first condition, we employed the pre-trained small, base, and large models, which underwent end-to-end average token fine-tuning. Additionally, we applied linear probing on the base model with the average token. For the second condition, we first performed the MAE pre-training task on the base model for 20 iterations, followed by end-to-end average token fine-tuning and average token linear probing. To better evaluate model performance, we varied the amount of training data from 5% to 60% of the total available data. With only 5% of the training data, there were 225 training examples across 5 classes. Additionally, we experimented to assess the effect of the mask ratio. In this experiment, the base model underwent several MAE calibration tasks with varying mask ratios and was subsequently fine-tuned and linearly probed on the same task as the previous experiment.

### A.5.1.3 DATA PREPROCESSING

Minimal data preprocessing was performed to evaluate the model's generalization ability. First, all data were downsampled to $128 \ Hz$, then standardized to have zero mean and unit standard deviation to align with the data features encountered during the pre-training step. It is important to note that the original data were band-pass filtered by the data provider between 0.5 and $85 \ Hz$, which differs from our preprocessed range of 1 to $64 \ Hz$ during the initial model pre-training stage. We chose to retain the original band-pass settings to better assess the model's generalization ability and the impact of calibration. For MAE calibration, each 23.6-second recording was segmented into 3-second windows using a sliding window with a 0.25-second hop size, resulting in a total of 41,500 calibration examples. For classification, each 23.6-second recording was similarly divided into 3-second segments, but with a 0.5-second overlap, producing a total of 4,500 classification examples.

### A.5.1.4 DATA SPLIT AND MODEL TRAINING SETTINGS

In the classification experiment, we varied the amount of training data to 5%, 10%, 20%, 40%, and 60% of the total dataset. These training samples were selected using a stratified split method to maintain class balance, implemented with the train_test_split function from sklearn.model_selection. It is important to note that since the dataset does not provide channel information, we adapted the ST-EEGFormer model to be channel-unaware. In this adaptation, the tokens only incorporate temporal positional embeddings (TPE, as shown in figure 1), without the inclusion of spatial positional embeddings (SPE, as depicted in figure 1). This modification allows the model to function as a general EEG feature extractor, disregarding channel differences. Since the dataset consists of only single-channel recordings, the linear model based on spatial filtering could not be applied. Therefore, we compared ST-EEGFormer with EEGNet and EEG Conformer, using the same training settings as listed in tables C.1 to C.3.

### A.5.2 TWO-SESSION ONLINE MI BCI EXPERIMENT

#### A.5.2.1 DATASET DESCRIPTION

This dataset (Stieger et al., 2021) contains 600 hours of 62-channel EEG recordings, sampled at $1000 \ Hz$, collected during online and continuous BCI control from 62 healthy adults, spanning multiple sessions across different days. The BCI paradigm involves imagining left, right, and both hand movements (opening and closing), as well as a resting state condition, to control a virtual cursor. The provided data consists of epoched trials of varying lengths, structured with a 2-second inter-trial interval, followed by a 2-second target presentation. The task imagination phase varies in length, with a maximum duration of up to 6.04 seconds, followed by a 1-second post-trial interval.

#### A.5.2.2 EXPERIMENT DETAILS

Slightly different from the seizure classification experiment, where the objective was to assess whether the learned normal EEG representations could transfer and aid in abnormal EEG classification, this experiment aimed to determine whether the pre-trained model could be easily transferred to a new MI task and perform well across sessions, which can help future BCI applications if applicable. To address this, we conducted the following two-step experiment: First, we trained our models using data from the first session, varying the amount of training data in the same manner as in the seizure classification experiment, and evaluated the model's performance on the remaining data from the first session. Then, for each model, we performed zero-shot testing on all data from the second session to assess cross-session performance. For the base model, we also evaluated the effects of calibration and mask ratio, as done in the seizure experiment.

#### A.5.2.3 DATA PREPROCESSING

First, we selected the last two session recordings of 20 subjects (subjects 1–20) out of 60 total subjects for this experiment to reduce computational load. For each recording, the same preprocessing steps as described in Appendix A.2.2 were applied. For MAE calibration, instead of using all available data, we only utilized the non-task resting-state 2-second signals preceding each trial ($-2s$ to $0s$). This approach reflects a more realistic BCI scenario, where little or no

labeled data are available. This resulted in a total of 18,000 pre-training examples across the two sessions. For classification, we used the first 3-second segment of the imagination phase, yielding 9,000 classification examples per session.

#### A.5.2.4 DATA SPLIT AND MODEL TRAINING SETTINGS

The same data split method used in the seizure classification experiment was applied here. However, since this dataset includes channel information, we utilized the standard ST-EEGFormer implementation as described in figure 1. The same training settings outlined in tables C.1 to C.3 were also employed for this experiment.

### A.6 MODEL VISUALIZATION

The model used in the following experiments is ST-EEGFormer base finetuned on the corresponding dataset using the class token.

#### A.6.1 ATTENTION ROLLOUT

Attention rollout tracks the information flow from the input layer to the final layer in a transformer model through Eq E.1 (Abnar & Zuidema, 2020), where, $\tilde{A}$ is the attention rollout, and $A(l_i)$ the raw attention matrix in layer $i$. In order to focus on the most important tokens while ignoring less relevant ones, we apply a discard ratio that retains only the largest rollout weights at each layer. For instance, a discard ratio of 0.9 will keep only the top 10% of the largest weights, setting the remaining weights to zero. After calculating the rollout, each head produces a weight matrix. The final weights are obtained by fusing the weights across different heads, using one of the following methods: mean fusion, where the final weight is the average of all head weights; max fusion, where the final weight is the maximum value across all heads; and min fusion, where the final weight is the minimum value across all heads.

$$\tilde{A}(l_i) = \begin{cases} (A(l_i) + I)\tilde{A}(l_{i-1}) & \text{if } i > 1 \\ A(l_i) + I & \text{if } i = 1 \end{cases} \tag{E.1}$$

#### A.6.2 GRADIENT ATTENTION ROLLOUT

The gradient rollout method for attention visualization in class-specific explanations aims to interpret the contribution of individual input features to the model's decision for a particular class. Similar to standard attention rollout, it recursively calculates a weighted rollout score, but in this case, the gradients with respect to the class of interest are used as weights. The gradients are obtained by selectively backpropagating the output score corresponding to the chosen class. This process replaces the $A(l_i)$ term in Eq E.1 with a weighted term, where the attention value $A_{ij}$ is multiplied by the corresponding gradient $grad_{ij}$.

#### A.6.3 SEIZURE CLASSIFICATION VISUALIZATION

#### A.6.3.1 ATTENTION ROLLOUT

In the seizure classification experiment, each signal contains 384 samples, which, after patching and tokenization, results in 24 tokens. The attention matrix, therefore, has a shape of $(25 \times 25)$, with an additional class token concatenated at the beginning. After calculating the final attention rollout matrix using Eq E.1, the first row of the matrix was extracted for visualization, as it represents how the class token "attends" to different patches of the signal. Before visualization, the weights corresponding to the patches were normalized by dividing them by the maximum weight, scaling them between 0 and 1.

#### A.6.3.2    GRADIENT ATTENTION ROLLOUT

Using the same trials as visualized in figure I.4, we performed a gradient attention rollout analysis to check the importance of each patch corresponding to the top-2 model prediction classes. For this comparison, we only used a discard ratio of 0.9, compared with the three head fusion methods.

### A.6.4    MULTI-CHANNEL MI CLASSIFICATION VISUALIZATION

We used the MI-BCI dataset (Cho et al., 2017) and the Large-MI-5F dataset (Kaya et al., 2018) for the following multi-channel MI classification visualization.

### A.6.5    COMPARE SPATIAL FILTERS WITH ATTENTION ROLLOUT RESULTS

For the linear model and EEGNet, we simply extracted the learned weights of the spatial filters. Specifically, for the linear model, we used the weights of the first `Conv1D` layer (table B.1). For EEGNet, the weights from the `Depthwise Conv2D` layer (table B.3) were used, for the EEG Conformer the weights of the second `Conv2D` layer (table B.4). For the ST-EEGFormer model, we calculated the average attention rollouts for each class by following these steps:

*1)*: For all testing trials, we computed the attention rollout for each trial using Eq E.1, with a discard ratio of 0.9. The single-trial attention rollouts for the same class were averaged to obtain a class-specific average attention rollout.

*2)*: For each class, an attention rollout matrix $\tilde{A}$ of shape $(Seq + 1 \times Seq + 1)$ was generated. As in the analysis in Appendix A.6.3.1, the first row $\tilde{A}_1$ was extracted to evaluate how the class token attends to other tokens.

*3)*: We reshaped $\tilde{A}_1$ (excluding the first element, which corresponds to the class token itself) into a spatial-temporal score matrix $S_{i,j} \in \mathbb{R}^{\text{channel} \times \text{number of patches}}$. The weight scores per channel were obtained by averaging $S_{i,j}$ over the number of patches.

Before visualizing the weight vector on a topographic map, we scaled the weights by first subtracting the mean and then dividing by the maximum absolute value (Eq E.2), so the final weights were scaled between -1 and 1.

$$\bar{w} = w - \text{mean}(w)$$
$$w_{\text{scaled}} = \frac{\bar{w}}{\max(|\bar{w}|)} \tag{E.2}$$

## A.7    PRE-TRAINING RESULTS

The pre-training learning curves of the small, base, and large models are shown in figure F.1. The large model had the lowest loss at 100 epochs, followed by the small model, while the base model had the largest loss. It can be observed that the small model had the smoothest learning curve, while the large and base models both showed some instability issues.

Some examples of the reconstructed signals compared to the original ones are presented in figures F.2 and F.3. It is noteworthy that the model was able to effectively reconstruct the low-frequency trends, though it encountered difficulties in accurately reconstructing high-frequency spikes. This could be attributed to the lower signal-to-noise ratio (SNR) of high-frequency EEG components, making them more challenging to learn. Given that the passband used in this study ranges from 1 to 64 $Hz$, future work could explore a wider frequency range to capture more information from high-frequency components.

## A.8    BENCHMARK RESULTS

The classification accuracy for each fold per MI dataset for the linear model, EEGNet, EEG Conformer, ST-EEGFormer small (average token fine-tuning), base (average token fine-tuning), large (average token fine-tuning), base-linear probing, and base-class token fine-tuning are listed in tables G.1 to G.5 and G.7 to G.10, respectively.

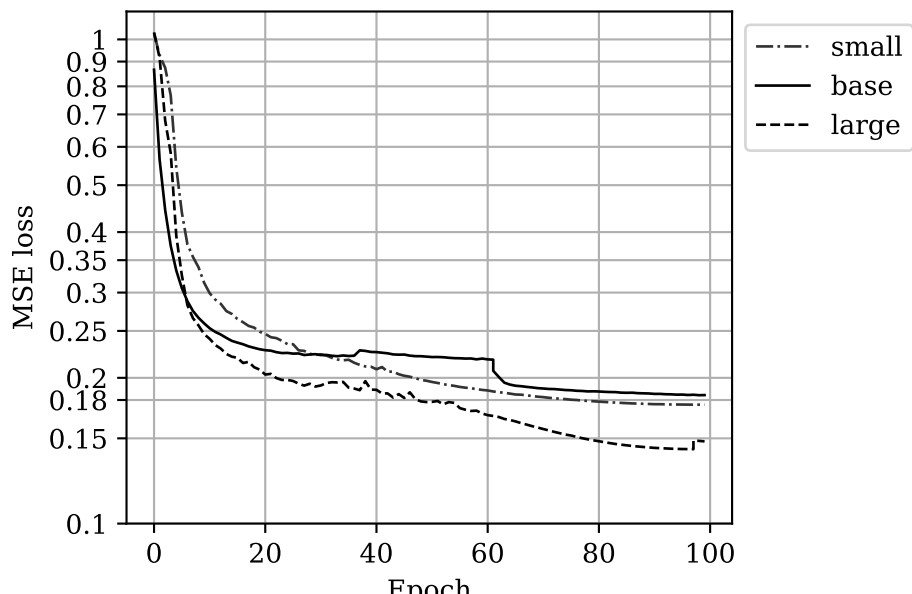

Figure F.1: Learning curves of the small, base, and large ST-EEG MAE models during the MAE pertaining phase.

The benchmark results of the SSVEP dataset are listed in tables G.11 and G.12.

Table G.1: Linear model benchmark results. "-cv" represents the test accuracy on the cross-validation set, and "-test" represents the test accuracy on the hidden test set. Model size is measured by the number of trainable parameters, expressed in millions.

| Fold | | | | | Dataset | | | | |
| | HGD-cv | HGD-test | BCI-Comp-IV2a-cv | BCI-Comp-IV2a-test | BCI-Comp-IV2b-cv | BCI-Comp-IV2b-test | Large-MI-Classic-cv | Large-MI-5F-cv | EEG-MI-BCI-cv |
|---|---|---|---|---|---|---|---|---|---|
| 0 | 0.6267 | 0.6049 | 0.4751 | 0.4255 | 0.6549 | 0.7018 | 0.4523 | 0.3176 | 0.6930 |
| 1 | 0.6089 | 0.5616 | 0.4387 | 0.4387 | 0.5992 | 0.6817 | 0.4433 | 0.3114 | 0.6740 |
| 2 | 0.6367 | 0.5897 | 0.3927 | 0.4437 | 0.6168 | 0.6908 | 0.4463 | 0.3181 | 0.6858 |
| 3 | 0.6566 | 0.6172 | 0.4483 | 0.4167 | 0.6182 | 0.7120 | 0.4399 | 0.3462 | 0.6844 |
| 4 | 0.6266 | 0.5924 | 0.4269 | 0.4286 | 0.6236 | 0.6982 | 0.4280 | 0.3066 | 0.6773 |
| mean | 0.6311 | 0.5931 | 0.4363 | 0.4306 | 0.6225 | 0.6969 | 0.4420 | 0.3200 | 0.6829 |
| std | 0.0174 | 0.0207 | 0.0302 | 0.0107 | 0.0203 | 0.0114 | 0.0090 | 0.0154 | 0.0075 |
| model size | 0.001412 | 0.001412 | 0.000564 | 0.000564 | 0.000218 | 0.000218 | 0.000075 | 0.000653 | 0.000706 |

Table G.2: EEGNet benchmark results. "-cv" represents the test accuracy on the cross-validation set, and "-test" represents the test accuracy on the hidden test set. Model size is measured by the number of trainable parameters, expressed in millions.

| Fold | | | | | Dataset | | | | |
| | HGD-cv | HGD-test | BCI-Comp-IV2a-cv | BCI-Comp-IV2a-test | BCI-Comp-IV2b-cv | BCI-Comp-IV2b-test | Large-MI-Classic-cv | Large-MI-5F-cv | EEG-MI-BCI-cv |
|---|---|---|---|---|---|---|---|---|---|
| 0 | 0.9009 | 0.8603 | 0.7222 | 0.6671 | 0.7541 | 0.8109 | 0.6429 | 0.4800 | 0.7771 |
| 1 | 0.9080 | 0.8580 | 0.6935 | 0.6543 | 0.7391 | 0.8049 | 0.6457 | 0.4732 | 0.7809 |
| 2 | 0.8831 | 0.8603 | 0.6686 | 0.6404 | 0.7649 | 0.8067 | 0.6477 | 0.4872 | 0.7652 |
| 3 | 0.8937 | 0.8536 | 0.6706 | 0.6393 | 0.7500 | 0.8144 | 0.6383 | 0.4722 | 0.7937 |
| 4 | 0.9070 | 0.8625 | 0.6628 | 0.6528 | 0.7378 | 0.8144 | 0.6476 | 0.4804 | 0.7894 |
| mean | 0.8985 | 0.8589 | 0.6835 | 0.6508 | 0.7492 | 0.8103 | 0.6444 | 0.4786 | 0.7813 |
| std | 0.0104 | 0.0034 | 0.0246 | 0.0114 | 0.0112 | 0.0045 | 0.0040 | 0.0061 | 0.0111 |
| model size | 0.010388 | 0.010388 | 0.006996 | 0.006996 | 0.004338 | 0.004338 | 0.004406 | 0.004149 | 0.005778 |

## A.9 PRACTICAL INFORMATION

The models were trained on high-performance computing clusters. Details on GPU usage and training time are provided in table H.1. Calibration and fine-tuning on downstream tasks were significantly faster; for the large model, each iteration took approximately 90 seconds on a P100 GPU.

Table G.3: EEG Conformer benchmark results. "-cv" represents the test accuracy on the cross-validation set, and "-test" represents the test accuracy on the hidden test set. Model size is measured by the number of trainable parameters, expressed in millions.

| Fold | HGD-cv | HGD-test | BCI-Comp-IV2a-cv | BCI-Comp-IV2a-test | Dataset BCI-Comp-IV2b-cv | BCI-Comp-IV2b-test | Large-MI-Classic-cv | Large-MI-5F-cv | EEG-MI-BCI-cv |
|---|---|---|---|---|---|---|---|---|---|
| 0 | 0.9151 | 0.8857 | 0.6226 | 0.5729 | 0.7649 | 0.8222 | 0.7198 | 0.5313 | 0.8398 |
| 1 | 0.9080 | 0.8634 | 0.6054 | 0.5432 | 0.7962 | 0.8335 | 0.7228 | 0.5287 | 0.8165 |
| 2 | 0.9151 | 0.8875 | 0.5920 | 0.5575 | 0.7867 | 0.8359 | 0.7275 | 0.5297 | 0.8088 |
| 3 | 0.9151 | 0.8728 | 0.5828 | 0.5899 | 0.7704 | 0.8335 | 0.7183 | 0.5317 | 0.8232 |
| 4 | 0.9154 | 0.8821 | 0.5887 | 0.5671 | 0.7677 | 0.8303 | 0.7214 | 0.5216 | 0.8189 |
| mean | 0.9137 | 0.8783 | 0.5983 | 0.5661 | 0.7772 | 0.8311 | 0.7220 | 0.5286 | 0.8213 |
| std | 0.0032 | 0.0101 | 0.0159 | 0.0174 | 0.0136 | 0.0053 | 0.0035 | 0.0041 | 0.0117 |
| model size | 1.000132 | 1.000132 | 0.830532 | 0.830532 | 0.800066 | 0.800066 | 0.265798 | 0.265798 | 0.713346 |

Table G.4: BIOT benchmark results. "-cv" represents the test accuracy on the cross-validation set, and "-test" represents the test accuracy on the hidden test set. Model size is measured by the number of trainable parameters, expressed in millions.

| Fold | HGD-cv | HGD-test | BCI-Comp-IV2a-cv | BCI-Comp-IV2a-test | Dataset BCI-Comp-IV2b-cv | BCI-Comp-IV2b-test | Large-MI-Classic-cv | Large-MI-5F-cv | EEG-MI-BCI-cv |
|---|---|---|---|---|---|---|---|---|---|
| 0 | 0.6529 | 0.6382 | NA | NA | NA | NA | 0.4677 | 0.2830 | 0.7374 |
| 1 | 0.6473 | 0.6099 | NA | NA | NA | NA | 0.4413 | 0.2830 | 0.7254 |
| 2 | 0.6529 | 0.6006 | NA | NA | NA | NA | 0.4536 | 0.2841 | 0.7013 |
| 3 | 0.6579 | 0.6006 | NA | NA | NA | NA | 0.4462 | 0.3001 | 0.7344 |
| 4 | 0.6451 | 0.6123 | NA | NA | NA | NA | 0.4683 | 0.2823 | 0.6923 |
| mean | 0.6512 | 0.6123 | NA | NA | NA | NA | 0.4554 | 0.2865 | 0.7182 |
| std | 0.0051 | 0.0154 | NA | NA | NA | NA | 0.0123 | 0.0076 | 0.0200 |
| model size | 3.187716 | 3.187973 | NA | NA | NA | NA | 3.18823 | 3.187973 | 3.187202 |

Table G.5: LaBraM benchmark results. "-cv" represents the test accuracy on the cross-validation set, and "-test" represents the test accuracy on the hidden test set. Model size is measured by the number of trainable parameters, expressed in millions.

| Fold | HGD-cv | HGD-test | BCI-Comp-IV2a-cv | BCI-Comp-IV2a-test | Dataset BCI-Comp-IV2b-cv | BCI-Comp-IV2b-test | Large-MI-Classic-cv | Large-MI-5F-cv | EEG-MI-BCI-cv |
|---|---|---|---|---|---|---|---|---|---|
| 0 | 0.8898 | 0.8844 | 0.4023 | 0.3912 | 0.7419 | 0.8039 | 0.7684 | 0.4551 | 0.7467 |
| 1 | 0.9044 | 0.8821 | 0.3506 | 0.3908 | 0.7160 | 0.7898 | 0.7663 | 0.4588 | 0.7414 |
| 2 | 0.8906 | 0.9738 | 0.3812 | 0.3773 | 0.7378 | 0.7937 | 0.7611 | 0.4911 | 0.7210 |
| 3 | 0.8888 | 0.8884 | 0.3957 | 0.3912 | 0.7459 | 0.8011 | 0.7561 | 0.4324 | 0.7405 |
| 4 | 0.8883 | 0.8799 | 0.3762 | 0.3928 | 0.7296 | 0.8004 | 0.7634 | 0.4815 | 0.7319 |
| Acc | 0.8924 | 0.9017 | 0.3812 | 0.3887 | 0.7342 | 0.7977 | 0.7631 | 0.4638 | 0.7363 |
| Std | 0.0068 | 0.0404 | 0.0201 | 0.0064 | 0.0118 | 0.0058 | 0.0048 | 0.0232 | 0.0101 |
| Model Size | 5.82554 | 5.82554 | 5.82554 | 5.82554 | 5.825138 | 5.825138 | 5.825942 | 5.825741 | 5.825138 |

Table G.6: ST-EEGFormer-small fine-tuning with average token benchmark results. "-cv" represents the test accuracy on the cross-validation set, and "-test" represents the test accuracy on the hidden test set. Model size is measured by the number of trainable parameters, expressed in millions.

| Fold | HGD-cv | HGD-test | BCI-Comp-IV2a-cv | BCI-Comp-IV2a-test | Dataset BCI-Comp-IV2b-cv | BCI-Comp-IV2b-test | Large-MI-Classic-cv | Large-MI-5F-cv | EEG-MI-BCI-cv |
|---|---|---|---|---|---|---|---|---|---|
| 0 | 0.8828 | 0.8714 | 0.5130 | 0.5270 | 0.7326 | 0.7734 | 0.7735 | 0.4957 | 0.9261 |
| 1 | 0.8929 | 0.8643 | 0.5026 | 0.4954 | 0.7378 | 0.7749 | 0.7611 | 0.4913 | 0.8870 |
| 2 | 0.8789 | 0.8513 | 0.4688 | 0.5066 | 0.7865 | 0.7805 | 0.7563 | 0.5038 | 0.8918 |
| 3 | 0.9023 | 0.8442 | 0.5361 | 0.5220 | 0.7743 | 0.7962 | 0.7545 | 0.5128 | 0.9285 |
| 4 | 0.8828 | 0.8576 | 0.4904 | 0.5004 | 0.7309 | 0.7536 | 0.7682 | 0.4965 | 0.8930 |
| mean | 0.8879 | 0.8578 | 0.5022 | 0.5103 | 0.7524 | 0.7757 | 0.7627 | 0.5001 | 0.9053 |
| std | 0.0096 | 0.0107 | 0.0251 | 0.0137 | 0.0260 | 0.0153 | 0.0080 | 0.0084 | 0.0202 |
| model size | 25.4008 | 25.4008 | 25.4008 | 25.4008 | 25.3998 | 25.3998 | 25.4019 | 25.4013 | 25.3998 |

Table G.7: ST-EEGFormer-base fine-tuning with average token benchmark results. "-cv" represents the test accuracy on the cross-validation set, and "-test" represents the test accuracy on the hidden test set. Model size is measured by the number of trainable parameters, expressed in millions.

| Fold | | | | | Dataset | | | | |
| | HGD-cv | HGD-test | BCI-Comp-IV2a-cv | BCI-Comp-IV2a-test | BCI-Comp-IV2b-cv | BCI-Comp-IV2b-test | Large-MI-Classic-cv | Large-MI-5F-cv | EEG-MI-BCI-cv |
|---|---|---|---|---|---|---|---|---|---|
| 0 | 0.8789 | 0.8388 | 0.5000 | 0.4857 | 0.7431 | 0.7649 | 0.7584 | 0.5007 | 0.9369 |
| 1 | 0.8895 | 0.8344 | 0.4792 | 0.4595 | 0.7483 | 0.7425 | 0.7538 | 0.4819 | 0.9387 |
| 2 | 0.8594 | 0.8299 | 0.4609 | 0.4653 | 0.7465 | 0.7557 | 0.7547 | 0.4778 | 0.9309 |
| 3 | 0.8744 | 0.8455 | 0.5072 | 0.4653 | 0.7604 | 0.7539 | 0.7604 | 0.4780 | 0.9435 |
| 4 | 0.8700 | 0.8388 | 0.4519 | 0.4846 | 0.6858 | 0.7408 | 0.7439 | 0.4792 | 0.9339 |
| mean | 0.8744 | 0.8375 | 0.4798 | 0.4721 | 0.7368 | 0.7516 | 0.7542 | 0.4832 | 0.9368 |
| std | 0.0111 | 0.0058 | 0.0239 | 0.0122 | 0.0293 | 0.0100 | 0.0064 | 0.0100 | 0.0048 |
| model size | 85.3271 | 85.3271 | 85.3271 | 85.3271 | 85.3256 | 85.3256 | 85.3256 | 85.3279 | 85.3256 |

Table G.8: ST-EEGFormer-large fine-tuning with average token benchmark results. "-cv" represents the test accuracy on the cross-validation set, and "-test" represents the test accuracy on the hidden test set. Model size is measured by the number of trainable parameters, expressed in millions.

| Fold | | | | | Dataset | | | | |
| | HGD-cv | HGD-test | BCI-Comp-IV2a-cv | BCI-Comp-IV2a-test | BCI-Comp-IV2b-cv | BCI-Comp-IV2b-test | Large-MI-Classic-cv | Large-MI-5F-cv | EEG-MI-BCI-cv |
|---|---|---|---|---|---|---|---|---|---|
| 0 | 0.9559 | 0.9362 | 0.6707 | 0.6408 | 0.7049 | 0.7376 | 0.8333 | 0.6502 | 0.9369 |
| 1 | 0.9487 | 0.9321 | 0.6274 | 0.6420 | 0.7135 | 0.7188 | 0.8333 | 0.6198 | 0.9243 |
| 2 | 0.9515 | 0.9366 | 0.6779 | 0.6478 | 0.6684 | 0.7308 | 0.8305 | 0.6250 | 0.9303 |
| 3 | 0.9548 | 0.9371 | 0.6947 | 0.6397 | 0.7378 | 0.7024 | 0.8270 | 0.6201 | 0.9321 |
| 4 | 0.9581 | 0.9353 | 0.6947 | 0.6381 | 0.6354 | 0.7184 | 0.8325 | 0.6219 | 0.9309 |
| mean | 0.9538 | 0.9354 | 0.6731 | 0.6417 | 0.7216 | 0.7213 | 0.8313 | 0.6274 | 0.9309 |
| std | 0.0038 | 0.0020 | 0.0276 | 0.0037 | 0.0403 | 0.0135 | 0.0027 | 0.0129 | 0.0045 |
| model size | 302.6729 | 302.6729 | 302.6729 | 302.6729 | 302.6709 | 302.6709 | 302.675 | 302.6729 | 302.6709 |

Table G.9: ST-EEGFormer-base linear probing with average token benchmark results. "-cv" represents the test accuracy on the cross-validation set, and "-test" represents the test accuracy on the hidden test set. Model size is measured by the number of trainable parameters, expressed in millions.

| Fold | | | | | Dataset | | | | |
| | HGD-cv | HGD-test | BCI-Comp-IV2a-cv | BCI-Comp-IV2a-test | BCI-Comp-IV2b-cv | BCI-Comp-IV2b-test | Large-MI-Classic-cv | Large-MI-5F-cv | EEG-MI-BCI-cv |
|---|---|---|---|---|---|---|---|---|---|
| 0 | 0.6138 | 0.5871 | 0.4453 | 0.4465 | 0.6927 | 0.7262 | 0.4346 | 0.2987 | 0.6761 |
| 1 | 0.6384 | 0.5795 | 0.4479 | 0.4418 | 0.6944 | 0.7234 | 0.4367 | 0.2802 | 0.6899 |
| 2 | 0.6412 | 0.5826 | 0.4219 | 0.4328 | 0.6997 | 0.7135 | 0.4367 | 0.2958 | 0.7067 |
| 3 | 0.6401 | 0.5540 | 0.4557 | 0.4449 | 0.7118 | 0.7205 | 0.4425 | 0.3003 | 0.6959 |
| 4 | 0.6150 | 0.5906 | 0.4036 | 0.4375 | 0.6806 | 0.7319 | 0.4426 | 0.2969 | 0.6959 |
| mean | 0.6297 | 0.5788 | 0.4349 | 0.4407 | 0.6958 | 0.7231 | 0.4386 | 0.2944 | 0.6929 |
| std | 0.0140 | 0.0145 | 0.0216 | 0.0056 | 0.0113 | 0.0068 | 0.0037 | 0.0081 | 0.0112 |
| model size | 0.0031 | 0.0031 | 0.0031 | 0.0031 | 0.0015 | 0.0015 | 0.0046 | 0.0038 | 0.0015 |

Table G.10: ST-EEGFormer-base fine-tuning with the class token benchmark results. "-cv" represents the test accuracy on the cross-validation set, and "-test" represents the test accuracy on the hidden test set. Model size is measured by the number of trainable parameters, expressed in millions.

| Fold | | | | | Dataset | | | | |
| | HGD-cv | HGD-test | BCI-Comp-IV2a-cv | BCI-Comp-IV2a-test | BCI-Comp-IV2b-cv | BCI-Comp-IV2b-test | Large-MI-Classic-cv | Large-MI-5F-cv | EEG-MI-BCI-cv |
|---|---|---|---|---|---|---|---|---|---|
| 0 | 0.8783 | 0.8183 | 0.4974 | 0.5063 | 0.7604 | 0.7724 | 0.7351 | 0.4663 | 0.9483 |
| 1 | 0.8767 | 0.8156 | 0.4844 | 0.4598 | 0.7378 | 0.7791 | 0.7297 | 0.4587 | 0.9267 |
| 2 | 0.8677 | 0.8210 | 0.4609 | 0.4520 | 0.7535 | 0.7727 | 0.7305 | 0.4611 | 0.9267 |
| 3 | 0.8599 | 0.8054 | 0.5078 | 0.4609 | 0.7448 | 0.7727 | 0.7244 | 0.4601 | 0.9447 |
| 4 | 0.8839 | 0.8241 | 0.4948 | 0.4965 | 0.7569 | 0.7802 | 0.7335 | 0.4622 | 0.9339 |
| mean | 0.8733 | 0.8169 | 0.4891 | 0.4751 | 0.7507 | 0.7754 | 0.7307 | 0.4619 | 0.9361 |
| std | 0.0095 | 0.0072 | 0.0178 | 0.0245 | 0.0092 | 0.0039 | 0.0041 | 0.0028 | 0.0101 |
| model size | 85.3271 | 85.3271 | 85.3271 | 85.3271 | 85.3256 | 85.3256 | 85.3256 | 85.3279 | 85.3256 |

Table G.11: SSVEP dataset benchmark results. Top-1 and top-2 classification accuracies from the leave-one-session-out experiment using 1-s EEG segments are reported. For ST-EEGFormer, the default fine-tuning strategy is end-to-end fine-tuning using the average token. Models denoted by "-cls" indicate end-to-end fine-tuned models utilizing the class token.

| Model | Top-1 accuracy | | | | | | Top-2 accuracy | | | | | |
|---|---|---|---|---|---|---|---|---|---|---|---|---|
| | Fold 0 | Fold 1 | Fold 2 | Fold 3 | Mean | Std | Fold 0 | Fold 1 | Fold 2 | Fold 3 | Mean | Std |
| Linear | 0.0462 | 0.0429 | 0.0472 | 0.0496 | 0.0465 | 0.0028 | 0.0852 | 0.0841 | 0.0912 | 0.0903 | 0.0877 | 0.0036 |
| EEGNet | 0.4313 | 0.4269 | 0.4327 | 0.4392 | 0.4325 | 0.0051 | 0.6215 | 0.6172 | 0.6300 | 0.6318 | 0.6251 | 0.0070 |
| EEG Conformer | 0.3226 | 0.3225 | 0.3220 | 0.3454 | 0.3281 | 0.0115 | 0.5069 | 0.5118 | 0.5114 | 0.5373 | 0.5169 | 0.0138 |
| BIOT | 0.3074 | 0.3194 | 0.3238 | 0.3123 | 0.3157 | 0.0073 | 0.4362 | 0.4512 | 0.4621 | 0.4453 | 0.4487 | 0.0108 |
| LaBraM | 0.5209 | 0.5338 | 0.4853 | 0.5325 | 0.5181 | 0.0226 | 0.66742 | 0.6870 | 0.6392 | 0.6803 | 0.6685 | 0.0211 |
| SSVEP DNN | 0.3845 | 0.3824 | 0.3669 | 0.4075 | 0.3853 | 0.0168 | 0.5719 | 0.5598 | 0.5477 | 0.5993 | 0.5697 | 0.0221 |
| ST-EEGFormer-small | 0.3869 | 0.3855 | 0.3843 | 0.3928 | 0.3874 | 0.0038 | 0.5495 | 0.5462 | 0.5498 | 0.5571 | 0.5507 | 0.0046 |
| ST-EEGFormer-base | 0.2075 | 0.2168 | 0.2225 | 0.2267 | 0.2184 | 0.0083 | 0.3374 | 0.3439 | 0.3442 | 0.3494 | 0.3437 | 0.0049 |
| ST-EEGFormer-large | 0.5721 | 0.5881 | 0.5934 | 0.6046 | 0.5895 | 0.0135 | 0.7309 | 0.7445 | 0.7515 | 0.7639 | 0.7477 | 0.0138 |
| ST-EEGFormer-base-cls | 0.2469 | 0.2509 | 0.2488 | 0.2557 | 0.2506 | 0.0038 | 0.3803 | 0.3820 | 0.3896 | 0.3868 | 0.3847 | 0.0043 |

Table G.12: SSVEP dataset benchmark results. Top-1 and top-2 classification accuracies from the leave-one-session-out experiment using 2-s EEG segments are reported. For ST-EEGFormer, the default fine-tuning strategy is end-to-end fine-tuning using the average token. Models denoted by "-cls" indicate end-to-end fine-tuned models utilizing the class token.

| Model | Top-1 accuracy | | | | | | Top-2 accuracy | | | | | |
|---|---|---|---|---|---|---|---|---|---|---|---|---|
| | Fold 0 | Fold 1 | Fold 2 | Fold 3 | Mean | Std | Fold 0 | Fold 1 | Fold 2 | Fold 3 | Mean | Std |
| Linear | 0.0427 | 0.0484 | 0.0452 | 0.0514 | 0.0469 | 0.0038 | 0.0841 | 0.0877 | 0.0884 | 0.0871 | 0.0868 | 0.0019 |
| EEGNet | 0.6101 | 0.6532 | 0.6444 | 0.6763 | 0.6460 | 0.0274 | 0.7582 | 0.7830 | 0.7905 | 0.8082 | 0.7850 | 0.0208 |
| EEG Conformer | 0.3906 | 0.4218 | 0.4292 | 0.4333 | 0.4187 | 0.0194 | 0.5862 | 0.6231 | 0.6221 | 0.6391 | 0.6176 | 0.0224 |
| BIOT | 0.4813 | 0.5007 | 0.4969 | 0.4892 | 0.4920 | 0.0086 | 0.6124 | 0.6363 | 0.6340 | 0.6244 | 0.6268 | 0.0109 |
| LaBraM | 0.68242 | 0.70823 | 0.71194 | 0.6956 | 0.6995 | 0.0134 | 0.79883 | 0.82653 | 0.8323 | 0.81552 | 0.8183 | 0.0147 |
| SSVEP DNN | 0.4460 | 0.4333 | 0.4199 | 0.4692 | 0.4421 | 0.0210 | 0.6115 | 0.5984 | 0.5709 | 0.6436 | 0.6061 | 0.0302 |
| ST-EEGFormer-small | 0.4409 | 0.4487 | 0.4271 | 0.4470 | 0.4409 | 0.0098 | 0.6011 | 0.6094 | 0.5960 | 0.6111 | 0.6044 | 0.0071 |
| ST-EEGFormer-base | 0.2137 | 0.2233 | 0.2093 | 0.2235 | 0.2174 | 0.0071 | 0.3392 | 0.3465 | 0.3391 | 0.3427 | 0.3419 | 0.0035 |
| ST-EEGFormer-large | 0.7908 | 0.7996 | 0.8176 | 0.8194 | 0.8068 | 0.0140 | 0.8742 | 0.8846 | 0.9040 | 0.9099 | 0.8932 | 0.0166 |
| ST-EEGFormer-base-cls | 0.2624 | 0.2689 | 0.2652 | 0.2714 | 0.2670 | 0.0040 | 0.3977 | 0.4033 | 0.4089 | 0.4065 | 0.4041 | 0.0048 |

Table H.1: Pre-training GPU usage, resource allocation, and training time

| Model | Num.cores | GPU | Num.GPUs | GPU Mem (Mb) | Batch size | training time |
|---|---|---|---|---|---|---|
| small | 16 | A100 40GB | 2 | 35539 | 64 | 9.7 days |
| base | 192 | A100 80GB | 16 | 67650 | 52 | 6.0 days |
| large | 288 | A100 80GB | 24 | 69085 | 32 | 9.8 days |

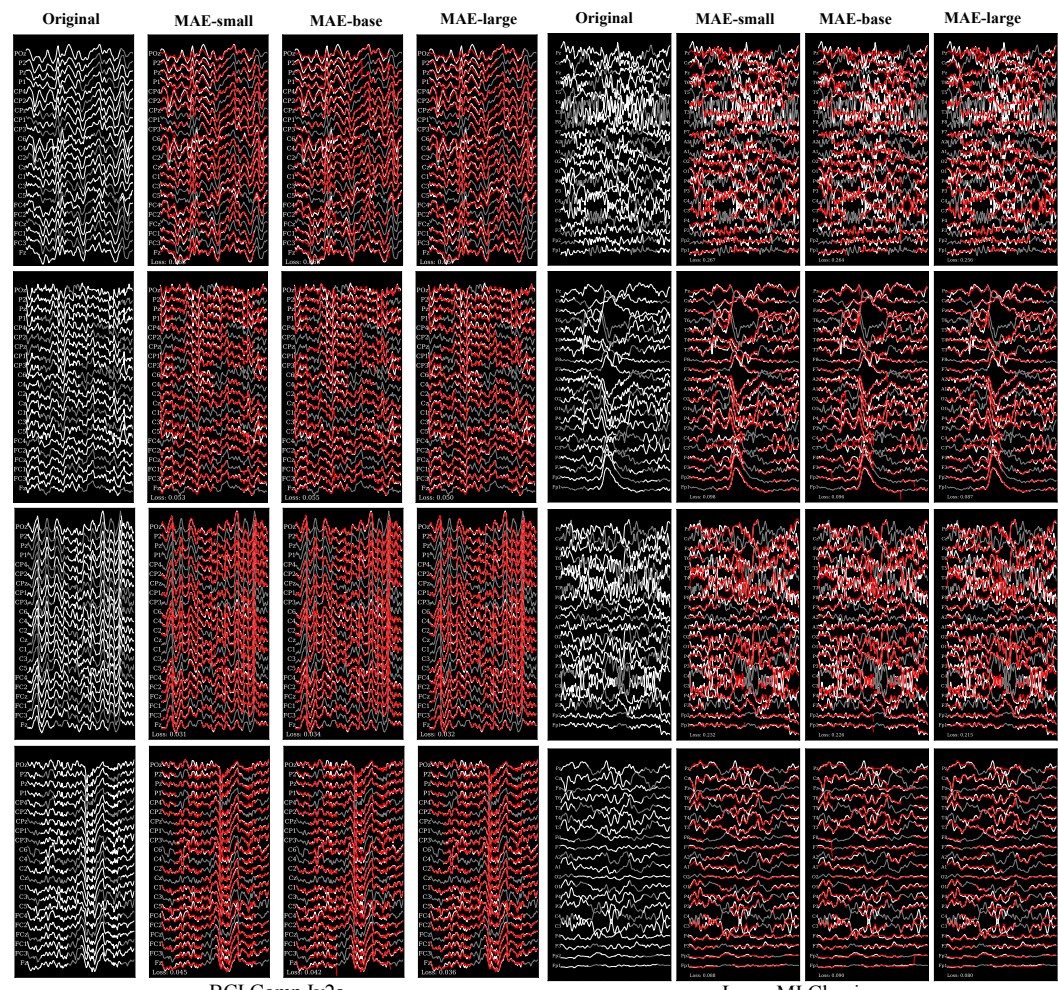

Figure F.2: Random samples from the BCI-Comp-IV2a (Tangermann et al., 2012), Large-MI-Classic (Kaya et al., 2018). For each example (4 rows), the following are shown, from left to right: the original signals with masked segments highlighted in white and unmasked segments in grey; the reconstructed signals in red produced by the MAE-Small model, overlaid on the original signals in white; the reconstructed signals in red produced by the MAE-Base model, overlaid on the original signals in white; and the reconstructed signals in red produced by the MAE-Large model, overlaid on the original signals in white. The corresponding mean squared error (MSE) loss is displayed at the bottom of each figure.

## A.10  MODEL INTERPRETABILITY

### A.10.1  LEARNED EEG CHANNEL EMBEDDINGS

Figures I.1 to I.3 display the learned EEG channel embedding clusters of the ST-EEGFormer small, base, and large respectively. In all three models, a consistent pattern emerged with two clusters. With three clusters, while all models identified clusters in the frontal, left and right temporal, and occipital regions, the small model grouped the left temporal and occipital regions, the base model clustered the frontal and right temporal regions, and the large model grouped the occipital region with the right temporal channels. With four clusters, the small and base models showed a similar pattern, with symmetrical clusters in the frontal, left, right, and occipital regions, whereas the large model tended to group central channels into a separate cluster. When six clusters were used, more fine-grained regions were grouped.

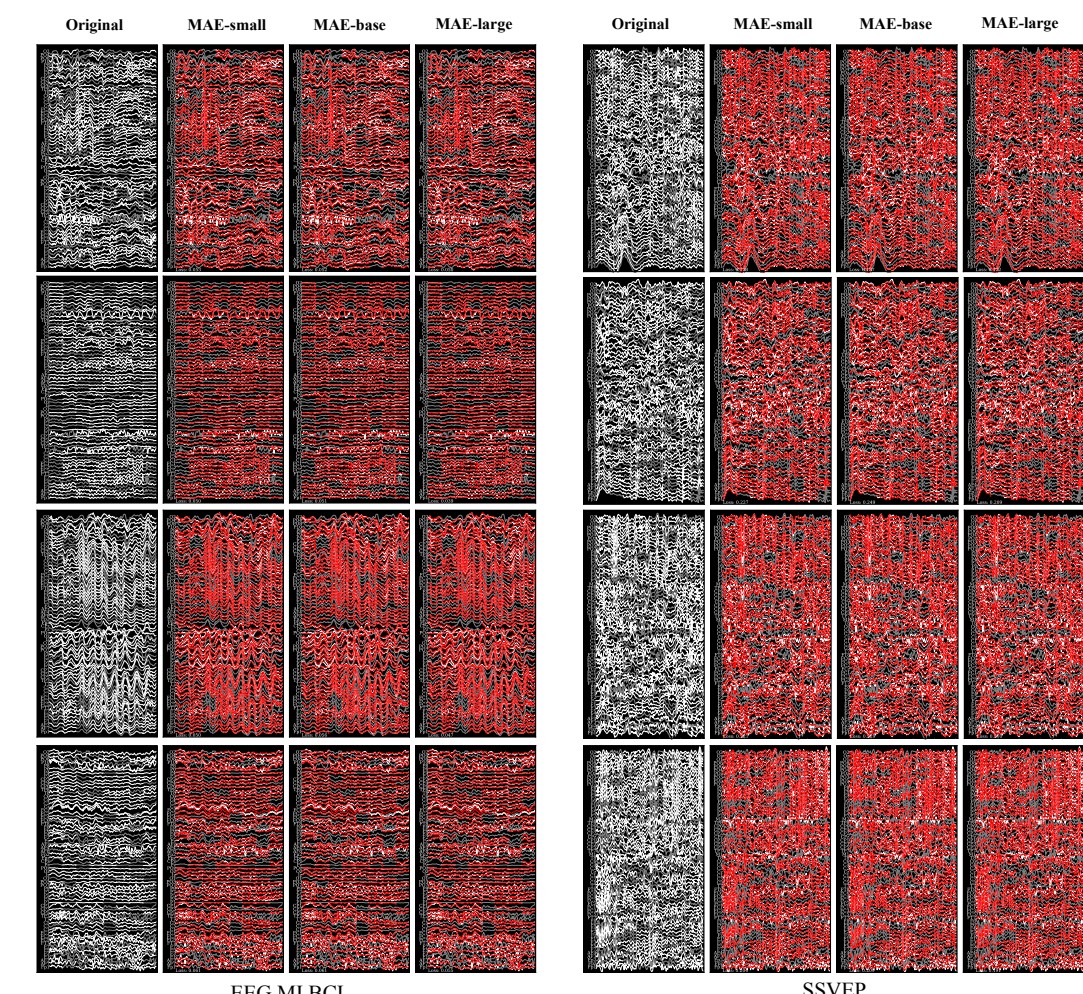

Figure F.3: Random samples from the EEG-MI-BCI (Cho et al., 2017), and SSVEP (Liu et al., 2020) datasets. For each example (4 rows), the following are shown, from left to right: the original signals with masked segments highlighted in white and unmasked segments in grey; the reconstructed signals in red produced by the MAE-Small model, overlaid on the original signals in white; the reconstructed signals in red produced by the MAE-Base model, overlaid on the original signals in white; and the reconstructed signals in red produced by the MAE-Large model, overlaid on the original signals in white. The corresponding mean squared error (MSE) loss is displayed at the bottom of each figure.

### A.10.2 VISUALIZATION OF ATTENTION WEIGHTS OF THE SEIZURE DATASET

Appendix figure I.4 shows the visualization of seizure classification attention weights using attention rollout. Appendix figure I.5 shows the visualization of the top-2 class-specific seizure classification attention weights using gradient attention rollout.

### A.10.3 VISUALIZATION OF ATTENTION WEIGHTS OF THE MI BCI DATASET

Appendix figures I.6 and I.7 compare the learned spatial weights by different benchmark models on the MI-BCI dataset and the Large-MI-5F dataset.

### A.10.4 SPATIOTEMPORAL VISUALIZATION OF ATTENTION WEIGHTS

Figures I.8 and I.9 show a left-hand MI trial and a right-hand MI trial, respectively, with raw EEG highlighted using attention rollout. The corresponding topographic maps at different time points are also included. Both trials are from the EEG-MI-BCI dataset (Cho et al., 2017). Similarly,

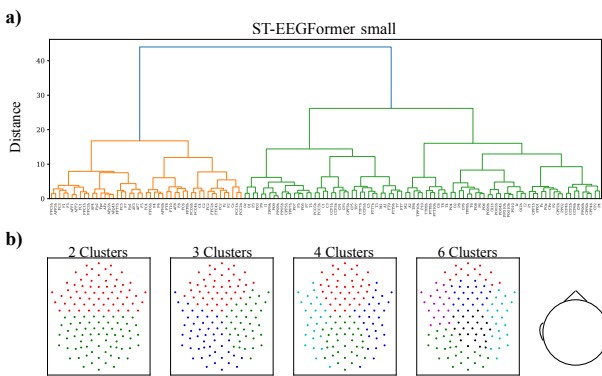

Figure I.1: Learned EEG channel embedding clusters of the pre-trained ST-EEGFormer small model. **a**): Dendrogram showing the cosine similarity between all channel embeddings. **b**): Visualization of 2, 3, 4, and 6 clusters based on the dendrogram results, with channels belonging to the same cluster represented by the same color.

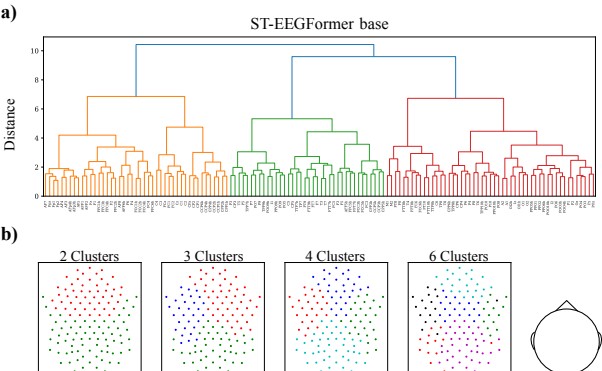

Figure I.2: Learned EEG channel embedding clusters of the pre-trained ST-EEGFormer base model. **a**): Dendrogram showing the cosine similarity between all channel embeddings. **b**): Visualization of 2, 3, 4, and 6 clusters based on the dendrogram results, with channels belonging to the same cluster represented by the same color.

figure I.10 to I.14 show 5-finger MI trials from the thumb, to the pinky finger, respectively, from the Large-MI-5F dateset (Kaya et al., 2018).

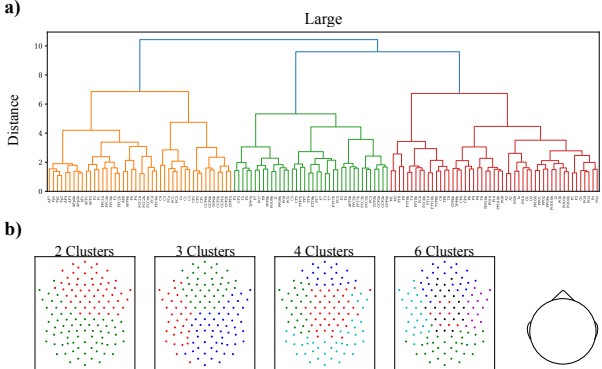

Figure I.3: Learned EEG channel embedding clusters of the pre-trained ST-EEGFormer large model. **a)**: Dendrogram showing the cosine similarity between all channel embeddings. **b)**: Visualization of 2, 3, 4, and 6 clusters based on the dendrogram results, with channels belonging to the same cluster represented by the same color.

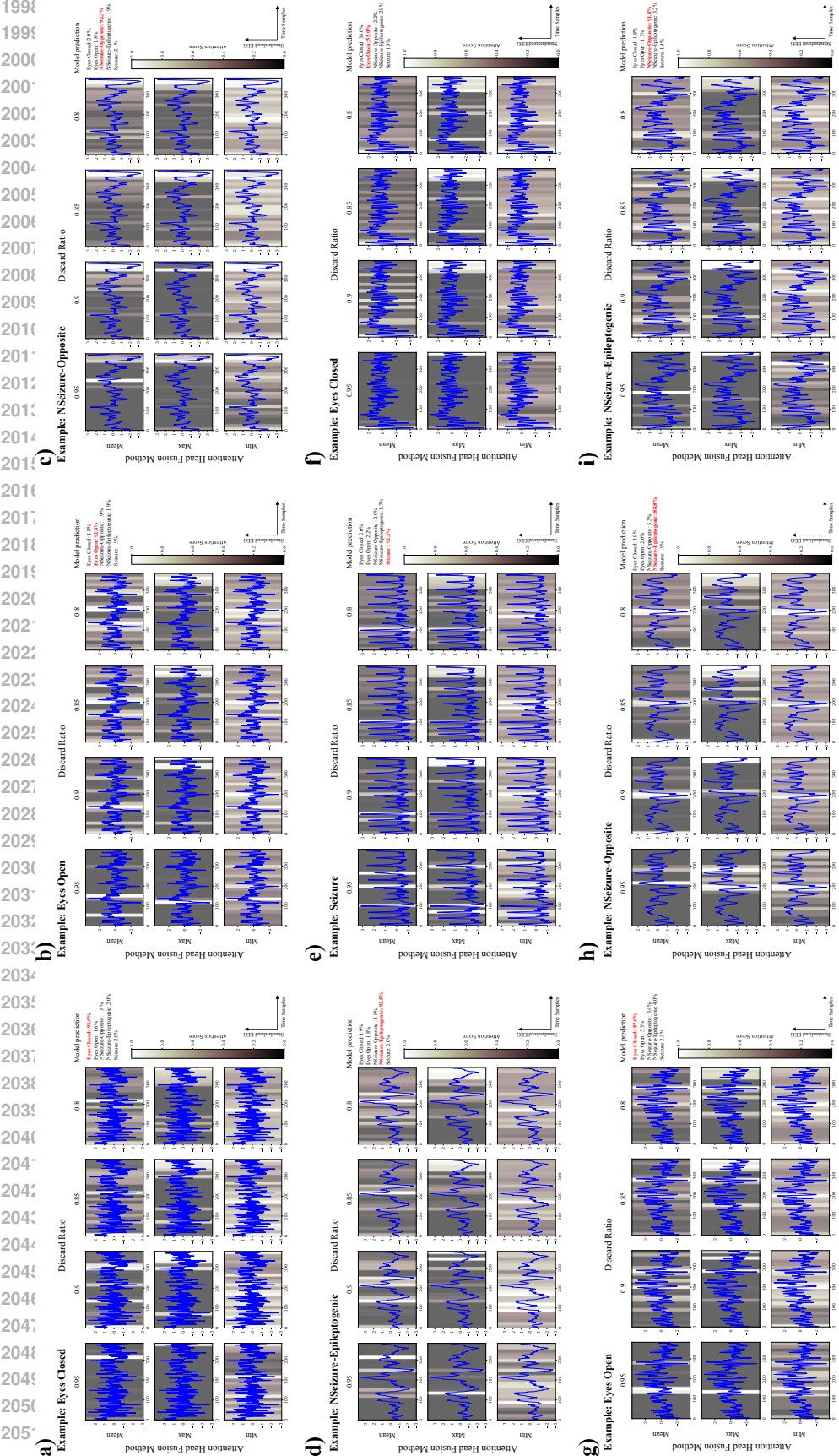

Figure I.4: Seizure classification attention weights visualization using attention rollout. The original signal is shown in blue, with the normalized attention weights visualized using a grayscale mask overlaid on the signal, where larger attention weights are highlighted in white. Different columns represent varying discard ratios, while the three rows correspond to different head fusion methods: mean fusion, max fusion, and min fusion across heads. Additionally, the model's prediction probabilities are displayed at the top right, with the highest probability highlighted in red, representing the final prediction. Figures a) to e) show five trials correctly classified by the model, while figures f) to i) show four trials misclassified by the model.

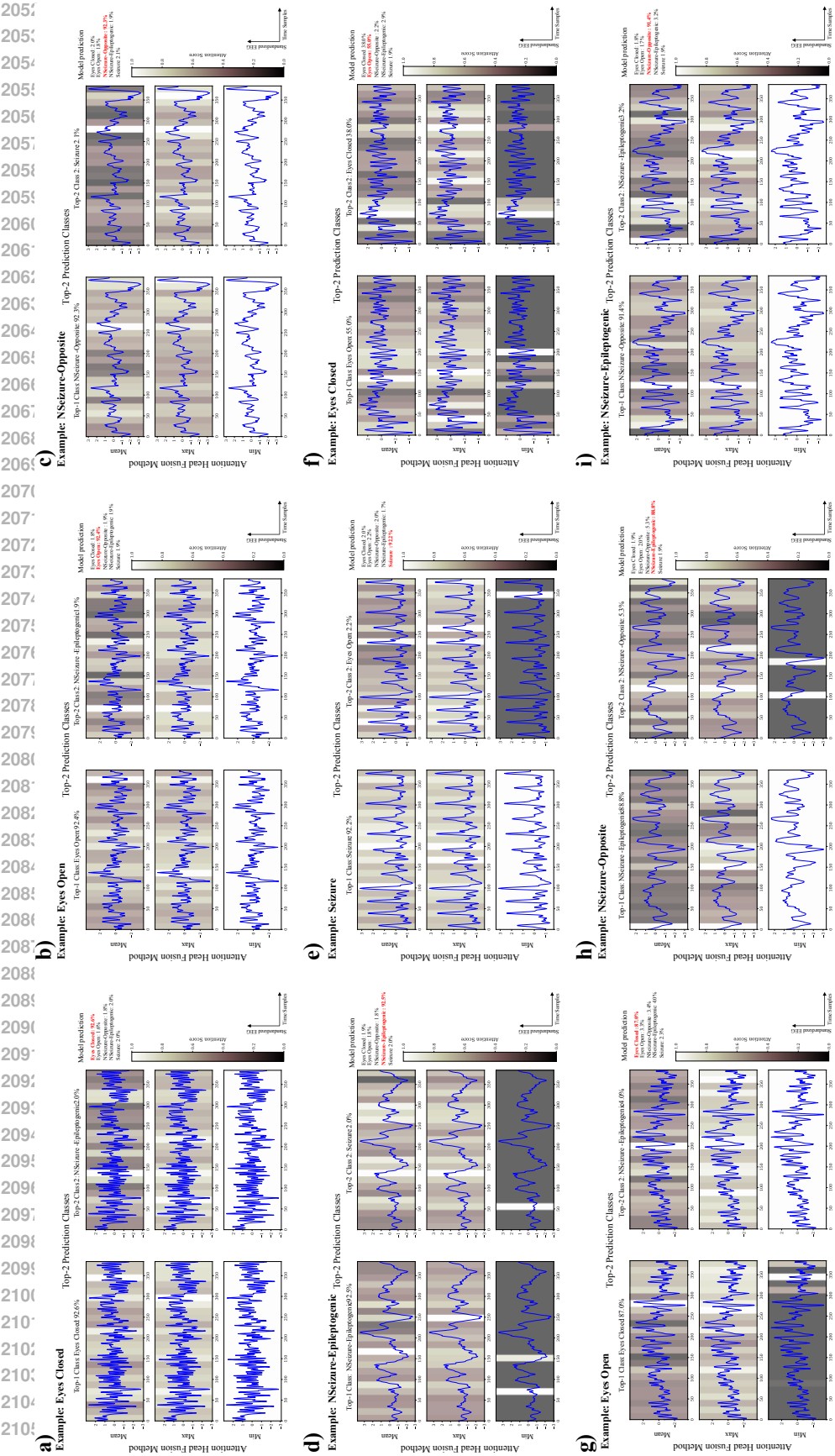

Figure I.5: Seizure classification attention weights visualization using gradient attention rollout. The original signal is shown in blue, with the normalized attention weights visualized using a grayscale mask overlaid on the signal, where larger attention weights are highlighted in white. The two columns represent the visualization for the top-2 model prediction class gradients, while the three rows correspond to different head fusion methods: mean fusion, max fusion, and min fusion. Additionally, the model's prediction probabilities are displayed at the top right, with the highest probability highlighted in red, representing the final prediction. Figures a) to e) show five trials correctly classified by the model, while figures f) to i) show four trials misclassified by the model.

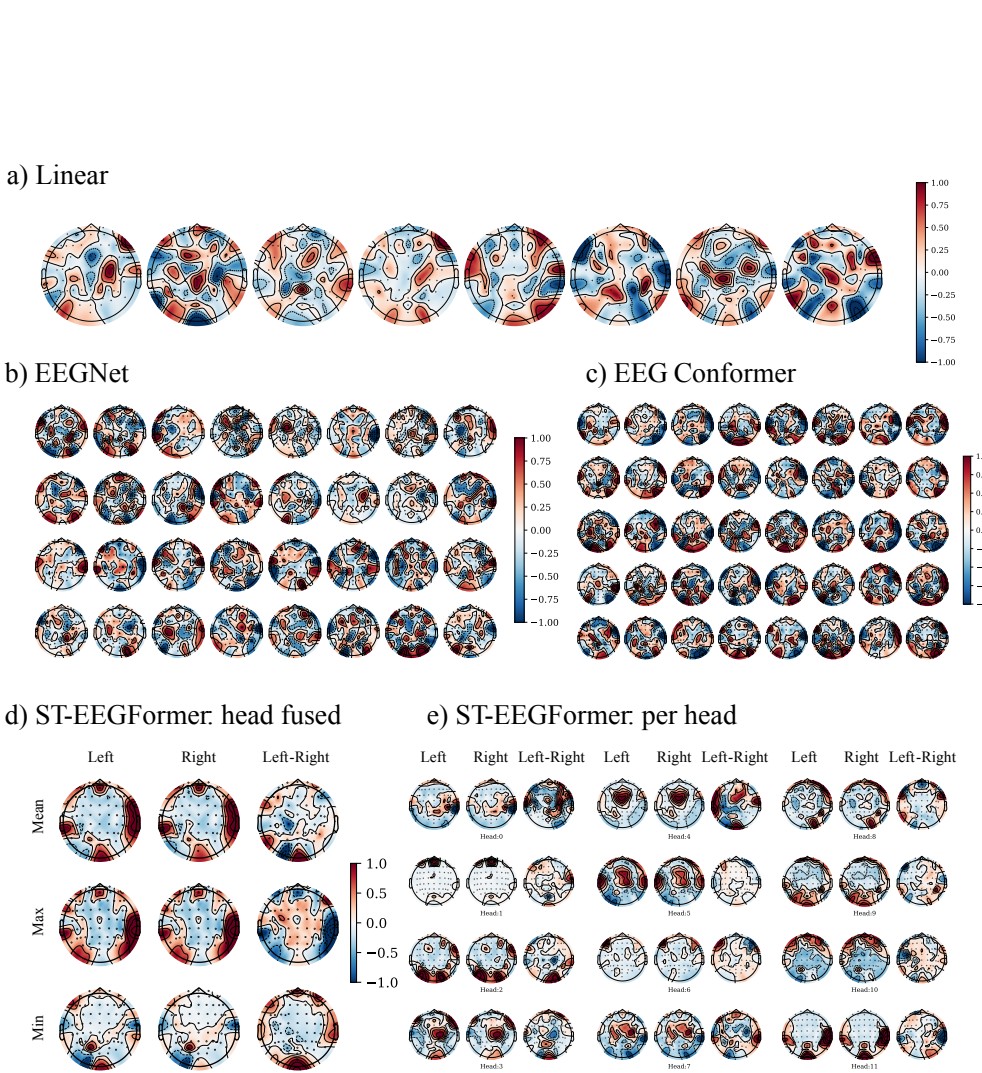

Figure I.6: Visualization of different models using topographic (topo) plots on the MI-BCI dataset (Cho et al., 2017). **a)**: Learned spatial filters of the linear model. **b)**: Learned spatial filters of EEGNet. **c)**: Learned spatial filters of EEG Conformer. **d)**: Averaged attention rollout weights per class, across all trials and sequences, from the ST-EEGFormer model. The attention weights for left-hand motor imagery trials are in the first column, right-hand motor imagery trials are in the middle column, and the difference between the two is in the third column. The three rows represent different head fusion methods: mean fusion (top), max fusion (middle), and min fusion (bottom). The discard ratio for attention rollout is set to 0.9. **e)**: Each triplet visualizes the same attention rollout as in **d)** but without head fusion.

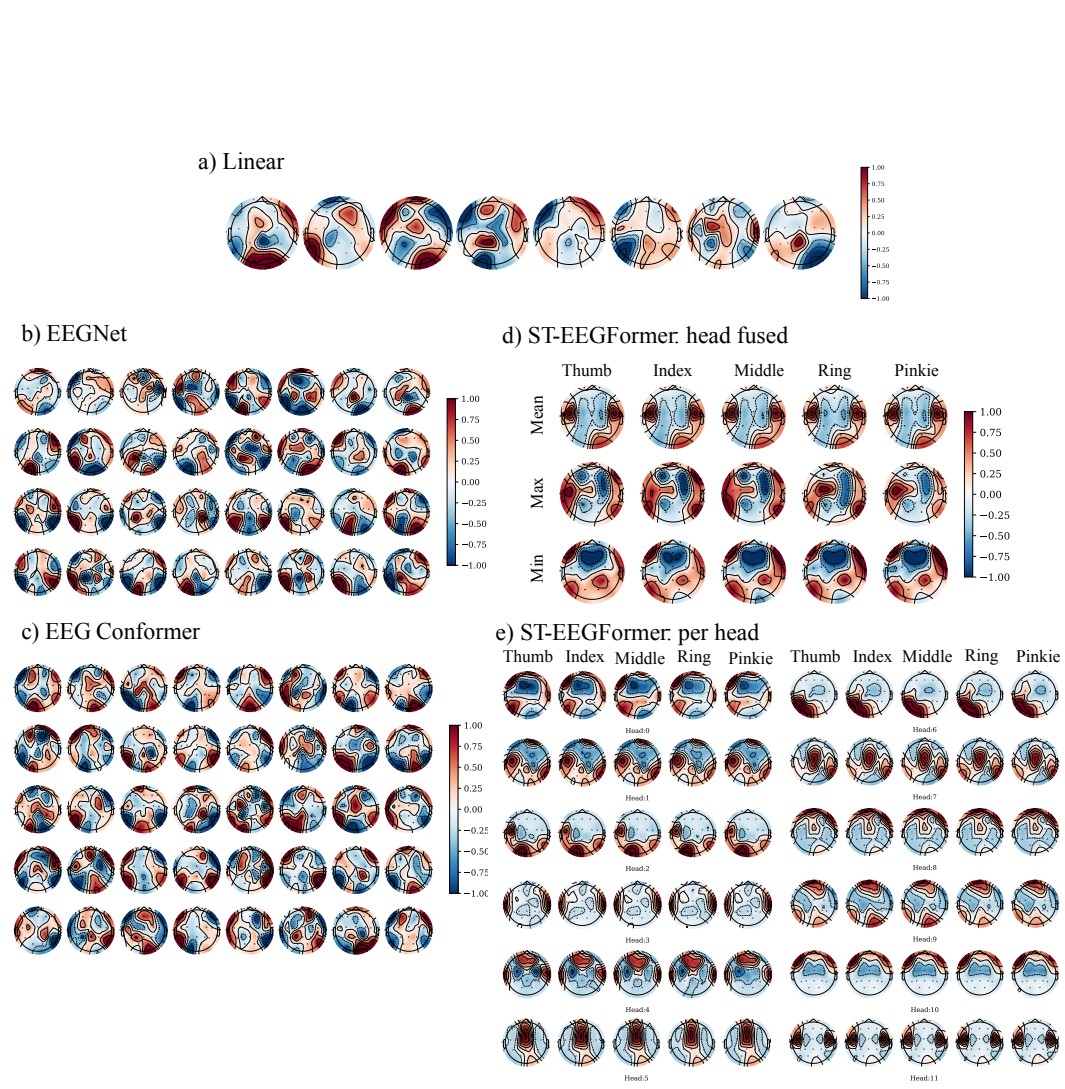

Figure I.7: Visualization of different models using topographic (topo) plots on the Large-MI-5F dataset (Kaya et al., 2018). **a)**: Learned spatial filters of the linear model. **b)**: Learned spatial filters of EEGNet. **c)**: Learned spatial filters of EEG Conformer. **d)**: Averaged attention rollout weights per class, across all trials and sequences, from the ST-EEGFormer model. The attention weights for the five fingers are in a row. The three rows represent different head fusion methods: mean fusion (top), max fusion (middle), and min fusion (bottom). The discard ratio for attention rollout is set to 0.9. **e)**: Each triplet visualizes the same attention rollout as in **d)** but without head fusion.

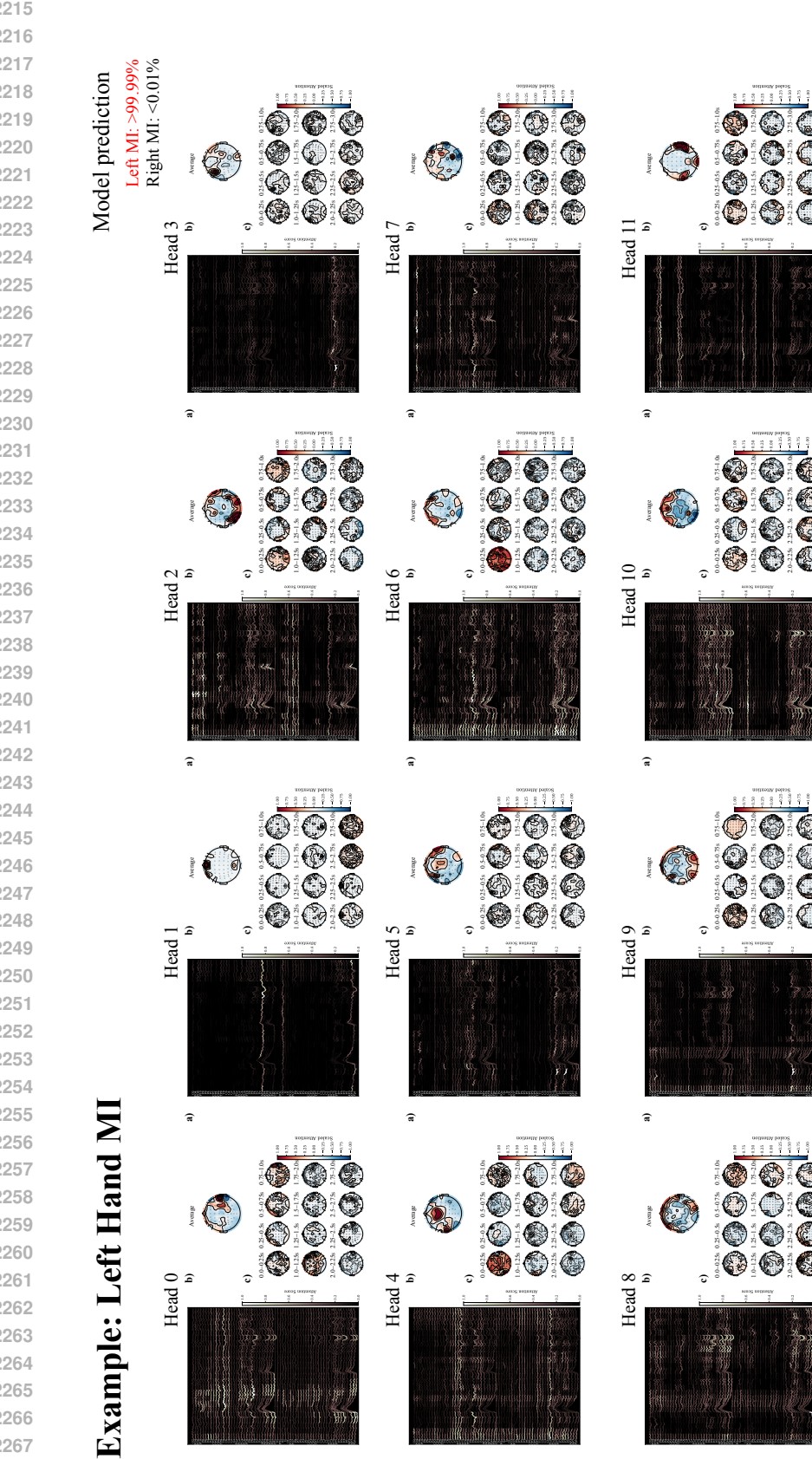

Figure I.8: Visualization of the attention weights across different heads on the raw EEG signals along with the corresponding topographic (topo) plots on a left-hand motor imagery trial from the MI-BCI dataset (Cho et al., 2017). The normalized attention weights are visualized using a grayscale mask overlaid on the signal, where larger attention weights are highlighted in white.

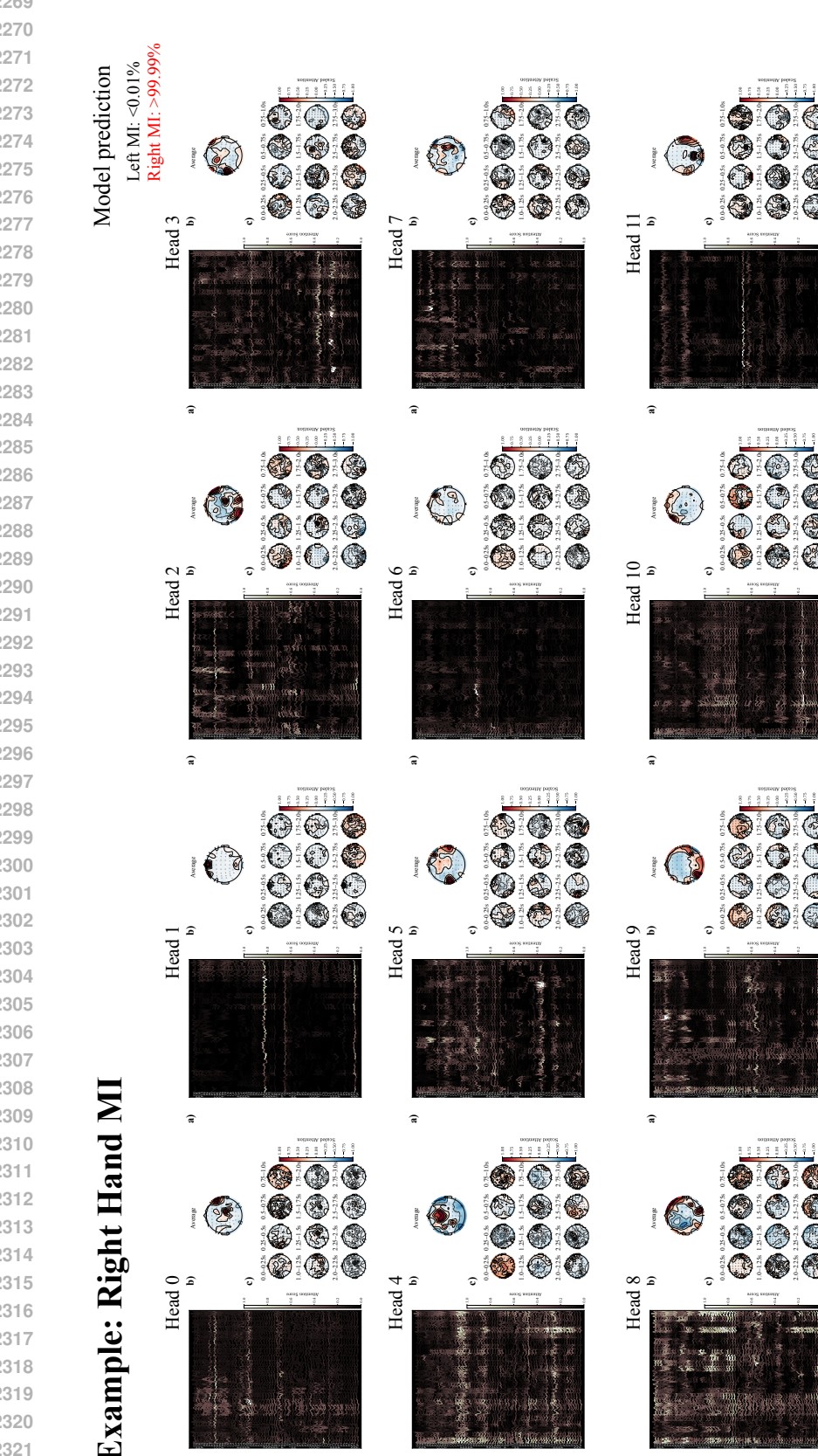

Figure I.9: Visualization of the attention weights across different heads on the raw EEG signals along with the corresponding topographic (topo) plots on a right-hand motor imagery trial from the MI-BCI dataset (Cho et al., 2017). The normalized attention weights are visualized using a grayscale mask overlaid on the signal, where larger attention weights are highlighted in white.

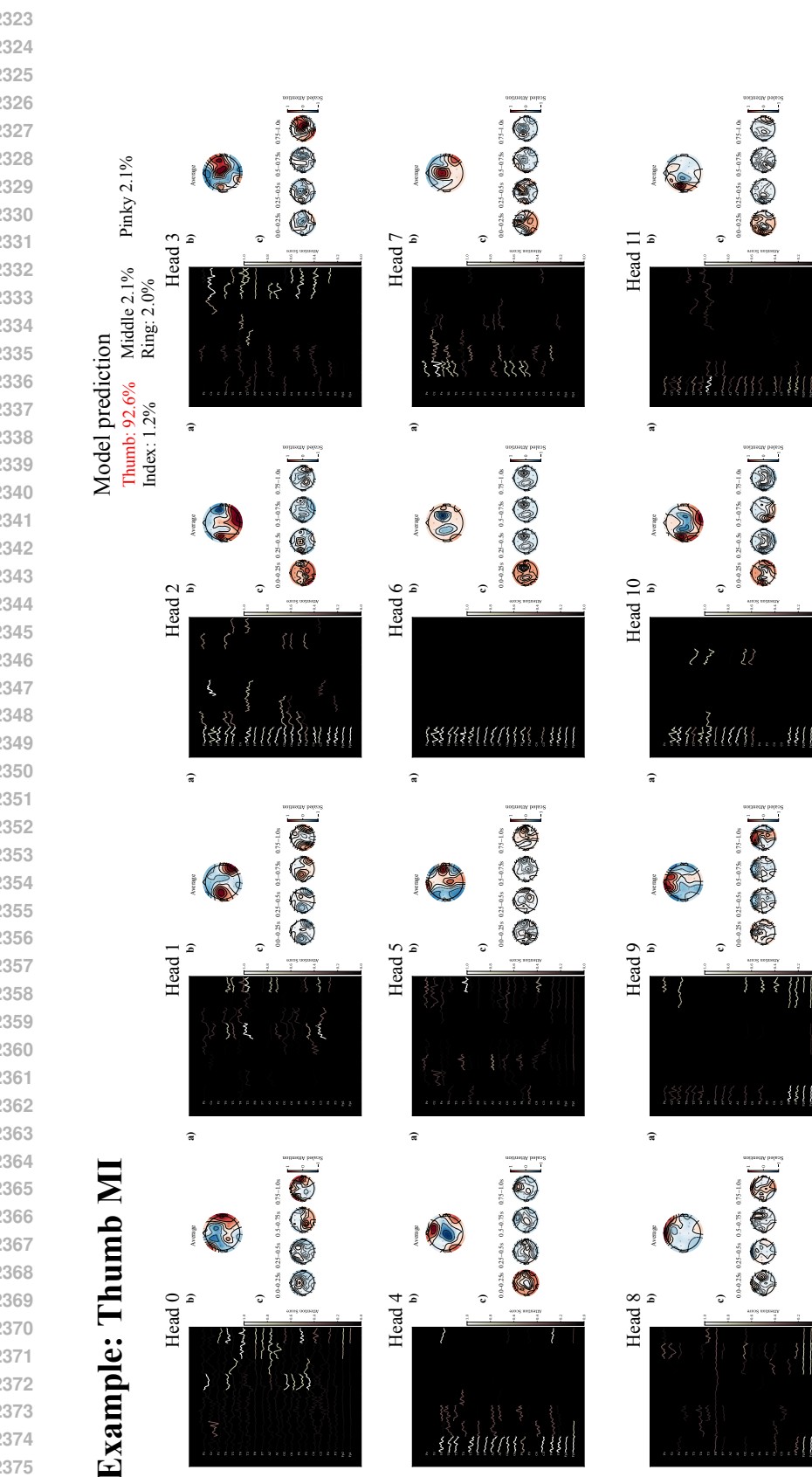

Figure I.10: Visualization of the attention weights across different heads on the raw EEG signals along with the corresponding topographic (topo) plots on a thumb motor imagery trial from the Large-MI-5F dataset (Kaya et al., 2018). The normalized attention weights are visualized using a grayscale mask overlaid on the signal, where larger attention weights are highlighted in white.

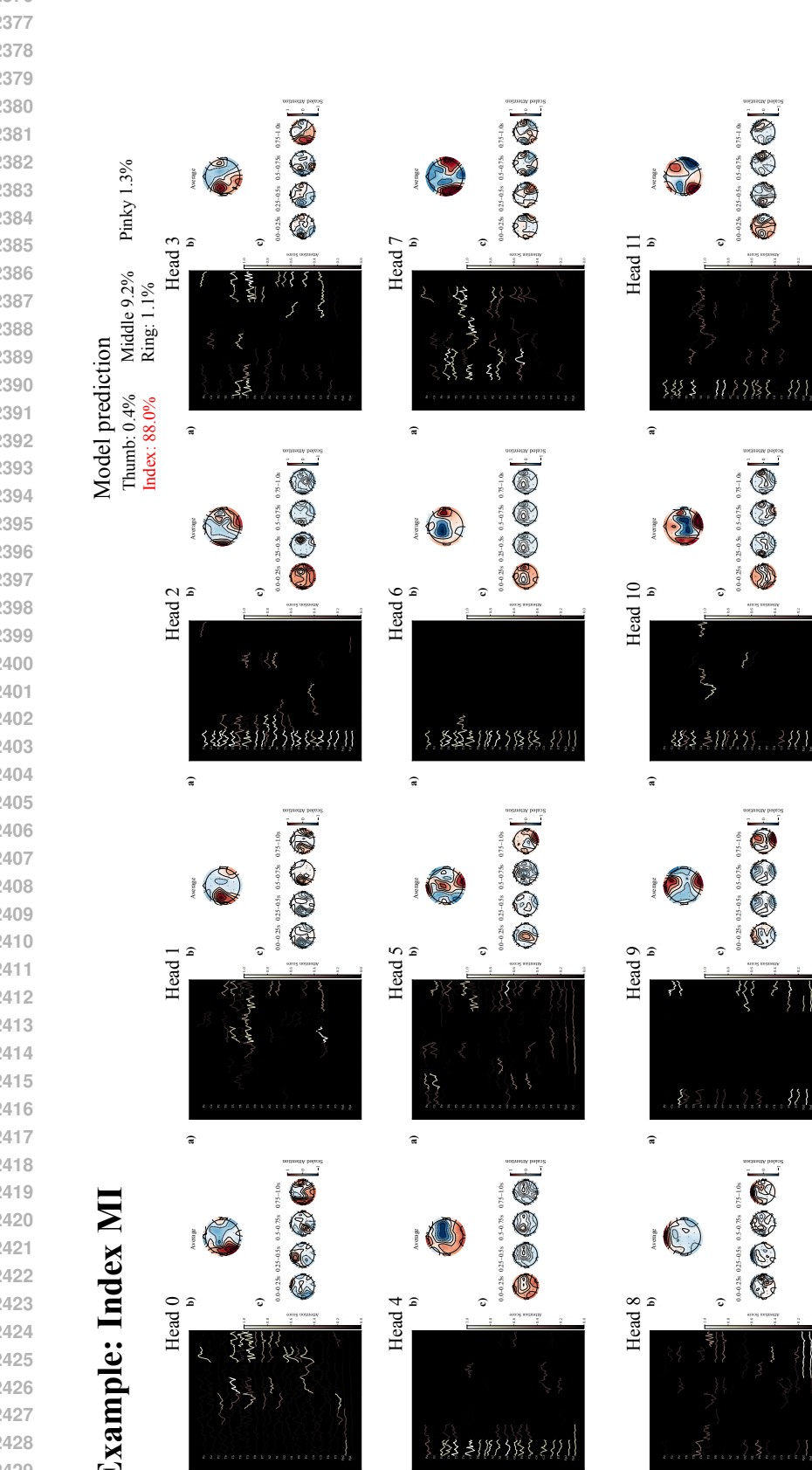

Figure I.11: Visualization of the attention weights across different heads on the raw EEG signals along with the corresponding topographic (topo) plots on an index finger motor imagery trial from the Large-MI-5F dataset (Kaya et al., 2018). The normalized attention weights are visualized using a grayscale mask overlaid on the signal, where larger attention weights are highlighted in white.

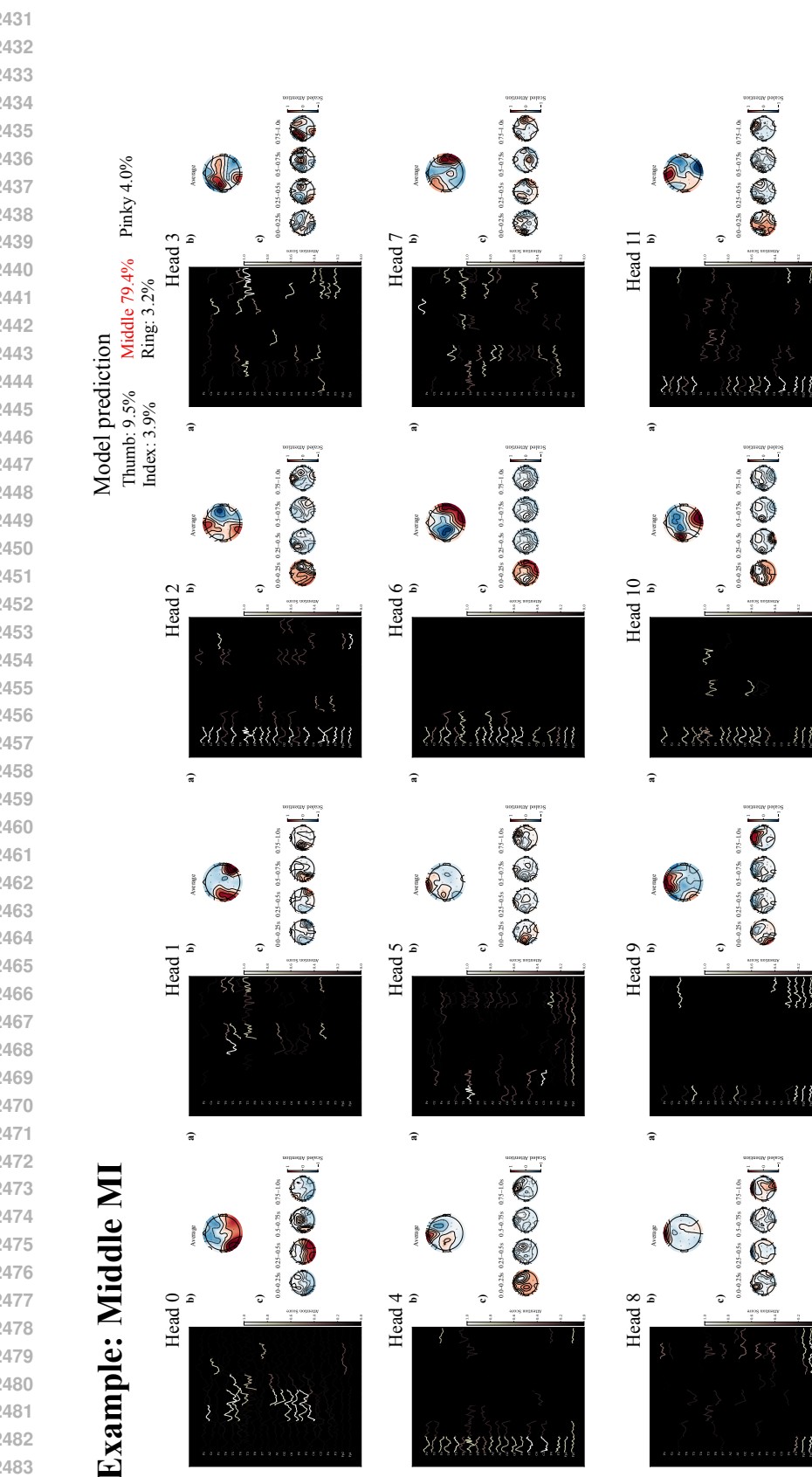

Figure I.12: Visualization of the attention weights across different heads on the raw EEG signals along with the corresponding topographic (topo) plots on a middle finger motor imagery trial from the Large-MI-5F dataset (Kaya et al., 2018). The normalized attention weights are visualized using a grayscale mask overlaid on the signal, where larger attention weights are highlighted in white.

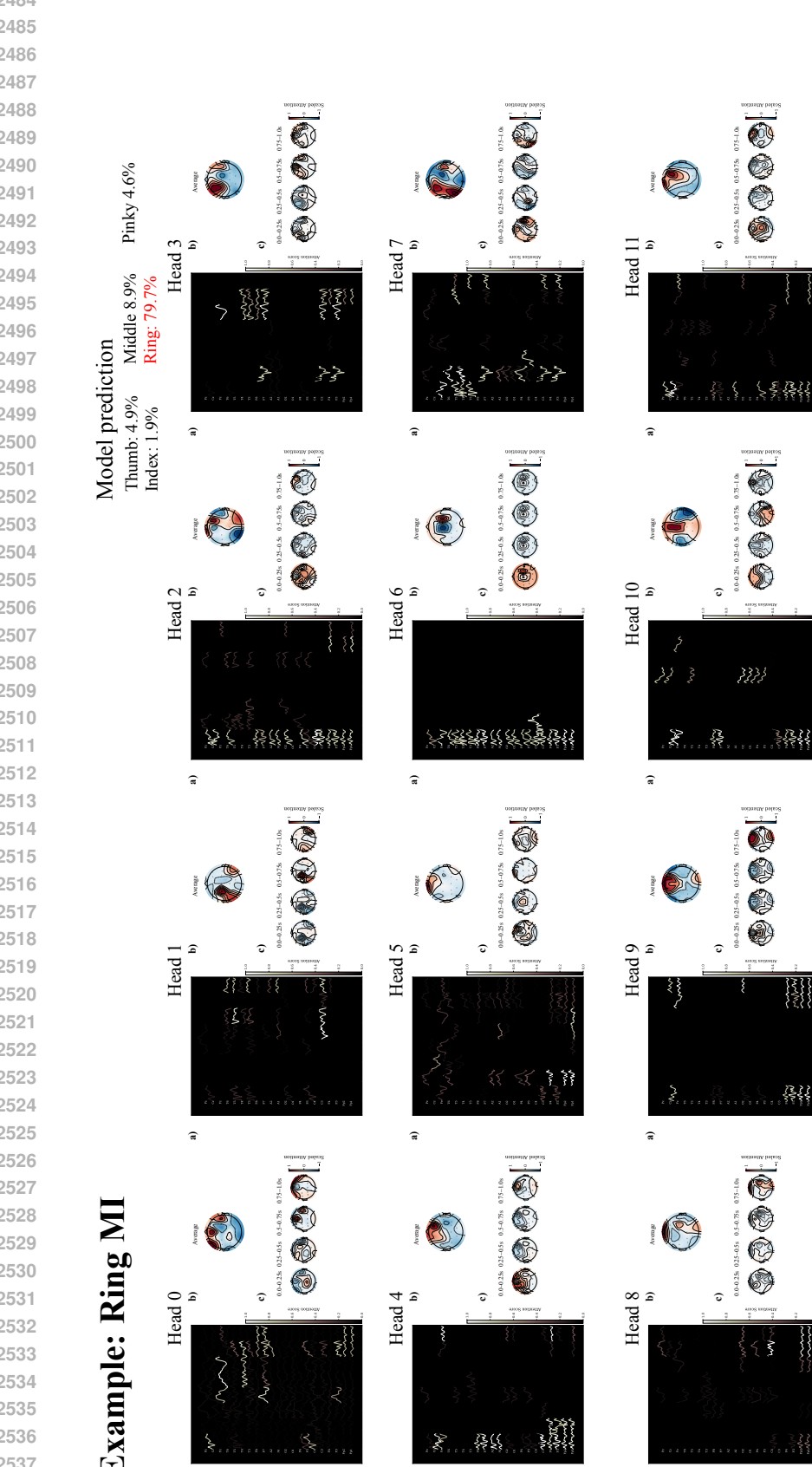

Figure I.13: Visualization of the attention weights across different heads on the raw EEG signals along with the corresponding topographic (topo) plots on a ring finger motor imagery trial from the Large-MI-5F dataset (Kaya et al., 2018). The normalized attention weights are visualized using a grayscale mask overlaid on the signal, where larger attention weights are highlighted in white.

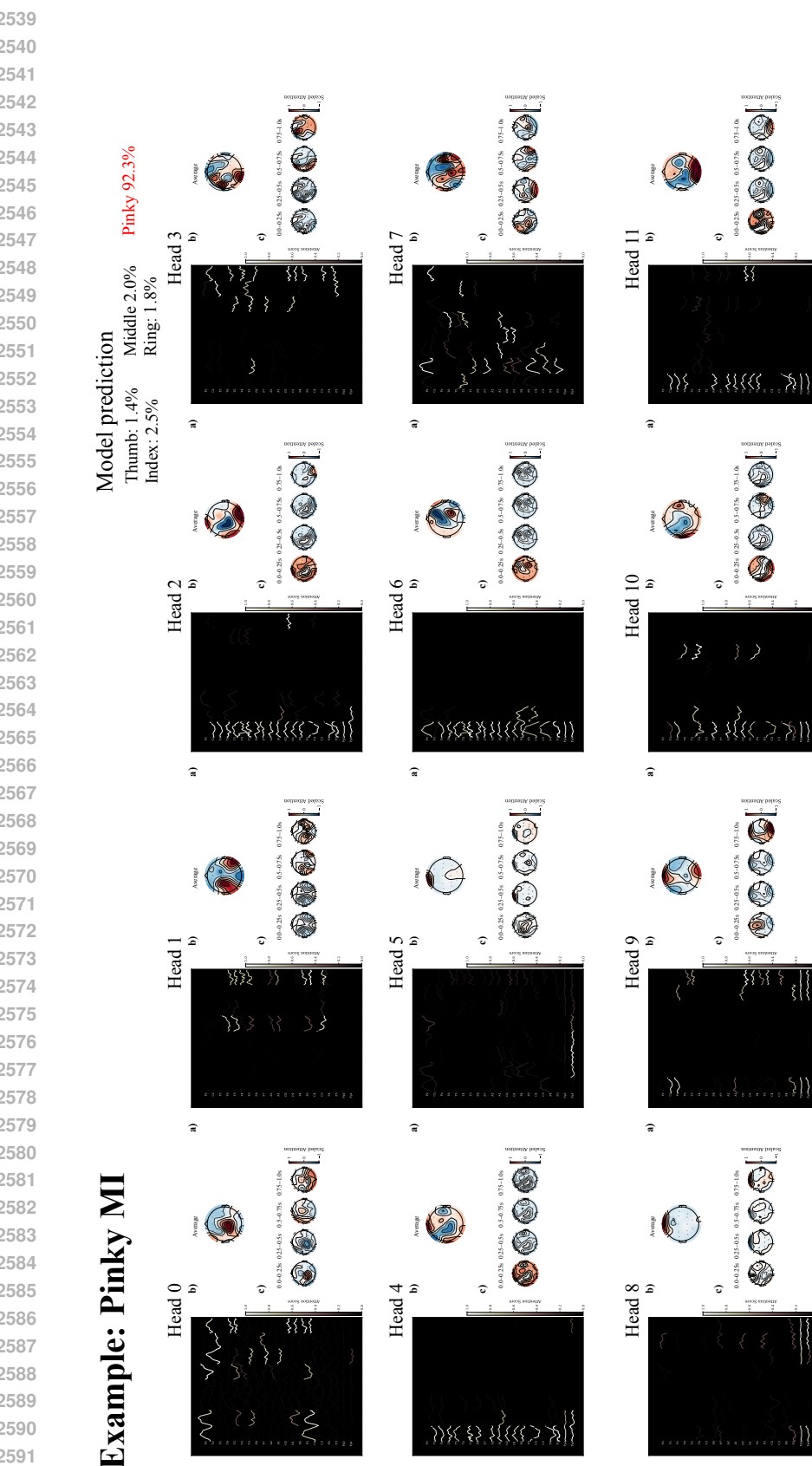

Figure I.14: Visualization of the attention weights across different heads on the raw EEG signals along with the corresponding topographic (topo) plots on a pinky finger motor imagery trial from the Large-MI-5F dataset (Kaya et al., 2018). The normalized attention weights are visualized using a grayscale mask overlaid on the signal, where larger attention weights are highlighted in white.

