# OpenReview forum: "Learning Robust EEG Representations with a Large Spatiotemporal Transformer as a Foundation Model"
_ICLR.cc/2025/Conference — Submitted to ICLR 2025_

### Official Review · Reviewer_ST9f · 2024-10-21

**Soundness:** 3
**Presentation:** 3
**Contribution:** 2
**Rating:** 6
**Confidence:** 5

**Summary:**

This paper proposes a universal spatiotemporal EEG transformer model, which is pretrained on segments of various EEG-based BCI experiment datasets (motor imagery, P300 speller, SSVEP). Pretraining is performed via a standard masked autoencoder (MAE) approach. The model is also shown to generalize well on a diverse set of downstream classification tasks, which were not used in pretraining (e.g., seizure classification).

**Strengths:**

- This work is unique in its collection of a wide range of pretraining datasets on diverse tasks (i.e., both including motor imagery, P300 speller, or SSVEP datasets), without being constrained to the number of EEG sensors.
- The success of generalization experiments on alternative datasets on other tasks (seizure classification, online MI) is a strong point.

**Weaknesses:**

- Methodologically, the work is solely based on the well-known MAE approach, and the main contribution appears to be only empirical.
- Some of the experimental results necessitate further details/clarifications (see questions).

**Questions:**

- It appears like the ST-EEGFormer-large model is often necessary to achieve state-of-the-art performance, in comparison to existing models. To justify and validate this comparison, one should also include the total computational overhead and the training costs of obtaining each of these models (small, base, large) for final use.

- Finger MI classification experiments (Large-MI-5F) demonstrate quite strong results. It would be interesting to see how much of model generalization could be performed on this dataset, if it was not part of the pretraining corpus (i.e., pretraining without Large-MI-5F at all and only testing on this dataset for downstream classification).

- The MAE pretraining strategy should be discussed a bit more in the main manuscript to be self-contained.

- The paper only considers EEGNet and EEG Conformer models as state-of-the-art model examples. However, there has been a wide range of attempts to improve generalization performance in EEG-based BCIs along this line, and the authors should discuss (if not empirically compare against) these relevant pieces of work.

[1] "Learning invariant representations from EEG via adversarial inference."

[2] "Contrastive learning of subject-invariant EEG representations for cross-subject emotion recognition."

[3] "BENDR: Using transformers and a contrastive self-supervised learning task to learn from massive amounts of EEG data."

[4] "SPD domain-specific batch normalization to crack interpretable unsupervised domain adaptation in EEG."

---

> ### Author Response · Authors · 2024-11-21
> **Response to reviewer ST9f**
>
> We thank the reviewer for the thoughtful feedback and suggestions. Below, we address the weaknesses and questions raised in detail:
> Weaknesses:
>
> 1. Methodological Contribution:
>
> While our study leverages the well-known MAE approach, the novelty lies in its application to EEG-based BCIs, demonstrating its effectiveness across diverse BCI paradigms. For the first time, we evaluated a foundation model of this scale on small BCI datasets, which typically have limited training data compared to clinical EEG datasets. Additionally, we conducted visualization and interpretability analyses to better understand model behaviour, an area less explored in prior studies.
>
> 2. Empirical Focus:
>
> We acknowledge that the main contributions are empirical. With our study, we aim to provide a robust model that can serve as a foundation for future research and BCI applications. The promising results obtained from all benchmark datasets validate our claims and contribute to advancing neuroscience applications.
>
> Questions:
>
> 1. Computational Overhead and Training Costs:
>
> We thank the reviewer for raising this important point. We have added practical details regarding computational overhead and training costs for each model size (small, base, and large) in Appendix Table~A.1. While pretraining from scratch is computationally expensive, we emphasize that once the model is pre-trained, it can be transferred to new datasets relatively quickly and efficiently.
>
> 2. Generalization on Finger MI Dataset (Large-MI-5F):
>
> We address this concern from two perspectives:
>
> o Firstly, our experiments demonstrate that a pre-trained model generalizes well to new datasets and that additional pretraining on unseen datasets can further enhance performance. Consequently, excluding Large-MI-5F from the pretraining corpus would reasonably lead to a performance drop on this dataset, as supported by similar findings in LaBraM (Jiang et al., 2024). For instance, excluding the TUAB dataset during pretraining led to a slight performance drop, while performance on TUEV improved.
>
> o Secondly, a deeper analysis of the Finger MI dataset, particularly the representations learned by the large model, would indeed be an interesting direction for future research. Such a study could provide valuable insights and is more suitable for a follow-up investigation.
>
> 3. Discussion on MAE Pretraining Strategy:
>
> We have added a brief introduction to the MAE pretraining strategy in the “Proposed Approach” section, where it is first mentioned. For further details, we have directed readers to the comprehensive explanation provided in the Appendix.
>
> 4. Comparison with Broader State-of-the-Art Models:
>
> We thank the reviewer for highlighting additional relevant works. In the revised manuscript, we have expanded the Related Works section to discuss these studies, emphasizing their contributions and differences from our approach:
>
> o "Learning invariant representations from EEG via adversarial inference (Özdenizci et al., 2020)" and "Contrastive learning of subject-invariant EEG representations for cross-subject emotion recognition (Shen et al., 2021)" focus on learning subject-invariant representations, which differ from our reconstruction-based strategy.
>
> o "BENDR: Using transformers and a contrastive self-supervised learning task to learn from massive amounts of EEG data (Kostas et al., 2021)" applies contrastive learning on limited datasets, whereas our work addresses a broader range of BCI tasks.
>
> o "SPD domain-specific batch normalization to crack interpretable unsupervised domain adaptation in EEG" explores domain-specific normalization, which complements but does not directly align with our approach.
>
> While empirical comparisons with all these methods are beyond the scope of this study, we have included a comparison with other recent EEG foundation models, BIOT and LaBraM, in the results and discussion sections. Additionally, we provide an overview of similarities and differences between relevant models in Appendix Table~A.1.

---

> > ### Comment · Reviewer_ST9f · 2024-11-25
> > **Response to Authors' Rebuttal**
> >
> > Thanks to the authors for their efforts during the rebuttal.
> >
> > I appreciate the clarifications on the experimental setup, how MAE pretraining was implemented, future work discussions, and additions to the related works. I agree that this submission contains a solid empirical novelty with its application to EEG-based BCIs. However, I tend to keep my initial score "marginally above the acceptance threshold", since this work lacks novelty from the methodological perspective.

---

> > > ### Author Response · Authors · 2024-11-25
> > > **Thank you for your comments**
> > >
> > > Dear Reviewer,
> > > We sincerely appreciate the time and effort you have invested in reading and evaluating our work.
> > > Regarding your concern about the novelty of our methodology, we would like to briefly address the following points:
> > >
> > > 1.	Our work focuses on BCI applications using large pretrained models within the scope of "applications to neuroscience and cognitive science" in this venue. Unlike related models such as BIOT (Yang et al., 2023) and LaBraM (Jiang et al., 2024)—which primarily target clinically oriented scenarios such as seizure detection and sleep stage classification—our research emphasizes the limitations of these models when applied to BCI tasks and explores their potential specifically in this domain.
> > >
> > > 2.	Compared to other models with similar underlying principles, we believe our proposed model distinguishes itself through its simplicity in both architecture and training process. By employing a linear tokenizer and a single-step MAE pretraining task, our approach ensures greater accessibility and ease of use, see Table A.1 for detailed comparisons.
> > >
> > > 3.	We have explored multiple strategies to tackle a key challenge in BCI research: the limited availability of labelled data (Khademi et al., 2022). Using our pretrained model, we demonstrate that direct fine-tuning achieves superior performance in data-limited scenarios. Additionally, model calibration using task-irrelevant resting-state signals further enhances performance, highlighting the flexibility and robustness of our approach.
> > >
> > > In summary, we hope our research offers a meaningful alternative in the BCI field and provides a foundation for future advancements. We remain open to further suggestions and would be delighted to hear any additional insights you may have.
> > > Thank you once again for your constructive comments and support.
> > >
> > > References:
> > > Khademi, Z., Ebrahimi, F., & Kordy, H. M. (2022). A review of critical challenges in MI-BCI: From conventional to deep learning methods. Journal of Neuroscience Methods, 383, 109736. https://doi.org/10.1016/j.jneumeth.2022.109736

---

### Official Review · Reviewer_Z5h9 · 2024-10-29

**Soundness:** 3
**Presentation:** 3
**Contribution:** 2
**Rating:** 6
**Confidence:** 5

**Summary:**

This paper addresses the challenges of developing a universal EEG-based brain-computer interface (BCI) by introducing a novel spatiotemporal EEG transformer (ST-EEGFormer). This work combines the objectives of enhancing existing applications in BCI technology and addressing the problem of limited labeled data in EEG research. By proposing a model that effectively handles diverse EEG recording setups and tasks through self-supervised learning, the authors aim to improve data efficiency and classification performance. Additionally, the study emphasizes the generalization capabilities of the ST-EEGFormer across various datasets, thus highlighting its potential impact on the field of EEG research and BCI applications. Overall, the paper seeks to provide both a practical solution for known challenges and a theoretical contribution to the understanding of EEG data representation.

**Strengths:**

A strong point of this work is the demonstrated capability of the pretrained ST-EEGFormer to learn robust EEG representations that achieve higher classification accuracies compared to benchmark models across various pretraining datasets. This indicates not only the effectiveness of the model but also its adaptability, as it exhibits strong generalization on new datasets, even when limited training data is available. Additionally, the inclusion of visualizations of model features enhances the interpretability of the results, providing insights into how the model operates. The commitment to making the pretrained model weights and code publicly available further promotes transparency and encourages further research in the field, enhancing the overall impact of the study.

**Weaknesses:**

One potential weakness of the study is the reliance on self-supervised learning, which may not fully address the challenges posed by the variability in EEG data across different tasks and recording setups. While the ST-EEGFormer shows strong performance, the model's effectiveness could be limited when faced with completely novel or atypical datasets that diverge significantly from those used in pretraining. Additionally, the complexity of the model may result in increased computational requirements, which could pose practical challenges for real-time BCI applications. Finally, while the paper reports improved classification accuracies, it may benefit from a more detailed analysis of the model's interpretability and potential biases in its learning process, ensuring that the results are not only accurate but also reliable across diverse populations and scenarios.

**Questions:**

1- The introduction provides a comprehensive overview of EEG and its applications in BCIs, effectively contextualizing the research. However, there are a few areas that could be improved: First, while it outlines various EEG paradigms and traditional methods, it could benefit from a clearer statement of the specific gap or problem the proposed ST-EEGFormer addresses more explicitly. This would help readers understand the motivation behind the research. Furthermore, the transition between discussing existing methods and introducing the new model feels somewhat abrupt; a smoother connection would enhance the flow of the narrative.

Ican see huge gap here: Consequently, small models that typically rely on convolutional neural networks (CNNs) impose
restrictions on input shape (e.g., the number of EEG channels, and the number of samples), further
complicating the use of different datasets as they usually exhibit different data formats.
Contributions: This paper presents, for the first time, a large transformer-based ST-EEGFormer
model that is channel-aware, flexible, and applicable across various datasets and tasks.

2- In the related work section, I suggest enhancing the discussion on recent advances in attention-based transformer models that have inspired BCI researchers. Specifically, you could include citations from foundational papers in this area, such as:

"A transformer-based approach combining deep learning networks and spatial-temporal information for raw EEG classification" (IEEE Transactions on Neural Systems and Rehabilitation Engineering, 2022, 30, pp. 2126-2136).
"An end-to-end CNN with attentional mechanism applied to raw EEG in a BCI classification task" (Journal of Neural Engineering, 2021, 18(4), p. 0460e3).
Including these earlier works would provide valuable context, allowing you to discuss how they relate to or differ from the current studies. Additionally, a brief exploration of the evolution of transformer-based approaches in EEG classification would enrich the narrative and demonstrate the progression in this research area. This could help clarify the significance of the EEG Conformer model in relation to previous methodologies.

3-It may be beneficial to consider both non-overlapping and overlapping segments during the tokenization process. Non-overlapping segments simplify the model's input structure, potentially enhancing processing efficiency. However, overlapping segments can capture greater temporal context and variations within the data, which may lead to richer feature representations. By incorporating both approaches, the model could leverage the strengths of each method, thereby improving its ability to generalize across different EEG tasks and enhancing overall performance. This dual strategy would also provide more flexibility in handling EEG data, accommodating various recording setups and experimental conditions. I recommend discussing the potential trade-offs between these approaches and comparing their performance specifically in the context of the ST-EEGFormer model.

4- I encourage the authors to include a discussion section in this paper to address the results in depth. This section should explore the implications of the findings, highlight any limitations, and suggest directions for future research. Including these elements will enhance the overall impact of the paper and provide valuable insights for readers.

---

> ### Author Response · Authors · 2024-11-21
> **Response to reviewer Z5h9**
>
> We sincerely thank the reviewer for the comments and suggestions. Below, we address the key points raised:
>
> 1. Regarding the weaknesses of effectiveness on new or atypical datasets, implications for real-time BCI applications, and interpretability and biases. We refer to our responses to Reviewer RZ9b, where these points were addressed in detail. Briefly, we have conducted experiments on unseen datasets, discussed practical implications for real-time applications, and added detailed visualization analyses to explore interpretability and potential biases. These updates are also discussed in the newly added discussion section.
>
> 2. Introduction Improvements:
>
> We thank the reviewer for pointing out the gap in the Introduction regarding the motivation for ST-EEGFormer. In the revised manuscript, we have revised the Abstract and Introduction to clearly state the challenges addressed by our work, such as:
>
> a. Limited studies evaluating the performance of large foundation models on diverse BCI datasets.
>
> b. Difficulties faced by small models in transferring between tasks and datasets.
>
> c. The limited availability of data for many BCI tasks.
>
> 3. Related Work Expansion:
>
> We have expanded the Related Works section to include recent developments in attention-based transformer models that have influenced BCI research. Specifically, we have incorporated discussions and citations from foundational works such as PatchTST (Nie et al., 2023), iTransformer (Liu et al., 2024), and other relevant studies. We also made a table listing similarities and differences between different foundation models in table A.1. The recommended papers are also relevant and are cited in the new Related works section (subsection of Supervised Learning in BCI)
>
> 4. Overlapping vs. Non-Overlapping Tokenization:
>
> We appreciate the insightful suggestion regarding tokenization strategies. Our choice of non-overlapping tokenization aligns with the original implementations of ViT and MAE tasks, where non-overlapping patches are used to avoid information leakage during reconstruction. While our current implementation uses non-overlapping segments to maximize temporal information, we acknowledge the value of exploring both approaches for enhanced flexibility and generalization.
>
> Additionally, tokenization with varying segment lengths, as demonstrated in the Swin Transformer (Liu et al., 2021), presents a promising avenue for future work. We have included a discussion on these possibilities in the Future Work subsection of the new Discussion section.
>
> 5. Addition of a Discussion Section:
>
> In response to the reviewer’s suggestion, we have added a dedicated discussion section. This section explores the implications of our findings, highlights limitations (e.g., computational demands and variability in EEG data), and suggests future research directions, including:
>
> a. Lightweight adaptations of ST-EEGFormer for real-time applications.
>
> b. Pretraining on more diverse datasets to enhance generalization.
>
> c. Improved tokenization strategies to capture richer temporal and spatial information.
>
> We believe this addition enhances the overall impact of the paper and provides valuable insights for readers.

---

### Official Review · Reviewer_RZ9b · 2024-10-31

**Soundness:** 3
**Presentation:** 3
**Contribution:** 2
**Rating:** 6
**Confidence:** 4

**Summary:**

This paper explores the difficulties associated with creating a universal EEG-based brain-computer interface (BCI) by presenting a new spatiotemporal EEG transformer (ST-EEGFormer). The authors aim to enhance current BCI applications while tackling the issue of scarce labeled data in EEG studies. Their model is designed to effectively manage a range of EEG recording configurations and tasks using self-supervised learning, thereby aiming to boost data efficiency and classification accuracy. The research also highlights the ST-EEGFormer’s ability to generalize across different datasets, underscoring its significant potential for advancing EEG research and BCI technology. Ultimately, this work offers both a practical approach to existing challenges and a theoretical insight into EEG data representation.

**Strengths:**

A key strength of this research is the ability of the pretrained ST-EEGFormer to develop robust EEG representations that yield superior classification accuracies compared to benchmark models across various pretraining datasets. This highlights not only the model’s effectiveness but also its adaptability, as it demonstrates strong generalization on new datasets, even with limited training data. Furthermore, the incorporation of visualizations of model features improves the interpretability of the results, shedding light on the model's functioning. By making the pretrained model weights and code publicly accessible, the study promotes transparency and encourages further exploration in the field, thereby enhancing its overall impact.

**Weaknesses:**

Although the ST-EEGFormer demonstrates strong performance, its effectiveness may be constrained when encountering entirely new or atypical datasets that significantly differ from those used in pretraining. Moreover, the model's complexity could lead to higher computational demands, presenting practical challenges for real-time BCI applications.
One potential limitation of this study is its reliance on self-supervised learning, which might not fully tackle the challenges arising from the variability of EEG data across different tasks and recording conditions. Lastly, while the paper highlights improved classification accuracies, a more thorough examination of the model's interpretability and any potential biases in its learning process would enhance the reliability of the results across various populations and contexts.

**Questions:**

In the related work section, I recommend expanding the discussion on recent developments in attention-based transformer models that have influenced BCI researchers. Specifically, incorporating citations from key foundational papers in this field would strengthen your analysis.

Including a discussion section is much more beneficial than simply reporting the results. You can put limitations and suggest future directions for research within the same context.

---

> ### Author Response · Authors · 2024-11-21
> **Response to reviewer RZ9b**
>
> We sincerely thank the reviewer for the feedback and suggestions, which have provided valuable insights that enhanced the quality and depth of our manuscript. Below, we address the main points raised:
>
> 1. Effectiveness on New or Atypical Datasets:
>
> We acknowledge the importance of examining the model to entirely new or atypical datasets that differ significantly from those used in pretraining. To address this concern, we tested the model to a seizure classification dataset (not in the pre-training datasets) and an online motor imagery (MI) BCI dataset (a new MI dataset). These experiments demonstrated the model's ability to generalize, particularly in scenarios with limited training data. While the results are promising, we recognize that further investigations, including pre-training on a broader variety of EEG datasets, are essential for improving robustness to atypical datasets. This is now highlighted in the newly added discussion section.
>
> 2. Implications for Real-Time Applications:
>
> We acknowledge that offline decoding analysis is different from online BCI application, we would like to first demonstrate the model's strong performance and versatility across diverse BCI tasks. In our revised manuscript, we have also included practical information regarding model pre-training and calibration. Although performing pre-training from scratch is time-consuming, we showed that performance could be improved by just performing a short calibration on a new dataset.
>
> 3. Interpretability and Biases:
>
> We appreciate the suggestion to delve deeper into the model's interpretability and potential biases. Indeed, we found that several pioneering works failed to dive into this direction. To address this, we have detailed visualization analyses of the model's learned features, including learned positional embeddings, attention rollouts and topographic plots. These provide insights into the regions and features the model focuses on during classification. Furthermore, we discuss potential biases and limitations in the newly added discussion section, emphasizing the need for broader evaluations across diverse populations and contexts.
>
> Questions:
>
> 1. Expansion of Related Work:
>
> We have expanded the related works section to include recent developments in attention-based transformer models that have influenced BCI research. Specifically, we have incorporated discussions and citations of foundational works such as PatchTST (Nie et al., 2023), iTransformer (Liu et al., 2024), and other relevant studies. We also made a table listing similarities and differences between different foundation models in table A.1.
>
> 2. Inclusion of a Discussion Section:
>
> We agree with the reviewer’s suggestion regarding the importance of a Discussion section. In the revised manuscript, we have added a dedicated discussion section that includes:
>
> o Insights from our experiments on seen and unseen datasets, emphasizing the value of further pretraining and calibration, as well as the performance comparison with the other models.
>
> o Limitations and future developments.

---

### Official Review · Reviewer_Qnd3 · 2024-11-04

**Soundness:** 3
**Presentation:** 3
**Contribution:** 2
**Rating:** 6
**Confidence:** 3

**Summary:**

The paper presents ST-EEGFormer, a large transformer-based model designed for learning representations from EEG data.  The model is channel-aware and flexible, capable of handling diverse datasets and tasks through self-supervised pretraining using the masked autoencoder approach.

**Strengths:**

1. Foundation models for EEG (and time series data in general) are not inherently designed to handle varying numbers of channels. The paper solves this issue by using both temporal and spatial positional encodings for each token, which enables the model to work with a variety of different datasets and tasks.
2. Existing foundation models for EEG such as Neuro-GPT [1] and BENDR [2] pre-trained on a single large dataset (TUH) whereas the current paper utilizes a variety of different datasets for pre-training of their model because of its ability to handle varying numbers of channels.

**Weaknesses:**

1. The novelty is not entirely clear. The authors claim that the novelty lies in their model’s ability to project segments of raw EEG data into an embedding space enriched with spatial and temporal embeddings. However, this appears to merge two existing and increasingly popular techniques in time-series research: (1) The paper by Xie et al. [3] published in 2022 uses a very similar method to encode both spatial and temporal position in a single transformer model. (2) Papers such as PatchTST [4] and Neuro-GPT [1] have performed chunking along the temporal axis.

2. While EEG-Conformer is a relevant model to compare with, It would be beneficial to the paper to conduct qualitative and quantitive comparisons with other papers that share the same spirit of foundation models for EEG such as Neuro-GPT [1], BENDR [2], and EEGFormer [5].

3. A major expectation from a foundation model is the ability to generalize on data it hasn’t seen before, i.e. data that it wasn’t pretrained on. The authors have evaluated the model on only two such datasets. It would be helpful to expand this evaluation to include more datasets that are not part of the training corpus, preferably alongside comparisons with other large EEG-models mentioned earlier.

Minor issues:
1. On line 149 of the text, it’s not very clear what the authors mean by “samll patches consist of all channels”. Perhaps, it is “small patches consisting of all channels”.
2. From 426-428, the following sentence is repeated: “However, differences were observed when additional clusters were introduced”.
3. In Table 2, it would be better to have the second-best accuracy underlined instead of bold for better distinction.

**Questions:**

1. Referring to point 1 in Weaknesses, could the authors please elaborate on the novelty of their work?

2. The original EEG Conformer paper reports an accuracy of 78.66% for the BCI Competition IV Dataset 2a, while the current paper reports EEG-conformer’s accuracy for the same dataset as 56%. For BCI Competition IV Dataset 2a, the reported accuracies in both the original (84.63%) and current paper (83.10%) are more or less the same. Can the authors kindly explain the significant difference in the reported accuracy for EEG Conformer on the BCI Competition IV Dataset 2a?

References:
1. Cui, Wenhui, et al. "Neuro-GPT: developing a foundation model for EEG." arXiv preprint arXiv:2311.03764 (2023).
2. Kostas, Demetres, Stephane Aroca-Ouellette, and Frank Rudzicz. "BENDR: Using transformers and a contrastive self-supervised learning task to learn from massive amounts of EEG data." Frontiers in Human Neuroscience 15 (2021): 653659.
3. Xie J, Zhang J, Sun J, Ma Z, Qin L, Li G, Zhou H, Zhan Y. A Transformer-Based Approach Combining Deep Learning Network and Spatial-Temporal Information for Raw EEG Classification. IEEE Trans Neural Syst Rehabil Eng. 2022
4. Nie, Yuqi, et al. "A time series is worth 64 words: Long-term forecasting with transformers." arXiv preprint arXiv:2211.14730 (2022).
5. Chen, Yuqi, et al. "EEGFormer: Towards transferable and interpretable large-scale EEG foundation model." arXiv preprint arXiv:2401.10278 (2024).

---

> ### Author Response · Authors · 2024-11-21
> **Response to reviewer Qnd3**
>
> We thank the reviewer for the valuable suggestions, which were addressed in the following way:
>
> 1. Novelty: While prior studies have introduced large EEG foundation models, the majority of them predominantly rely on well-known clinical datasets such as TUH (Obeid & Picone, 2016), which limits their adaptability to varying channel configurations (e.g., BENDR, NeuroGPT, BIOT). LaBraM, despite being pre-trained on a large amount of EEG data, focuses primarily on clinical applications such as seizure detection, abnormal EEG detection, emotion recognition, and gait prediction. In contrast, our work targets diverse BCI applications, which typically have smaller datasets compared to the cited clinical EEG ones. For the first time, we demonstrate promising results across various BCI paradigms, providing new insights into model performance. Additionally, the proposed ST-EEGFormer features simplicity in both architecture and pretraining steps, making it accessible for a broader range of applications. A detailed comparison of ST-EEGFormer with related EEG foundation models is provided in Table A1. We believe the simplicity and effectiveness of our approach offer a valuable contribution to the classic BCI field, supporting future advancements and facilitating research in this area.
>
> 2. Comparison with other papers: We thank the reviewer for suggesting additional comparisons. In the revised manuscript, we have included additional studies and acknowledged their contributions. For instance: In the introduction (line 106), we now discuss the contributions of PatchTST (Nie et al., 2023), iTransformer (Liu et al., 2024), and the spatial-transformer and temporal-transformer proposed by Xie et al. (2022). Additionally, a comparison table (Table A1) in the Appendix highlights the similarities and differences between our approach and that of others.
>
> 3. Generalization to a new dataset: We agree that generalization to unseen datasets is crucial for foundation models. To address this, we conducted two additional experiments, showcasing the generalization of our model to new datasets. Many existing works, such as BIOT (Yang et al., 2023), NeuroGPT (Cui et al., 2024), EEGFormer (Chen et al., 2024), and LaBraM (Jiang et al., 2024), provide a limited evaluation on unseen datasets. For example:
>
> a. BIOT does not conduct additional experiments on unseen data.
>
> b. NeuroGPT evaluates performance on only one motor imagery dataset.
>
> c. EEGFormer tests on one additional sleep stage dataset.
>
> d. LaBraM evaluates on one emotion recognition and one gait prediction dataset.
>
> Based on our experimental results, we provide the following novel insights relevant to future applications of our model:
>
> a. With limited training data, the pre-trained model demonstrates better generalization compared to classic models trained from scratch.
>
> b. Performance can be further improved by performing the MAE pretraining step (calibration) on the new dataset.
>
> In the revised manuscript, we also included a comparison with BIOT and LaBraM, showing that the proposed ST-EEGFormer achieves superior performance on BCI tasks.
>
> 4. Minor issues: We thank the reviewer for carefully identifying errors. We have corrected the mentioned mistakes and, for Table 2, added a box to indicate the highest accuracy for clarity’s sake.
>
> 5. Question 1: Please refer to our responses to points 1 and 2 for more details. In summary, this study conducts comprehensive experiments on classic BCI tasks and presents a model that achieves promising results. Additionally, we provide extensive visualization analyses that can aid BCI researchers.
>
> 6. Question 2: The performance difference between EEG Conformer in our manuscript and the original paper stems primarily from differing experimental settings. We evaluated the performance of a population model, while the original paper focused on subject-dependent models.
>
> The motivation for developing a population model is explained in Section 3.2. Population models are computationally efficient when benchmarking multiple datasets with numerous subjects. Moreover, they generalize task-related, subject-invariant features, and our analyses of topoplots provide insights into brain regions of importance across subjects.

---

> > ### Comment · Reviewer_Qnd3 · 2024-11-27
> >
> > Thank you authors for your response.
> >
> > This does solve some of my concerns. I have raised my score accordingly.

---

> > > ### Author Response · Authors · 2024-11-28
> > > **Thank you for your time and effort**
> > >
> > > Dear reviewer
> > >
> > > Thank you for taking the time to review our work. We greatly appreciate your comments and suggestions, which have helped improve the quality of our manuscript.
> > >
> > > We sincerely value your effort and consideration in evaluating our submission.
> > >
> > > Best wishes,
> > > The Authors

---

### Official Review · Reviewer_aBB7 · 2024-11-04

**Soundness:** 3
**Presentation:** 3
**Contribution:** 2
**Rating:** 3
**Confidence:** 5

**Summary:**

The paper proposes a spatio-temporal EEG transformer to build an EEG foundation model. The model is pretrained using a masked autoencoder (MAE) on various public datasets and demonstrates strong generalization on new datasets, outperforming existing work such as EEGNet and EEG Conformer in various classification tasks.

**Strengths:**

1. The authors conducted model pretraining using very large amounts of data from 8 EEG datasets with various tasks.
2. The proposed network effectively captures both temporal and spatial information from EEG data.
3. The method achieves state-of-the-art performance across multiple datasets.

**Weaknesses:**

Overall, it is difficult to evaluate the contribution of this work due to (1) limited novelty and (2) missing comparison with recently published work. The proposed architecture is similar or a simple version of LaBraM [2] Please see details as follows:

1. Questionable Contribution Claims: The authors make several problematic claims regarding the novelty and contributions of the proposed ST-EEGFormer. For example, in line 134-135, '...However, since their encoder module works with patches of EEG signals in fixed channels and time length, transferring between datasets remains unrealistic and challenging', this statement on current methods not being able to adapt to different datasets due to their reliance on fixed channels and time lengths is NOT true. The authors fail to mention recent published work, such as BIOT [1] (NeurIPS' 23) and LaBraM [2] (ICLR' 24), which were explicitly designed to handle diverse EEG datasets with varying channels and lengths. Therefore, the contribution claim 'This paper presents, for the first time, a large transformer-based ST-EEGFormer model that is channel-aware, flexible, and applicable across various datasets and tasks' (line 079-080) is inaccurate and misleading.

2. Inappropriate Statements in Contribution Section: In line 086, ' Upon acceptance of this paper, the code and pretrained model weights will be made open-source', this statement should NOT be in the Contribution section, as these resources are not currently available to  the EEG community or to reviewers who are evaluating the work. Similarly, in line 087-089, the authors claim 'We believe that the demonstrated effectiveness of the proposed ST-EEGFormer model can serve as a pioneering effort, encouraging further studies on developing better end-to-end EEG-BCI foundation models to accelerate BCI development and neuroscience research.' , which would be better suited to the discussion section rather than contributions.

3. Lack of Comparison with Closly-related Recent Work: This manuscript fails to conduct comparisons with recently published work, including BIOT [1] (NeurIPS' 23) and LaBraM [2] (ICLR' 24), which have made their code and pretrained weights publicy available, making direct implementation and evaluation easy. Please see the pretrained weights here:
BIOT [1] pretrained weights: https://github.com/ycq091044/BIOT/tree/main/pretrained-models
LaBraM [2] pretrained weights: https://github.com/935963004/LaBraM/tree/main/checkpoints


[1] Yang, Chaoqi, M. Westover, and Jimeng Sun. "Biot: Biosignal transformer for cross-data learning in the wild." Advances in Neural Information Processing Systems 36 (2024).

[2] Jiang, Wei-Bang, Li-Ming Zhao, and Bao-Liang Lu. "Large brain model for learning generic representations with tremendous EEG data in BCI." arXiv preprint arXiv:2405.18765 (2024).

**Questions:**

The authors should provide more details on the pretraining setup, including GPU usage and time used for pretraining.

---

> ### Author Response · Authors · 2024-11-21
> **Response to reviewer aBB7**
>
> We sincerely thank the reviewer for the thorough and insightful feedback. The comments have been instrumental in identifying key flaws and areas for improvement of our manuscript. Below, we address the reviewer’s concerns and highlight the changes we made to accommodate them:
>
> 1.	Acknowledgment of Related Work:
>
> We acknowledge the significant omission in the initial manuscript of closely related works, some of which were very recent and unknown to us at the time of submission. We appreciate the reviewer’s effort in pointing this out, and we have now thoroughly reviewed these studies.
>
> 2.	Revised Claims and Acknowledgment of Pioneering Efforts:
>
> In response to the reviewer’s concerns regarding questionable claims and sentences, we have revised the text to clarify and appropriately acknowledge the pioneering efforts of works such as PatchTST (Nie et al., 2023), BIOT (Yang et al., 2023), LaBraM (Jiang et al.,2024), and EEGFormer(Chen et al., 2024). These works are now cited in the Related Works section and referred to throughout the manuscript where relevant. Moreover, we moved the sentences regarding open source to a subsection of the Discussion, entitled “Reproducibility”.
>
> 3.	Comparison with BIOT and LaBraM:
>
> Based on the reviewer’s suggestions, we conducted a detailed comparison using the open-sourced BIOT and LaBraM models. Their performance has been benchmarked against our proposed model, and the findings are discussed in detail in the newly added Discussion section.
>
> 4.	Practical Information on Pretraining:
>
> To address the reviewer’s request for more practical details, we have added a section in the Appendix that lists GPU usage, training time, and number of CPU cores required for pretraining our model. We hope this addition will provide valuable information for researchers aiming to reproduce or extend our work.
>
> 5.	Addressing Novelty:
>
> Regarding the reviewer’s concerns about novelty, we would like to highlight three key aspects:
>
> o	Bridging the Gap in BCI Research: There is a significant gap between advanced deep-learning models and classic BCI research, where small models (and even linear models) are still widely applied to small datasets. Our work is the first to apply a foundation model approach to diverse classic BCI tasks, achieving promising results, especially with limited training data. We believe this aligns with the interests of the conference, particularly in relation to advancing neuroscience applications.
>
> o	Visualization and Interpretability: Our study places a strong emphasis on visualizations and model interpretability, exploring various methods to interpret the learned representations. This aspect, which was not stressed in prior studies, contributes to understanding the learned representations.
>
> o	A Simple Yet Effective Approach: As detailed in Appendix Table A.1, performing a masking-reconstruction task is a common pretraining strategy for developing large foundation models. Our approach leverages the simplest MAE task with a straightforward ViT architecture and still delivers promising results. This simplicity contrasts with more complex approaches, such as LaBraM, which employs a two-step pretraining strategy. While LaBraM’s approach is effective, our study demonstrates that a simpler methodology can also achieve good results, offering an alternative for researchers looking for straightforward yet impactful solutions. In the revised manuscript, we have benchmarked both BIOT and LaBraM across all of our datasets. The results suggest that our model achieves better performance on many classic BCI tasks, offering an effective alternative for BCI researchers, particularly those who may not have the computational resources to pre-train such a large model from scratch. This reinforces the practicality and accessibility of our approach to advancing EEG-based BCI research.
>
> Once again, we extend our gratitude for the reviewer’s constructive feedback, which has significantly strengthened our manuscript. We hope the revised version addresses your concerns and meets the reviewer’s expectations. We are also open to the reviewer’s suggestions to further improve our manuscript.

---

> > ### Comment · Reviewer_aBB7 · 2024-12-03
> >
> > I would like to thank all the authors for their efforts in revising the manuscript and addressing the comments. I acknowledge the response and will adjust my evaluation accordingly.

---

> > > ### Author Response · Authors · 2024-12-03
> > > **Thank you**
> > >
> > > Dear Reviewer,
> > >
> > > Thank you for taking the time and effort to review our work. We look forward to your feedback and any adjustments to your scores. We are open to further suggestions and questions and would happily address any concerns you may have.
> > >
> > > Best regards,
> > >
> > > The Authors

---

### Author Response · Authors · 2024-11-21
**Overview of the revised manuscript**

We sincerely thank the reviewers for their detailed feedback on our manuscript. We carefully considered all comments and revised our manuscript accordingly. Below, we detail how we addressed the reviewer comments:

Corrections and improvements:
o Corrected a mistake: the SSVEP dataset benchmarked in our study is the BETA dataset. (Liu, B., Huang, X., Wang, Y., Chen, X., & Gao, X. (2020). BETA: A Large Benchmark Database Toward SSVEP-BCI Application. Frontiers in Neuroscience, 14, 544547. https://doi.org/10.3389/fnins.2020.00627)

o Fixed typos and formatting inconsistencies throughout the manuscript for improved readability.

Inclusion of related studies:
o Added discussions on several additional related works, particularly focusing on similar EEG foundation models, to provide a broader and more comprehensive context for our study.

o In line with common practice, the Related Works section has been moved to the first chapter in the Appendix.

Comparison with new relevant models:
o Conducted thorough benchmarking of two recent foundation models, BIOT and LaBraM, using the same datasets as for our model.

Addition of a discussion section:
o Added a discussion section on the results of our findings, limitations of our study, and potential future research directions.

Addition of practical details:
o Included a new section in the Appendix providing practical details about pretraining the model, including GPU usage, training time, and the number of CPU cores utilized.

Major modifications are highlighted in yellow so that they can be easily spotted. We believe these enhancements significantly strengthen the manuscript by addressing concerns about novelty, by expanding the comparative evaluation, and by improving its structure.

We look forward to your feedback and suggestions.

---

### Comment · Area_Chair_anvH · 2024-11-23
**Rebuttal**

Dear Reviewers,

I encourage you to review the rebuttal and reach out to the authors with any additional questions or requests for clarification.

Best,\
AC

---

### Comment · Area_Chair_anvH · 2024-11-25
**Discussion**

Dear Reviewers,

As we approach the end of the discussion period, I would like to encourage you again to review the rebuttal and engage with the authors if you have any additional questions or need further clarification. If you have no further questions, please acknowledge that you have received and reviewed the rebuttal.

Best,\
AC

---

### Meta-Review · Area_Chair_anvH · 2024-12-20

**Metareview:**

The paper introduces ST-EEGFormer, a spatiotemporal transformer-based EEG foundation model trained using a self-supervised masked autoencoder for EEG. The paper has a number of notable strengths. First, it can handle diverse EEG configurations and tasks effectively with its use of temporal and spatial positional embeddings. It performs well on unseen datasets, demonstrating robustness with limited labeled data. The benchmarks used in the paper include diverse BCI applications, outperforming baselines like EEGNet and EEG Conformer. The pretrained weights and code are publicly available (upon publication).

However, the paper also has a few shortcomings. First, it relies heavily on existing MAE and transformer methodologies, with minimal architectural innovation. Moreover, the benchmarking could have been more comprehensive, including additional key models such as Neuro-GPT and BENDR. Lastly, the key issue with the paper is its limited novelty following a large number of existing foundation models already available at this stage. When asked about this, the authors responded "we believe our proposed model distinguishes itself through its simplicity in both architecture and training process". I find this argument quite unconvincing. It should be clearly discussed, and backed by empirical experiments, that "how" exactly the model is more "simple", and more importantly, "why" is this simplicity important/useful.

**Additional Comments On Reviewer Discussion:**

The paper received 3, 6, 6, 6, 6. The key issue raised by the reviewers, and later emphasized by Reviewer aBB7 (score: 3) and supported by Reviewer ST9f (score: 6), is the lack of novelty and clear distinction against existing methods in other areas and over existing foundation models for EEG. In response to this, the authors had mentioned in their rebuttal that the paper is distinguished through its simplicity. However, the lack of justification about "how" and "why" the simplicity versus existing methods is advantageous was not explored in depth. Accordingly, I believe the paper requires further experiments and discussions to position itself properly with respect to existing works in this area.

---

### Decision · Program_Chairs · 2025-01-22

Reject